# Learning From Dictionary: Enhancing Robustness of Machine-Generated Text Detection in Zero-Shot Language via Adversarial Training

**Yuanfan Li**[†,*]**, Qi Zhou**[†]**, Zexuan Xie**[†]
Faculty of Electronic and Information Engineering, Xi'an Jiaotong University
[†] Equal contribution, [*] Corresponding author: `liyuan7716@gmail.com`

## Abstract

Machine-generated text (MGT) detection is critical for safeguarding online content integrity and preventing the spread of misleading information. Although existing detectors achieve high accuracy in monolingual settings, they exhibit severe performance degradation on zero-shot languages and are vulnerable to adversarial attacks. To tackle these challenges, we propose a robust adversarial training framework named **T**ranslation-based **A**ttacker **S**trengthens Mul**T**ilingual Def**E**nder (Taste). Taste comprises two core components: an attacker that performs code-switching by querying translation dictionaries to generate adversarial examples, and a detector trained to resist these attacks while generalizing to unseen languages. We further introduce a novel Language-Agnostic Adversarial Loss (LAAL), which encourages the detector to learn language-invariant feature representations and thus enhances zero-shot detection performance and robustness against unseen attacks. Additionally, the attacker and detector are synchronously updated, enabling continuous improvement of defensive capabilities. Experimental results on 9 languages and 8 attack types show that our Taste surpasses 8 SOTA detectors, improving the average F1 score by **0.064** and reducing the average Attack Success Rate (ASR) by **3.8%**. Our framework offers a promising approach for building robust, multilingual MGT detectors with strong generalization to real-world adversarial scenarios. Our codes are available in `https://github.com/Liyuuuu111/MGT-Eval`, and our datasets and pretrained checkpoint are available in `https://drive.google.com/file/d/1w1hbdiZMS_JzPntVMWM3qrTQ4KxJf-t6`.

## 1 Introduction

The explosive growth of large language models (LLMs) such as ChatGPT (Achiam et al., 2023), LLaMA (Dubey et al., 2024) and Deepseek (Guo et al., 2025), has made it trivially easy to generate fluent, human-quality text at scale. While this capability unlocks many positive applications, it also facilitates the mass production of misleading or malicious content—ranging from fabricated news to sophisticated phishing messages—without clear attribution. To address these risks, researchers have developed various methods to detect machine-generated text (MGT) (Su et al., 2023; Hans et al., 2024; Bao et al., 2024; Liu et al., 2024; Li et al., 2025; Park et al., 2025) that help find where the text really comes from and let readers know.

Although there has been much progress in detecting MGTs, a recent benchmark (Wang et al., 2024c) shows a surprising result: *all* detectors suffer a significant drop in performance when applied to zero-shot languages. Even worse, as shown in Figure 1, their robustness on zero-shot languages degrades even further: changing just two words can cause the detectors' detection accuracy drops about 20-40%. This means that detectors can be easily bypassed in languages they were not trained on. However, most detectors are still developed and benchmarked primarily on English (Guo et al., 2023; He et al., 2024; Li et al., 2024), leaving billions of non-English readers with little protection.

**Motivation.** We rely on *adversarial training* to improve the MGT detector's zero-shot language detection performance and robustness against different kinds of attacks. Current adversarial training methods (Hu et al., 2023; Koike et al., 2024; Li et al., 2025) mainly focus on monolingual settings, which makes it difficult for detectors to generalize to multilingual environments. Moreover, some recent studies (Avram et al., 2025; Wu et al., 2025) find that adversarial training generally requires large-scale corpora to ensure multilingual robustness.

However, in the real world, high-resource languages can readily access abundant training data, whereas medium-resource languages often lack sufficient annotated corpora and mostly depend on translation dictionaries. In the case of low-resource languages, even basic lexical resources are scarce, making it almost impossible to collect data for adversarial training.

To overcome these limitations, we propose an effective adversarial training framework that works in multilingual setting. Our approach relies solely on data and labels from a single language and utilizes translation dictionaries from medium-resource languages, enabling the detector to generalize better to zero-shot languages and maintain strong robustness under attacks.

**Our work.** In this paper, we propose an adversarial training framework for training a robust multilingual MGT detector, named **T**ranslation-based **A**ttacker **S**trengthens

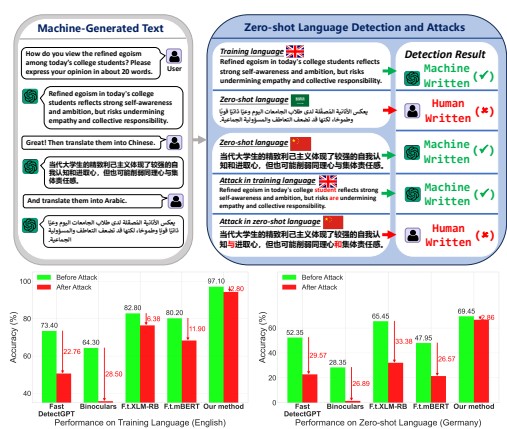

Figure 1: Performance drop of different MGT detectors under zero-shot language detection and attacks.

Mul**T**ilingual Def**E**nder (TASTE). TASTE consists of an attacker and a detector. The detector learns to distinguish between human-written texts (HWTs) and MGTs, and acquires language-agnostic feature representations during adversarial training to enhance its generalization ability to zero-shot languages, while the attacker performs code-switching on MGTs by querying translation dictionaries to deceive the detector. Due to the attacker is restricted by the black-box scenario where **only outputs** from the target detector are available, we use an open-source surrogate model to obtain gradient information and identify important tokens in the prediction. After that, we use a static translation dictionary to replace the original English tokens to other languages and obtain the adversarial examples. To encourage the detector to learn language-agnostic feature representations, we design a loss function named Language-Agnostic Adversarial Loss (LAAL), which enforces consistent predictions on original and adversarially perturbed samples across languages, thus improving zero-shot generalization and robustness. During the training stage, we update the detector and the attacker within the same training step, enabling the detector to learn from adversarial examples and generalize to zero-shot languages as well as unseen attacks. Experimental results demonstrate that our method achieves an average F1 score of **0.773** on the M4GT dataset (Wang et al., 2024b), surpassing the SOTA detector by **0.064**. We also find that TASTE achieves an average Attack Success Rate (ASR) of **18.0%** under various attack scenarios, which is **3.8%** lower than all existing SOTA detectors.

To sum up, our contributions are as follows:

- **Effective Adversarial Training Framework.** We propose TASTE, an adversarial training framework that improves a detector's generalization and robustness to zero-shot languages. The attacker generates adversarial examples by querying translation dictionaries, while the detector learns from both adversarial and original examples and is updated synchronously with the attacker, enabling it to withstand unseen attack strategies. Our TASTE only relies on translation dictionaries instead of large-scale multilingual corpora, which greatly reduces data collection costs.

- **Language-Agnostic Adversarial Loss.** We introduce a simple yet powerful LAAL objective for multilingual detector training in the black-box setting. LAAL couples a gradient-reversal language discriminator with the main classifier, forcing hidden representations to

discard language-specific cues and retain only semantic signals that transfer across languages, enabling the detector to gain in zero-shot accuracy and robustness against unseen attacks.

- **Outstanding Performance.** Evaluation on 9 languages shows that our detector outperforms 8 SOTA detectors in zero-shot language detection performance and improves robustness under 8 attacks.

## 2 RELATED WORK

**Machine-Generated Text (MGT) Detectors.** Due to the rapid development of LLMs, numerous MGT detectors have been proposed. Many works (Mitchell et al., 2023; Bao et al., 2024; Liu et al., 2024; Kushnareva et al., 2024; Li et al., 2025; Su et al., 2025; Ma et al., 2025; Su et al., 2025; Liu et al., 2025) have focused on accurately identifying machine-generated texts, motivated by the remarkable ability of LLMs to produce fluent, coherent, and human-like content, which has facilitated the rapid spread of unverified or misleading information online. Although current MGT detectors have made significant progress in detection accuracy, recent studies show that their performance drops substantially on zero-shot languages and that they remain highly vulnerable to attacks. A recent benchmark (Macko et al., 2024a) evaluated the zero-shot language detection performance of ten detectors and found that all of them suffered from severe drops in accuracy when applied to zero-shot languages. Some studies (Wang et al., 2024a; Macko et al., 2024b; Wang et al., 2024c; Li et al., 2024) have also found that existing MGT detectors fail to maintain robustness under attacks; even modifying a few words can change the detector's prediction. To address the above limitations of current MGT detectors, our work focuses on improving their zero-shot language detection performance and robustness against attacks.

**Adversarial Training.** Adversarial training aims to optimize the detector to maintain correct predictions on adversarial examples, which are intentionally crafted to mislead the detector. Hu et al. (2023) uses an LLM to generate adversarial examples via paraphrasing attacks. Koike et al. (2024) applies in-context learning for adversarial training. Li et al. (2025) improves detector robustness by updating the attacker and the detector simultaneously within the same training step. However, although these methods have made significant progress in enhancing model robustness, they cannot generalize to multilingual scenarios. Additionally, some studies (Avram et al., 2025; Wu et al., 2025) have pointed out that multilingual adversarial training requires a large amount of multilingual data, which is difficult to collect in practice. Different from previous works, our method uses an attacker that performs code-switching with translation dictionaries to create diverse multilingual adversarial samples. Meanwhile, we design Language-Agnostic Adversarial Loss to encourage the detector to learn language-agnostic features, and we update the attacker and the detector synchronously to enable the detector to generalize to unseen attacks.

## 3 MODEL ASSUMPTION

In this section, we follow standard modeling practices (Koike et al., 2024; Li et al., 2025) and specify the capabilities and goals of both the attacker and the detector in our adversarial-training setup.

**Attacker's Capability and Goals.** Given an MGT passage, the attacker aims to alter the text so that the detector misclassifies it. An MGT sample that successfully triggers a misclassification is called an *adversarial example*. Following prior work (Jin et al., 2020; Li et al., 2020b; 2025), we assume the attacker in a **black-box** setting: it has no access to the detector's weights, architecture, or training data, and can observe only the detector's **output labels**. The attacker *does* have access to (i) multilingual translation dictionaries and (ii) any publicly available models, which it can use to craft strong code-switching adversarial examples.

**Detector's Capability and Goals.** The detector's goal is to correctly distinguish HWTs from MGTs and remain robust to adversarial examples. During training, the detector can access only a labeled corpus in a single high-resource language and the adversarial samples produced by the attacker. It has *no direct access* to the translation dictionaries or to any large-scale multilingual corpora. Because the detector does not know in advance which attack strategies or code-switching patterns the attacker will deploy, it must learn language-agnostic feature representations from this limited supervision, enabling it to generalize to zero-shot languages and unseen attack strategies.

## 4 METHODOLOGY

In this section, we introduce our TASTE framework. The architecture of TASTE is shown in Figure 2. Following the general adversarial training procedure introduced in Li et al. (2025), we first describe the workflows of our attacker and detector, and then systematically outline the adversarial training process.

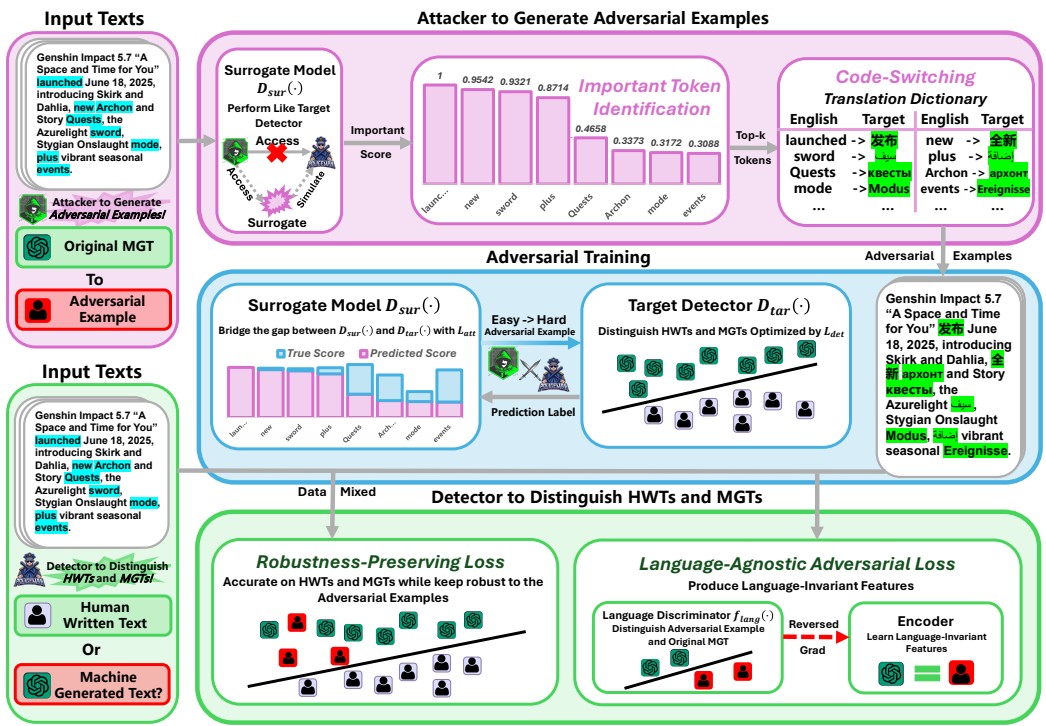

Figure 2: **Pipeline of TASTE.** The attacker identifies important tokens in original MGT and performs code-switching for important tokens to obtain adversarial examples, which are then fed to the target detector and participate in adversarial training.

### 4.1 ATTACKER TO GENERATE ADVERSARIAL EXAMPLES

Our attacker includes a surrogate detector $\mathcal{D}_{\text{sur}}(\cdot)$. To achieve the *attacker's goal* stated in *Model Assumption* Section, we develop a powerful attacker. Specifically, the attacker generates adversarial examples through the following steps: Important Token Identification and Code-Switching.

**Important Token Identification.** Miralles-González et al. (2025) demonstrate that not all tokens contribute equally during MGT detection. Therefore, perturbing key tokens is crucial for generating strong adversarial examples. However, under black-box setting, we cannot directly use the gradients of the target detector $\mathcal{D}_{tar}(.)$ to identify important tokens as done in Yoo & Qi (2021). To address this, we introduce a surrogate model to simulate the detector and estimate token importance scores.

Given an original MGT $X = [x_1, x_2, \ldots, x_T]$ consisting of $T$ tokens and its label $l$, we leverage a surrogate model $\mathcal{D}_{\text{sur}}(\cdot)$ to obtain the last-layer hidden states for all tokens:

$$H = \mathcal{D}_{\text{sur}}(X) = [\mathbf{h}_1, \mathbf{h}_2, \ldots, \mathbf{h}_T], \tag{1}$$

where $\mathbf{h}_t$ denotes the final hidden state corresponding to the $t$-th token $x_t$ produced by $\mathcal{D}_{\text{sur}}(\cdot)$.

Following the approach in Zhou et al. (2024), we compute the importance score $g_t$ of token $x_t$ using the following formula:

$$\mathcal{L}_{\text{cls}} = -\log P_{\text{sur}}(l \mid X), \tag{2}$$

$$g_t = \left\| \frac{\partial \mathcal{L}_{\text{cls}}}{\partial \mathbf{h}_t} \right\|_2, \quad t = 1, 2, \ldots, T, \tag{3}$$

where $P_{\text{sur}}(l \mid X)$ denotes the probability that the surrogate model $\mathcal{D}_{\text{sur}}(\cdot)$ assigns to the true label $l$ given the input $X$. We show the effective of our important estimator in Appendix B.2.

Then, we select the top-$k$ tokens with the highest importance scores in text $X$ and construct the important-token set $\mathbf{I}$:

$$\mathbf{I} = top-k\left([(x_t, g_t) \mid t = 1, 2, \ldots, T]\right). \tag{4}$$

**Code-Switching.** We perform code-switching on each token in $\mathbf{I}$ to obtain a multilingual adversarial example $\tilde{X}$. Specifically, given a translation dictionary $\mathbf{T}$, we query the translation for each token and replace it with the corresponding word in the target language. An example can be found in Figure 2. We investigate the impact of dictionary errors on the results of our method in Section 6.2.

**Attacker's Loss.** For each sample $X_i$ in a batch of size $N$, since we only have access to the **output label** $\hat{l}_i$ returned by the target detector $\mathcal{D}_{\text{tar}}(\cdot)$, we take $\hat{l}_i = \arg\max \mathcal{D}_{\text{tar}}(X_i)$ as a pseudo label, and let $P_{\text{sur}}(\hat{l}_i \mid X_i)$ denote the probability assigned to this label by the surrogate. The attacker loss is then defined as

$$\mathcal{L}_{\text{att}} = \frac{1}{N} \sum_{i=1}^{N} \left[-\log P_{\text{sur}}(\hat{l}_i \mid X_i)\right]. \tag{5}$$

Minimizing loss function Eq. (5) distills the evolving knowledge of the target detector into the surrogate model, providing more reliable gradients for token-level importance estimation. We demonstrate in Appendix B.3 that our distillation strategy is effective, and we analyze the impact of the label noise in Appendix B.4. We also evaluate the quality of our adversarial example in Appendix B.6.

In each training step, we update the surrogate model using the loss function Eq. (5) while keeping the detector frozen. Detailed procedures are described in Section 4.3.

## 4.2 DETECTOR TO DISTINGUISH HWTS AND MGTS

Our detector includes a target detector $\mathcal{D}_{\text{tar}}(\cdot)$. To achieve the *detector's goal* stated in the *Model Assumption* section, we design two loss functions for the detector: a robustness-preserving loss and a language-agnostic adversarial loss.

**Robustness-Preserving Loss.** The goal of the Robustness-Preserving Loss (RPL) is to keep the detector accurate on HWTs and MGTs while remaining resistant to the adversarial samples produced by the attacker. Formally, let $\mathcal{D}_{\text{tar}}(\cdot)$ be the target detector, $X_i$ an original MGT with ground-truth label $l_i$, and $\tilde{X}_i$ its adversarial example generated by the attacker. The detector is required to assign the same correct label to both $X_i$ and $\tilde{X}_i$. Accordingly, the loss is defined as

$$\mathcal{L}_{\text{RPL}} = \frac{1}{N} \sum_{i=1}^{N} \left[-\log P_{\text{tar}}(l_i \mid X_i) - \log P_{\text{tar}}(l_i \mid \tilde{X}_i)\right], \tag{6}$$

where $N$ is the batch size and $P_{\text{tar}}(l_i \mid X_i)$ denotes the probability that the target detector assigns to the correct label $l_i$ for sample $X_i$. Minimizing loss function (Eq. (6)) encourages the detector to maintain its decision on the original sample even after adversarial perturbation, thereby preserving robustness.

**Language-Agnostic Adversarial Loss.** The Language-Agnostic Adversarial Loss (LAAL) objective encourages the encoder to produce *language-invariant* features so that the detector can generalize to zero-shot languages. Encouraged by Bhattacharjee et al. (2024), we adopt a domain-adversarial scheme: the [CLS] vector is fed to a lightweight *language discriminator* $f_{\text{lang}}(\cdot)$ that tries to tell whether an input is *code-switched* ($y_{\text{lang}}^{(i)} = 1$) or *clean* ($y_{\text{lang}}^{(i)} = 0$). During backpropagation, the gradient flowing from the discriminator to the detector's encoder is multiplied by a negative factor $-\lambda_{\text{lang}}$; this *reversal* forces the encoder to confuse the discriminator and thereby erase language-specific cues.

Formally, let $\mathbf{h}_{\text{cls}}^{(i)}$ be the [CLS] representation of the $i$-th sample ($X_i$ or $\tilde{X}_i$) and $\mathbf{p}_{\text{lang}}^{(i)} = \text{softmax}\left(f_{\text{lang}}(\mathbf{h}_{\text{cls}}^{(i)})\right)$ the discriminator's predicted probability vector. The LAAL term is

$$\mathcal{L}_{\text{LAAL}} = -\frac{1}{N} \sum_{i=1}^{N} \left(y_{\text{lang}}^{(i)} \log p_{\text{lang}}^{(i)} + (1 - y_{\text{lang}}^{(i)}) \log(1 - p_{\text{lang}}^{(i)})\right), \tag{7}$$

---

**Algorithm 1** Adversarial Training Procedure of TASTE

---

1: **Input:** Training set $D_{\text{train}}$, translation dictionaries $\mathcal{T}$.
2: **Initialize:** Target detector $\mathcal{D}_{\text{tar}}$, surrogate detector $\mathcal{D}_{\text{sur}}$, language discriminator $f_{\text{lang}}$, attack strength $s = 1$, maximum strength $s_{\text{max}}$, epoch $= 0$.
3: **while** $epoch < E_{\text{max}}$ **do**
4:     **for** mini-batch $\{(X_i, l_i)\}_{i=1}^{N} \subset D_{\text{train}}$ **do**
                        ***Step A — update attacker***
5:         $\tilde{D} \leftarrow \varnothing$
6:         **for** each MGT $X_i$ in the batch **do**
7:             Obtain hidden states $H$ via Eq. (1)
8:             Compute token scores $g_t$ via Eq. (3)
9:             Select $\mathbf{I}$ with size $|\mathbf{I}| = s$ using Eq. (4)
10:             Code-switch tokens in $\mathbf{I}$ with $\mathcal{T}$ to get $\tilde{X}_i$
11:             $\tilde{D} \leftarrow \tilde{D} \cup \{(\tilde{X}_i, l_i)\}$
12:         **end for**
13:         Get the detector's return label of $X_i$ in mini-batch using $\hat{l}_i = \arg\max \mathcal{D}_{\text{tar}}(X_i)$.
14:         Compute attacker loss $\mathcal{L}_{\text{att}}$ (Eq. (5)) on mini-batch $\{(X_i, \hat{l}_i)\}_{i=1}$
15:         Update $\mathcal{D}_{\text{sur}}$ (detector frozen)
                        ***Step B — update detector***
16:         Compute $\mathcal{L}_{\text{RPL}}$ (Eq. (6)) on $D_{\text{train}} \cup \tilde{D}$
17:         Compute $\mathcal{L}_{\text{LAAL}}$ (Eq. (7))
18:         $\mathcal{L}_{\text{det}} = \mathcal{L}_{\text{RPL}} + \lambda_{\text{lang}} \mathcal{L}_{\text{LAAL}}$ (Eq. (8))
19:         Update $\mathcal{D}_{\text{tar}}$ and $f_{\text{lang}}$ (attacker frozen)
                        ***Step C — schedule attack strength***
20:         $s \leftarrow \min(s + \Delta, s_{\text{max}})$                    ▷ gradually increase difficulty
21:     **end for**
22:     $epoch \leftarrow epoch + 1$
23: **end while**
24: **Output:** Trained target detector $\mathcal{D}_{\text{tar}}$.

---

where $N$ is the batch size and $p_{\text{lang}}^{(i)}$ denotes the probability assigned to the class *code-switched*.

The discriminator minimizes loss function Eq. (7) to improve its classification accuracy, while the encoder receives the *reversed* gradient $-\lambda_{\text{lang}} \frac{\partial \mathcal{L}_{\text{LAAL}}}{\partial \mathbf{h}_{\text{cls}}^{(i)}}$ and therefore maximizes the same objective.

This adversarial interplay drives the hidden representations toward language-agnostic subspaces, enabling the detector to transfer its decision boundary to unseen languages. We provide a formal proof of the cross-lingual generalization ability of LAAL in Appendix C.

**Total Loss.** The total loss of our detector is defined as:

$$\mathcal{L}_{\text{det}} = \mathcal{L}_{\text{RPL}} + \lambda_{\text{lang}} \mathcal{L}_{\text{LAAL}}. \tag{8}$$

In each training step, we update the detector using the loss function Eq. (8) while keeping the attacker frozen. The detailed procedure is described in Section 4.3.

## 4.3 ADVERSARIAL TRAINING

We propose an adversarial training framework where the detector and the attacker are updated within the same training step to achieve co-evolution. Unlike traditional adversarial training approaches that use a fixed adversarial intensity (Jia et al., 2022; Feng et al., 2025; Gaber et al., 2025) or static adversarial examples (Zeng et al., 2023; Huang et al., 2024), our attacker adopts a *dynamic adversarial strength* strategy and generate dynamic adversarial examples. As training progresses, the difficulty of adversarial examples generated by the attacker gradually increases, thereby enhancing the detector's generalization ability and mitigating underfitting.

**Training Process.** In each training step $t$, we first update the attacker using the loss function Eq. (5) while keeping the detector frozen. The updated attacker then generates the adversarial examples $\tilde{X}$. Next, we update the detector using the loss function Eq. (8) while keeping the attacker frozen. During training, we control the attack strength via the size of code-switching tokens set $\mathbf{I}$. As

training progresses, $|\mathbf{I}|$ gradually increases, enabling the detector to learn from progressively harder examples. We provide detailed descriptions of the adversarial training process in Algorithm 1, and we study the impact of attack strength on the experimental results in Section 6.1. We also analyze the training dynamics in Appendix B.9.

## 5 EXPERIMENTS AND RESULTS

We conducted comprehensive experiments to validate the zero-shot language detection capability of TASTE and its robustness against attacks. Due to space constraints, we present the ablation study results in the Appendix B.1.

### 5.1 DETECTION PERFORMANCE IN MULTILINGUAL SETTING

**Experiment Setting.** We evaluate the zero-shot cross-lingual detection capability of our TASTE model on nine languages from the M4GT dataset (Wang et al., 2024b). The compared baselines include: (i) Metric-based detectors: Fast DetectGPT (Bao et al., 2024), Binoculars (Hans et al., 2024), and LRR (Su et al., 2023); and (ii) Model-based detectors: XLM-RoBERTa-Base (Conneau et al., 2019), mDeBERTa-v3-Base (He et al., 2021), mBERT (Devlin et al., 2019), RADAR (Hu et al., 2023), and GREATER-D (Li et al., 2025). All model-based detectors are trained in the same *English dataset*. For fair comparison, all adversarial training methods share the same backbone model mBERT. Our TASTE constructs adversarial samples using translation dictionaries for Arabic, Chinese, German, and Russian. We use a GPT-2 model (Solaiman et al., 2019) fine-tuned on the dataset from Wang et al. (2024a) as the surrogate model, and we provide a detailed analysis of the impact of replacing the surrogate model on the experimental results in Appendix B.3. Detailed descriptions of the baseline models and implementation details are provided in the Appendix A.4. We also report inference time in the test dataset in Appendix B.8.

| Lang | Metric | Metric-based Detectors | | | Model-based Detectors | | | | | |
|------|--------|----------------|------------|-------|------------------|-----------------|-------|--------|-----------|--------------|
| | | Fast DetectGPT | Binoculars | LRR | XLM-RoBERTa-Base | mDeBERT-v3-Base | mBERT | RADAR* | GREATER-D * | TASTE (Ours)* |
| *en* | *Acc* ↑ | 0.734 | 0.643 | 0.473 | 0.828 | 0.839 | 0.802 | 0.883 | 0.918 | **0.971** |
| | *F1* ↑ | 0.738 | 0.631 | 0.303 | 0.823 | 0.839 | 0.802 | 0.871 | 0.893 | **0.971** |
| *ar* | *Acc* ↑ | 0.592 | 0.353 | 0.366 | 0.891 | **0.892** | 0.811 | 0.726 | 0.783 | 0.828 |
| | *F1* ↑ | 0.642 | 0.364 | 0.247 | **0.886** | 0.872 | 0.820 | 0.711 | 0.761 | 0.790 |
| *zh* | *Acc* ↑ | **0.873** | 0.843 | 0.559 | 0.715 | 0.792 | 0.684 | 0.681 | 0.653 | 0.701 |
| | *F1* ↑ | **0.873** | 0.837 | 0.401 | 0.703 | 0.793 | 0.672 | 0.642 | 0.622 | 0.689 |
| *de* | *Acc* ↑ | 0.523 | 0.283 | 0.450 | 0.654 | **0.773** | 0.479 | 0.366 | 0.331 | 0.695 |
| | *F1* ↑ | 0.642 | 0.382 | 0.205 | 0.752 | **0.833** | 0.607 | 0.493 | 0.428 | 0.781 |
| *ru* | *Acc* ↑ | **0.633** | 0.538 | 0.527 | 0.465 | 0.393 | 0.418 | 0.521 | 0.491 | 0.617 |
| | *F1* ↑ | **0.615** | 0.400 | 0.363 | 0.346 | 0.294 | 0.415 | 0.544 | 0.503 | 0.584 |
| *bg*† | *Acc* ↑ | 0.682 | 0.662 | 0.514 | 0.513 | 0.463 | 0.774 | 0.712 | 0.754 | **0.817** |
| | *F1* ↑ | 0.670 | 0.614 | 0.349 | 0.474 | 0.443 | 0.774 | 0.726 | 0.752 | **0.813** |
| *id*† | *Acc* ↑ | 0.818 | **0.870** | 0.591 | 0.580 | 0.589 | 0.830 | 0.710 | 0.773 | 0.840 |
| | *F1* ↑ | 0.765 | 0.828 | 0.324 | 0.485 | 0.500 | 0.829 | 0.701 | 0.762 | **0.836** |
| *it*† | *Acc* ↑ | 0.240 | 0.028 | 0.461 | 0.680 | **0.815** | 0.688 | 0.452 | 0.601 | 0.694 |
| | *F1* ↑ | 0.388 | 0.054 | 0.238 | 0.809 | **0.898** | 0.815 | 0.513 | 0.713 | 0.819 |
| *ur*† | *Acc* ↑ | 0.708 | 0.612 | 0.542 | 0.561 | 0.667 | 0.671 | 0.702 | **0.713** | 0.697 |
| | *F1* ↑ | 0.655 | 0.598 | 0.381 | 0.560 | 0.571 | 0.646 | 0.671 | **0.699** | 0.677 |
| *Zero-shot Avg.* | *Acc* ↑ | 0.612 | 0.543 | 0.527 | 0.584 | 0.634 | 0.741 | 0.644 | 0.710 | **0.762** |
| | *F1* ↑ | 0.619 | 0.524 | 0.323 | 0.582 | 0.603 | 0.766 | 0.653 | 0.731 | **0.786** |
| *Avg.* | *Acc* ↑ | 0.645 | 0.537 | 0.498 | 0.654 | 0.691 | 0.684 | 0.639 | 0.669 | **0.762** |
| | *F1* ↑ | 0.665 | 0.523 | 0.312 | 0.649 | 0.671 | 0.709 | 0.652 | 0.681 | **0.773** |

Table 1: **Performance of detectors in M4GT dataset.** We report accuracy (*Acc*) and F1 score (*F1*). Best in each row is **bold**, second-best is underlined. † denotes *zero-shot* languages (bg/id/it/ur); *Zero-shot Avg.* is the macro-average over these four languages. * indicates adversarial-training-based methods. We report TPR@FPR=1% of this dataset in Appendix B.7.

**Experiment Result.** We present the zero-shot cross-lingual detection performance of TASTE in Table 1 and unveil the following three key insights: i) **Highest overall performance.** TASTE achieves the best averaged scores (Acc = 0.762, F1 = 0.773), outperforming the second-best system by +0.071 Acc (+10.3 %) and +0.064 F1 (+9.0 %). ii) **Consistent gains across languages.** It delivers the top result in four language–metric pairs and the second place in others. Representative margins over the next-best method are +5.3 pp Acc / +7.8 pp F1 on English and +4.3 pp Acc / +3.9 pp F1 on Bulgarian, showing reliable improvements even on typologically diverse languages. iii) **Zero-shot**

**generalization via adversarial training.** Despite being trained solely on English, adversarial training enables TASTE to maintain strong accuracy and F1 in every language, confirming its superior generalization in truly unseen linguistic settings. Our TASTE also exhibits a certain degree of generalization ability to unseen datasets, as detailed in Appendix B.5.

## 5.2 ROBUSTNESS AGAINST ATTACKS

**Experiment Setting.** In this section, we evaluate the robustness of MGT detectors against attacks under the zero-shot detection setting. Following the robustness evaluation protocol proposed by Wang et al. (2024a), we extend the M4GT dataset with 8 common attack strategies: random deletion (Delete) (Kukich, 1992), word repeat (Repeat) (Fishchuk, 2023), word insertion (Insert) (Gabrilovich & Gontmakher, 2002), sentence swapping (Swap) (Shi et al., 2024), code-switching (Code-Switching) (Li et al., 2020a), human obfuscation (Human-Obf.) (Macko et al., 2024b), back-translation (Sennrich et al., 2015) and paraphrasing (Iyyer et al., 2018). Detailed descriptions of these attack methods are provided in the Appendix A.5.

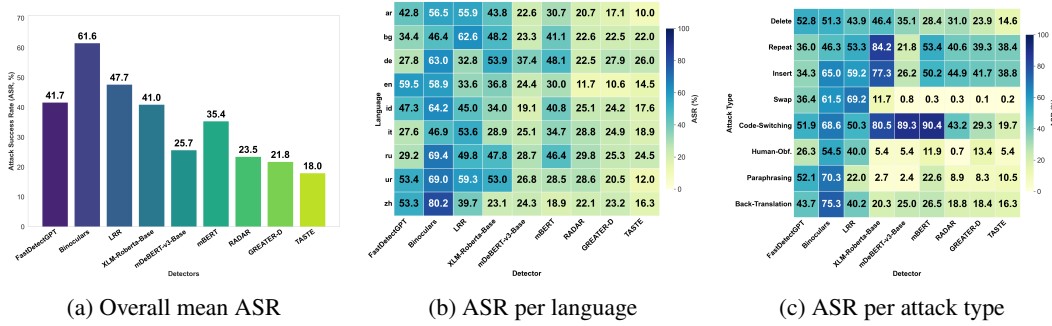

(a) Overall mean ASR  (b) ASR per language  (c) ASR per attack type

Figure 3: **Performance of all detectors under attacks.** A lower ASR (%) indicates better model performance.

**Experiment Result.** Figure 3 shows that TASTE achieves the lowest *average* ASR at **18.0%**, yielding a **17.4%** relative reduction compared with the strongest SOTA defense GREATER-D (21.8%) and a **56.8%** reduction versus the best metric-based baseline Fast DetectGPT (41.7%). Across languages, TASTE wins in **8/9** cases with an average per-language absolute drop of **3.82** ASR points (notably *ar*: $17.1 \rightarrow 10.0$, *ur*: $20.5 \rightarrow 12.0$, *zh*: $23.2 \rightarrow 16.3$), and a paired sign test over languages confirms *statistical significance* ($p = 0.039$, two-sided). Attack-wise, TASTE delivers large relative reductions against key shifts compared with GREATER-D: DELETE **–38.9%** ($23.9 \rightarrow 14.6$), CODE-SWITCHING **–32.8%** ($29.3 \rightarrow 19.7$), HUMAN-OBF. **–59.7%** ($13.4 \rightarrow 5.4$), BACK-TRANSLATION **–11.4%** ($18.4 \rightarrow 16.3$), and a further **–7.0%** on INSERT ($41.7 \rightarrow 38.8$), while maintaining near-zero vulnerability under SWAP. Collectively, these results demonstrate that the model-based, adversarially trained TASTE consistently lowers vulnerability across languages and perturbation types.

## 6 DISCUSSION

### 6.1 IMPACT OF ATTACK STRENGTH

In this section, we investigate the impact of attack strength on the zero-shot language detection capability and the robustness against adversarial attacks of the target detector.

Follow the definition in Li et al. (2025), we define the attack strength as the ratio of the maximum size of the important token set $\mathbf{I}$, denoted as $\max |\mathbf{I}|$, to the total number of tokens in the text $X$. During training, $|\mathbf{I}|$ starts from 1 and gradually increases until it reaches the upper bound $\max |\mathbf{I}|$. Hence, $\max |\mathbf{I}|$ reflects the strength of the hardest adversarial examples: a larger $\max |\mathbf{I}|$ indicates that more tokens in the text have been replaced, making it more challenging for the target detector $\mathcal{D}_{\mathrm{tar}}(\cdot)$ to correctly classify the sample.

Figure 4 presents our results. As can be observed, increasing the attack strength does not continuously enhance zero-shot detection accuracy or robustness against adversarial attacks; instead, it

leads to performance degradation beyond a certain point. Specifically, the model achieves its highest average accuracy of **76.2%** when the attack strength ratio is 0.05, but this drops to **73.1%** at 0.25. Similarly, the average ASR is minimized at **18.0%** at 0.05, but increases to **34.2%** at 0.25. This phenomenon suggests that a moderate attack strength is most effective in improving the model during adversarial training. Excessively strong perturbations, however, introduce substantial noise and cause the model to focus excessively on fitting these extreme adversarial examples. This leads to overfitting on high-strength attacks and compromises both generalization ability and robustness to unseen perturbations in zero-shot settings.

Moreover, as shown in Figure 4, the training time significantly increases with higher attack strengths: from **150.34s** at 0.00 to **2299.33s** at 0.05, and up to **9467.94s** at 0.30. This highlights a critical trade-off between improving robustness and maintaining practical training efficiency. These results collectively indicate that an optimal, moderate attack strength (here, around **0.05**) strikes the best balance between detection accuracy, adversarial robustness, and computational cost.

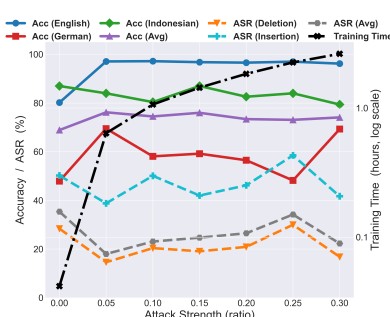

Figure 4: **Impact of Attack Strength.** A higher accuracy (Acc) or a lower ASR (%) indicates better model performance.

## 6.2 IMPACT OF WRONG DICTIONARY

In this section, we investigate how errors in the translation dictionary affect detection performance. Since *LAAL* learns language-agnostic feature representations from code-switching-based adversarial examples, erroneous dictionary entries can alter the semantics of these examples and thereby impede the detector's ability to acquire language-invariant features.

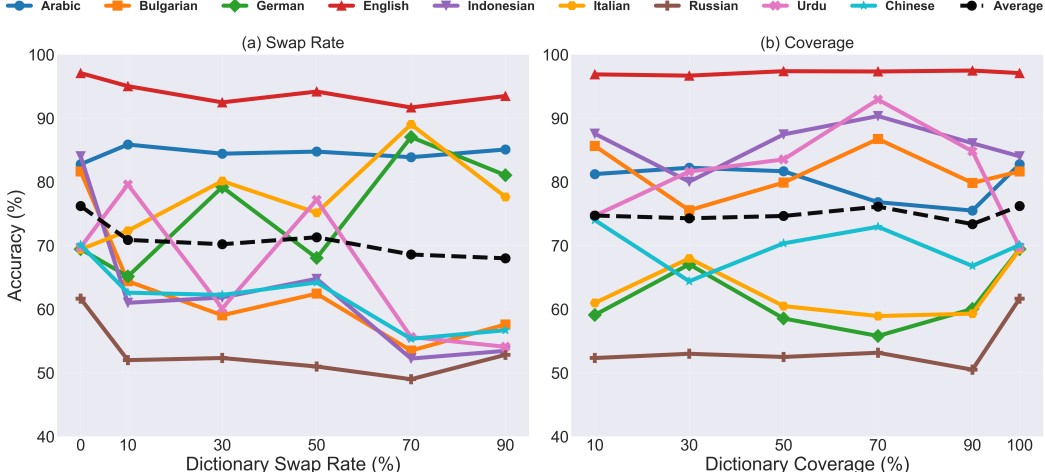

Figure 5: **Impact of wrong dictionary and dictionary coverage.** We evaluate the detection accuracy of each model for every language under different (a) dictionary swap rates and (b) dictionary coverage rate.

We define the *dictionary error rate* as the proportion of entries whose translations are incorrect, and we operationalize it via the *Dictionary Swap Rate*, i.e., the percentage of dictionary entries that are deliberately swapped with incorrect translations. We train detectors with dictionaries at swap rates of $[10, 30, 50, 70, 90]\%$ and evaluate them on the test set. The results are shown in Figure 5 (a). Intuitively, as the swap rate increases, the translation-based attacker provides less reliable cross-lingual supervision: when the dictionary is accurate, code-switched adversarial examples carry informative signals that propagate English supervision to other languages, whereas with highly corrupted dictionaries many swapped tokens no longer correspond to meaningful translations, and the adversarial component gradually degenerates toward standard monolingual English training. In this limit, per-

formance on non-English languages is expected to *approach* (but not necessarily match) that of an English-only baseline trained without effective multilingual adversarial perturbations.

Across languages, using a wrong dictionary consistently depresses average accuracy relative to the clean (0%) baseline. The cross-language mean Acc drops from **76.2%** at 0% to **70.9%** (10%), **70.2%** (30%), rebounding to **71.3%** (50%), before declining to **68.6%** (70%) and **68.0%** (90%). Per-language effects vary but align with this semantic-drift view: **English** remains relatively robust (97.1% at 0% vs. 95.0–91.7% for 10–70%), while languages that rely more heavily on cross-lingual transfer, such as **Chinese** and **Indonesian**, are more fragile (Chinese: 70.1% → 62.6% @10%, 56.7% @90%; Indonesian: 84.0% → 61.0% @10%, 53.5% @90%). Moderate swap rates can still help languages where code-switching reduces spurious lexical cues: **Italian** rises from 69.4% (0%) to 89.0% @70%, **German** from 69.5% to 87.1% @70%, and **Urdu** peaks at 77.2% @50% (vs. 69.7%), but collapses to 54.1% @90%. Mechanistically, LAAL benefits when small-to-medium swaps preserve most semantics and act as hard but valid augmentations that steer the encoder toward language-invariant cues. At high swap rates, however, the adversarial examples themselves become severely corrupted, effectively injecting label noise: the discriminator is then pushed to be invariant to artifacts induced by erroneous translations rather than to true semantic content, which explains the overall degradation and the language-dependent variance.

In addition, we explicitly study the effect of dictionary noise in Section 6.2 by plotting detection accuracy as a function of dictionary error rates. The results can be found in Figure 5 (b). This result show that although degrading dictionary quality slightly hurts performance, our model still remains above the baselines, demonstrating that our method can yield gains even with relatively noisy dictionaries. Since this experiment may not be fully sufficient, we further conduct an additional study where we vary dictionary coverage. Specifically, we subsample [10, 30, 50, 70, 90]% of the entries from our dictionaries to construct smaller dictionaries, train with these reduced dictionaries, and evaluate on the same test sets as in Table 1.

From these results, we observe that using very small dictionaries (10% coverage) leads to some degradation in accuracy compared to higher-coverage settings, but the performance remains strong and above the average accuracy of the best baseline detector (0.691, as reported in Table 1). As dictionary coverage increases, the model's accuracy improves and gradually stabilizes, with the mean accuracy saturating around full coverage. This indicates that our method does not rely on perfectly complete dictionaries: even moderately sized dictionaries are sufficient to provide substantial gains, and further enlarging the dictionary mainly brings diminishing but stabilizing improvements rather than being strictly required for the method to work. Therefore, even when high-quality dictionaries are unavailable for a target low-resource language, TASTE can still improve its detection performance in a zero-shot manner by leveraging dictionaries from other languages, and our noise/-coverage studies show that these dictionaries do not need to be perfectly accurate or complete.

## 7    CONCLUSION

In this paper, we propose an adversarial training framework for training a robust multilingual MGT detector, named **T**ranslation-based **A**ttacker **S**trengthens Mul**T**ilingual Def**E**nder (TASTE). We propose a novel adversarial training loss, named Language-Agnostic Adversarial Loss (LAAL), to enhance the generalization of detectors to zero-shot languages. In TASTE, our attacker and detector are updated synchronously, enabling the detector to generalize to unseen languages and unseen attacks. Experimental results demonstrate that TASTE outperforms 8 SOTA detectors in detection accuracy and remains robust under 8 different attack types. Our study further reveals that a moderate attack strength can effectively balance robustness, detection accuracy, and computational cost, and that our method remains reasonably robust to a certain degree of noise in the translation dictionaries.

## ACKNOWLEDGMENT

We thank the anonymous reviewers and AC for their careful reading and constructive feedback, which significantly improved the quality and clarity of this work.

## ETHICS STATEMENT

**Intended use and societal benefits.** This work studies robust detection of Machine-generated text (MGT) across languages, aiming to strengthen online content integrity and reduce harms from undisclosed synthetic text. We focus on improving zero-shot multilingual robustness via translation-dictionary–based code-switching attacks and adversarial training, which may help platforms, educators, and researchers identify MGT at scale while accounting for cross-lingual disparity.

**Data provenance and subject protection.** Our experiments rely on publicly available corpora and benchmarks reported in prior work (human-written texts from sources such as Wikipedia/Reddit/ELI5/WikiHow/PeerRead/arXiv; machine outputs from widely known generators) and the multilingual M4GT benchmark. No new human data were collected; no personally identifiable information (PII) was annotated or created. All datasets used in this study are treated under their original licenses and terms of use; we redistributed nothing that would violate those terms. When pre/post-processing, we avoid storing raw user identifiers and only use text content as provided by the benchmarks.

**Model behavior, bias, and fairness.** Detection errors can have downstream consequences (e.g., false positives against under-represented languages or communities). We therefore (i) report per-language results; (ii) analyze robustness under multiple perturbations; and (iii) explicitly discuss failure modes observed for certain languages. We *do not* claim perfect language equity; residual disparities may persist due to dictionary coverage and domain shifts. We encourage deployment only with human oversight, clear error disclaimers, and opt-out avenues.

**Dual-use and misuse risks.** Adversarial analysis can inform stronger *attacks*. To mitigate dual-use, we: (i) release code primarily for *evaluation and defense* with safety notes and rate limits; (ii) avoid providing turn-key evasion scripts beyond what is necessary for reproducible research; (iii) withhold any undisclosed private attack resources; and (iv) recommend pairing detectors with complementary safeguards (attribution, disclosure norms, provenance signals) rather than high-stakes automation.

**Computational resources and environmental impact.** We strive to minimize the environmental footprint of our experiments by favoring moderate-size encoders, mixed-precision training, gradient accumulation, and early stopping. We will release a reproducible log of hardware, training steps, and GPU hours together with an estimated carbon footprint (via standard calculators and region-specific carbon intensity), and we encourage labs reproducing our results to adopt renewable-powered compute where available. Our ablations are designed to reuse checkpoints and shared dataloaders to avoid redundant runs.

**Transparency, accountability, and limitations in deployment.** We will provide model cards, datasheets, and evaluation scripts to document intended use, coverage, and known failure modes (e.g., languages with lower robustness or domains unseen in training). We caution against punitive or high-stakes use (e.g., grading, employment, or moderation actions) without human review and an appeals process, and we recommend periodic audits, per-language calibration, and monitoring for distribution shift. Access to released artifacts will include versioning and changelogs to support accountability and responsible updates.

## REPRODUCIBILITY STATEMENT

**Code and artifacts.** We commit to releasing: (1) training/evaluation code; (2) datasets used in this paper; (3) trained detector checkpoints; and (4) the translation-dictionary resources we used or scripts to construct them from public sources (with licenses).

**Data and splits.** We describe all datasets, languages, and splits used for training/validation/test in Appendix A and provide scripts to regenerate them deterministically from public releases.

**Training details.** We enumerate all hyperparameters (optimizer, learning rates for detector/surrogate, batch sizes, epochs, `fp16`, gradient clipping), schedules (e.g., GRL weight schedule), and attack-strength curriculum (initial tokens, maximum ratio, increment per step) in Appendix A.

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

# A EXPERIMENT SETTING

## A.1 DATASET

We use the M4GT dataset (Wang et al., 2024b) as the foundational training corpus for our detector. This large-scale dataset comprises machine-generated texts (MGTs) produced by models such as Davinci003, ChatGPT, GPT-4, Cohere, Dolly-v2, and BLOOMZ, along with human-written texts sourced from platforms including Wikipedia, Reddit, ELI5, WikiHow, PeerRead, and arXiv abstracts. The corpus spans nine languages—English, Chinese, Russian, Arabic, Italian, German, Indonesian, Bulgarian, and Urdu—ensuring broad multilingual coverage.

## A.2 IMPLEMENTATION

TASTE is implemented and deployed on a server equipped with two NVIDIA RTX 4090 GPUs, running Ubuntu 22.04 and CUDA 12.5. For all experiments in the main text and the appendix, we used only a single RTX 4090 GPU. For all adversarial-training-based detectors, we adopt mBERT (110M) as the backbone model. Each model-based detector is trained on **10,000** English samples, with a standard data split ratio of 8:1:1 for training, validation, and testing, respectively. For metric-based detectors that rely on LLMs, we use the best-performing LLMs as reported in their original papers. All baseline detectors follow the official configurations and implementation details described in their corresponding publications.

| Hyperparameter | Value / Schedule |
|---|---|
| Batch size $M$ | 16 |
| Training epochs $E_{\max}$ | 3 |
| Detector learning rate $\eta_{\text{det}}$ | $2 \times 10^{-5}$ |
| Surrogate learning rate $\eta_{\text{sur}}$ | $1 \times 10^{-5}$ |
| Adversarial–standard mix ratio | 0.15 (adv) : 0.85 (std) |
| Initial attack strength $s$ | 1 token |
| Maximum attack strength $s_{\max}$ | 5 % of tokens |
| Attack-strength increment $\Delta$ | +1 token / step |
| $\lambda_{\text{lang}}$ (GRL weight) | $0.10 \rightarrow 0.50$ |
| Dropout (language discriminator) | 0.20 |
| Optimizer | Adam |
| Gradient precision | FP16 (AMP) |
| Random seed | 42 |

Table 2: Key hyperparameters used for training TASTE.

We use a GPT2-based detector fine-tuned on the dataset from Wang et al. (2024a) as the surrogate model. As discussed in Appendix B.3, the choice of surrogate model and training corpus has minimal impact on the final results.

The hyperparameters used in our method are shown in Table 2.

To enforce language-invariant representations, we attach a lightweight language discriminator $f_{\text{lang}}(\cdot)$ to the `[CLS]` embedding produced by the detector. Concretely, $f_{\text{lang}}(\cdot)$ is a three-layer multilayer perceptron (MLP):

$$\text{MLP: } 768 \;\rightarrow\; 512 \;\rightarrow\; 128 \;\rightarrow\; 2 \quad (\text{ReLU, Dropout} = 0.2)$$

where the final layer outputs a 2-way softmax indicating whether an input is *code-switched* (1) or *clean* (0). During training the detector's encoder is connected to $f_{\text{lang}}(\cdot)$ through a gradient-reversal layer (GRL). The discriminator itself minimizes a standard cross-entropy loss, while the GRL multiplies its back-propagated gradient by $-\lambda_{\text{DA}}$, forcing the encoder to *maximize* the same objective and thus remove language-specific cues. This adversarial interplay encourages the hidden features to reside in a language-agnostic subspace, which empirically improves zero-shot detection accuracy and robustness to multilingual adversarial attacks. With only ~0.5M additional parameters, the discriminator introduces negligible overhead while substantially enhancing cross-lingual generalization.

## A.3 EVALUATION METRICS

For evaluating the accuracy of detectors, we adopt **Accuracy (Acc)** and **F1-score (F1)** as the primary metrics. To assess the robustness of detectors against adversarial attacks, we use the **Attack Success Rate (ASR)**, where a lower ASR indicates better robustness.

Based on our attacker assumption that only **machine-generated texts (MGTs)** can be perturbed, ASR is computed as:

$$\text{ASR} = \frac{\text{\# MGTs misclassified after attack}}{\text{\# MGTs correctly classified before attack}} \tag{9}$$

## A.4 EXPERIMENTAL COMPARISON METHODS

To better evaluate the effectiveness of our method, we compare it against several state-of-the-art (SOTA) detectors. The compared detectors are as follows:

- **Fast DetectGPT** (Bao et al., 2024): A metric-based zero-shot detector that builds upon the original DetectGPT framework while addressing its computational inefficiencies. Instead of relying on costly perturbation-based scoring, Fast DetectGPT introduces a novel metric termed *conditional probability curvature*, which captures discrepancies in token-level generation likelihoods between human- and machine-authored texts. This curvature-based score enables highly efficient detection by leveraging a single-pass sampling strategy, significantly reducing computation without compromising performance. Empirical results demonstrate that Fast DetectGPT achieves comparable or superior accuracy to DetectGPT while accelerating inference by up to 340× across both white-box and black-box settings.

- **Binoculars** (Hans et al., 2024): A metric-based zero-shot detection method that identifies machine-generated text by contrasting the output probabilities of two closely related pre-trained language models. Rather than relying on supervised training, Binoculars computes a lightweight divergence score that reflects subtle inconsistencies between human-authored and machine-generated text. This model-agnostic approach requires no fine-tuning or labeled data, yet consistently achieves high detection accuracy across diverse document types and language models. Notably, Binoculars detects over 90% of ChatGPT outputs with a false positive rate as low as 0.01%, demonstrating its effectiveness and generalizability in real-world scenarios.

- **LRR** (Su et al., 2023): A metric-based zero-shot detector that leverages log-rank statistics to distinguish machine-generated text from human-written content without requiring any labeled training data. Specifically, LRR computes the average log-rank of tokens under a reference language model and identifies unnatural token distributions typical of LLM generations. As a lightweight variant of DetectLLM, LRR avoids costly perturbation steps, making it significantly faster and more efficient while maintaining strong detection performance. Empirical results across three benchmark datasets and seven generation models

demonstrate that LRR outperforms prior state-of-the-art methods by up to 3.9 AUROC points, offering a favorable balance between accuracy and inference speed.

- **mBERT** (Devlin et al., 2019): A model-based detector built on multilingual BERT, which contains 110M parameters. It is pretrained on the Wikipedia texts of 104 languages, using a shared subword vocabulary. We fine-tune this model on our classification task to distinguish machine-generated text from human-written text.

- **mDeBERTa** (He et al., 2020): A model-based detector based on multilingual DeBERTa-v3, with approximately 180M parameters. It is pretrained on multilingual web text covering over 100 languages. We fine-tune it for binary classification in our detection setting.

- **XLM-RoBERTa-Base** (Conneau et al., 2019): A model-based detector built on the XLM-R-Base model, containing 270M parameters. It is pretrained on a large-scale corpus of 2.5TB CommonCrawl data from 100 languages. We fine-tune this model to perform machine-generated text detection across languages.

- **RADAR** (Hu et al., 2023): A model-based detector trained via adversarial training. During training, a paraphraser acts as the attacker to generate semantically equivalent paraphrases, while the detector, built upon an mBERT backbone, learns to distinguish machine-generated from human-written text even after paraphrasing. This process improves the detector's robustness against paraphrased attacks. Unlike our multilingual adversarial setup, RADAR only considers monolingual paraphrase-based attacks during training.

- **GREATER-D** (Li et al., 2025): A model-based detector trained via adversarial training against token-level perturbation attacks. In each iteration, a detector and an attacker are jointly optimized: the attacker identifies and perturbs important tokens to mislead the detector, while the detector is trained to correctly classify both original and adversarial samples. This synchronized update improves the robustness of the detector against various word-level attacks. This method focuses on English-only settings and does not consider multilingual generalization or translation-based attacks.

## A.5 ATTACK METHODS

Here we provide detailed descriptions of the 8 attack methods used in the main paper.

- **Delete** (Kukich, 1992): A token-level perturbation that randomly deletes about 20% of tokens in the machine-generated text. This simulates typing errors or intentional omissions and tests the detector's robustness to missing information.

- **Repeat** (Fishchuk, 2023): A token-level attack that duplicates existing tokens in the text with a 20% probability, mimicking redundancy or emphasis used in informal writing, which can confuse the detector.

- **Insert** (Gabrilovich & Gontmakher, 2002): A token-level perturbation that inserts additional tokens at random positions. For each token, there is a 20% chance to insert another token after it, thereby increasing noise and changing context.

- **Swap** (Shi et al., 2024): A sentence-level perturbation that randomly shuffles the order of sentences in the text. This breaks the coherence of the discourse without altering individual sentence content, challenging detectors reliant on narrative flow.

- **Code-Switching** (Li et al., 2020a): A token-level perturbation that replaces up to 50 tokens with their translated equivalents using a multilingual dictionary. This introduces cross-lingual noise while preserving semantics, simulating a real-world scenario of multilingual interference.

- **Human Obfuscation** (Macko et al., 2024b): A token-level attack where the first 20% of tokens in a machine-generated text are replaced by tokens sampled from human-written text of the same language. This aims to mimic human writing style and evade detection by semantic camouflage.

- **Back-Translation** (Sennrich et al., 2015): A sentence-level paraphrase attack where each machine-generated text is round-trip translated through a pivot language to induce lexical and syntactic variation while preserving meaning. Concretely, for any non-English language we translate the text into English and then back to the original language; for English

| System Prompt | You are a ruthless paraphraser. You will aggressively and completely rewrite the user's text while preserving the core meaning and factual content. Keep the SAME LANGUAGE as the input. Do NOT add disclaimers, meta-comments, safety notes, citations, or new facts. Do NOT shorten excessively; keep roughly similar length unless instructed otherwise. Produce ONLY the rewritten text as plain text, with no surrounding quotes or markers. |
|---|---|
| User Instruction | Paraphrase the following text aggressively (completely rewrite) while preserving its core meaning. Keep the SAME LANGUAGE as the input. Output only the rewritten text. Text: {text} |

Table 3: Prompts used for the DeepSeek V3.1 paraphrasing attack.

texts we pivot through Chinese. We implement the translations with the public Helsinki-NLP OPUS-MT models (Tiedemann & Thottingal, 2020). The resulting back-translated output replaces the original text, creating semantics-preserving paraphrases that disrupt detectors relying on surface cues, lexical likelihoods, or n-gram memorization, thereby improving the attack's ability to evade detection via semantic camouflage.

- **Paraphrasing** (Iyyer et al., 2018): A sentence-level paraphrase attack where each machine-generated text is aggressively rewritten by a strong LLM (DeepSeek V3.1 (Guo et al., 2025)) while preserving its core meaning and language. This aims to alter lexical choice, phrasing, and local syntax so as to mimic human style and evade detectors via semantic camouflage. Our prompt can be found in Table 3.

## B    ADDITIONAL RESULT

### B.1    ABLATION STUDY

To validate the effectiveness of each individual component, we conduct a comprehensive ablation study. Specifically, the ablation compares three training variants that share the *same* backbone (mBERT) and identical data/schedule, differing only in which terms of Eq. (8) are enabled: (i) **mBERT (CE only)** trains on clean examples with standard classification loss $\mathcal{L}_{\text{CE}}(x, y)$; (ii) **mBERT+AT (RPL only, w/o LAAL)** performs adversarial training but *removes* LAAL, and uses only Robust Prediction Loss (RPL), i.e., $\mathcal{L}_{\text{RPL}} = \mathcal{L}_{\text{CE}}(x, y) + \mathcal{L}_{\text{CE}}(\tilde{x}, y)$ where $\tilde{x}$ is the attacker-generated code-switched sample; (iii) **TASTE** is the full method with all terms in Eq. (8), adding LAAL on top of (ii). We do not include a "LAAL-only" (or "w/o RPL") setting because LAAL is not a detection objective by itself and the detector does not converge without a classification loss. We evaluate the zero-shot cross-lingual detection capability and robustness against attacks of our model on the M4 dataset and its extended variant, respectively. The overview results are summarized in Table 4.

Table 4 isolates the effect of LAAL by comparing **mBERT+AT** (adversarial training with RPL only) against the full TASTE (RPL+LAAL). Adding LAAL yields a consistent gain in zero-shot detection accuracy (Avg.: 70.9→76.2, +5.3 points) while further reducing the mean ASR (20.8→18.0, -2.8 points), showing that LAAL improves both cross-lingual generalization and adversarial robustness beyond what is achieved by adversarial training alone. The effect is especially pronounced in harder zero-shot languages such as German (*de*), where accuracy increases from 50.3% to 69.5% while ASR drops from 31.2% to 26.0%.

### B.2    ABLATION ON IMPORTANCE-SCORE DEFINITIONS

To better understand the role of the importance estimator used by the translation-based attacker, we conduct an ablation study over several alternative definitions of token importance. Throughout, we follow the notation: let the surrogate detector be $f_\theta$, the input sequence be $x = (x_1, \ldots, x_T)$, and

let

$$p(y = 1 \mid x) = f_\theta(x)_1$$

denote the predicted probability of the *machine-generated* class. Based on this surrogate, we define three variants of TASTE that differ only in how they compute token importance scores; all other components, including the training setup, are kept identical to the main method.

**(a) Attention-based importance (Attn-TASTE).** For the attention-based variant, we derive token saliency directly from the self-attention maps of the surrogate. Let $A^{(l,h)} \in \mathbb{R}^{T \times T}$ be the attention matrix at layer $l$ and head $h$, where $A_{t,i}^{(l,h)}$ is the attention weight from position $t$ to position $i$. We define the importance of token $x_i$ as the average attention mass it receives across all layers, heads, and source positions:

$$s_i^{\text{attn}} = \frac{1}{LHT} \sum_{l=1}^{L} \sum_{h=1}^{H} \sum_{t=1}^{T} A_{t,i}^{(l,h)}.$$

We then apply min–max normalization over $\{s_i^{\text{attn}}\}_{i=1}^{T}$ to obtain scores in $[0, 1]$ and select the top-$k$ tokens as important positions for cross-lingual replacement.

**(b) Perturbation-based importance (Perturb-TASTE).** For the perturbation-based variant, we measure how sensitive the surrogate's output is to single-token perturbations. Let $x^{(i)}$ be the sequence obtained by replacing token $x_i$ with a neutral placeholder (e.g., EOS or UNK). We define the importance of $x_i$ as the drop in machine probability caused by this local perturbation:

$$s_i^{\text{pert}} = \max\big(0, f_\theta(x)_1 - f_\theta(x^{(i)})_1\big).$$

Intuitively, if perturbing token $x_i$ significantly reduces the predicted probability of the machine class, then $x_i$ is considered more important for the detector's decision and is thus a more promising candidate for adversarial translation. As in the main method, we normalize $\{s_i^{\text{pert}}\}_{i=1}^{T}$ and select the top-$k$ tokens.

| Lang | mBERT | mBERT+AT | TASTE |
|------|-------|----------|-------|
| **Detection Accuracy (%) ↑** | | | |
| en | 80.2 | 96.0 | **97.1** |
| ar | 81.1 | 82.2 | **82.8** |
| bg | 77.4 | 73.2 | **81.7** |
| de | 47.9 | 50.3 | **69.5** |
| id | 83.0 | 80.2 | **84.0** |
| it | 68.8 | 63.4 | **69.4** |
| ru | 41.8 | 55.3 | **61.7** |
| ur | 67.1 | 67.6 | **69.7** |
| zh | 68.4 | 70.0 | **70.1** |
| Avg. ↑ | 68.4 | 70.9 | **76.2** |
| **ASR (%) under Attacks ↓** | | | |
| en | 30.0 | 16.1 | **14.5** |
| ar | 30.7 | **9.8** | 10.0 |
| bg | 41.1 | 22.4 | **22.0** |
| de | 48.1 | 31.2 | **26.0** |
| id | 40.8 | 18.7 | **17.6** |
| it | 34.7 | 21.3 | **18.9** |
| ru | 46.4 | 26.6 | **24.5** |
| ur | 28.5 | 24.8 | **12.0** |
| zh | 18.9 | 16.6 | **16.3** |
| Avg. ↓ | 35.5 | 20.8 | **18.0** |

Table 4: **Ablation on Eq. (8) components on M4GT. mBERT**: CE-only on clean $x$. **mBERT+AT**: adversarial training with RPL only (CE on $x$ and $\tilde{x}$), *without* LAAL. **TASTE**: full model (RPL + LAAL). Best in each row is **bolded**.

**(c) Random importance baseline (Random-TASTE).** As an additional sanity check, we include a purely random baseline where the attacker selects important tokens uniformly at random, independent of the surrogate. Concretely, for a sequence of length $T$ we uniformly sample a subset $\mathcal{I} \subset \{1, \ldots, T\}$ with $|\mathcal{I}| = k$, and assign random scores $s_i^{\text{rand}} \sim \mathcal{U}(0, 1)$ for $i \in \mathcal{I}$. This variant isolates the effect of *where* we perturb from any modeling choice in the importance estimator.

We train Attn-TASTE, Perturb-TASTE, Random-TASTE, and the original gradient-based TASTE under the same setup, and report test accuracy on 9 languages. The accuracies and their cross-lingual means are summarized in Table 5.

**Clarification.** Table 5 reports the *final detector*'s clean-test accuracy after training with adversarial samples generated by each importance estimator. It does *not* measure the surrogate/attacker accuracy, nor the intrinsic quality of an importance estimator. In particular, Random-TASTE samples token positions uniformly at random, which is equivalent to an *unguided (no-importance) code-switch augmentation* baseline.

From Table 5 we observe that all variants achieve strong multilingual performance, indicating that TASTE is robust to the choice of importance estimator. The original gradient-based TASTE achieves the highest cross-lingual mean accuracy (0.762), slightly outperforming Attn-TASTE, Perturb-TASTE, and Random-TASTE. Attention-based and perturbation-based scores behave similarly to

| Method | en | ar | zh | de | ru | bg | id | it | ur | Mean |
|---|---|---|---|---|---|---|---|---|---|---|
| Attn-TASTE | 0.974 | 0.779 | 0.746 | 0.511 | 0.557 | 0.830 | 0.869 | 0.621 | 0.879 | 0.752 |
| Perturb-TASTE | 0.961 | 0.815 | 0.680 | 0.589 | 0.517 | 0.807 | 0.867 | 0.647 | 0.920 | 0.756 |
| Random-TASTE | 0.970 | 0.777 | 0.679 | 0.587 | 0.562 | 0.820 | 0.904 | 0.625 | 0.882 | 0.756 |
| TASTE | 0.971 | 0.828 | 0.701 | 0.695 | 0.617 | 0.817 | 0.840 | 0.694 | 0.697 | 0.762 |

Table 5: **Detector clean-test accuracy (Acc) after adversarial training with samples generated by different importance estimators** across 9 languages.

| Attack | Attn-TASTE | Perturb-TASTE | Random-TASTE | TASTE |
|---|---|---|---|---|
| Insert | 69.5 | 49.0 | 57.6 | **38.8** |
| Back-Translation | 19.7 | 12.9 | **12.8** | 16.3 |
| Code-Switching | 74.4 | 71.2 | 65.6 | **19.7** |
| Delete | 33.0 | 23.3 | 28.3 | **14.6** |
| Human Obfuscation | 8.4 | **5.3** | 8.4 | 5.4 |
| Paraphrasing | 28.1 | **4.2** | **4.2** | 10.5 |
| Repeat | 70.2 | 53.8 | 61.8 | **38.4** |
| Swap | 1.0 | 1.0 | 1.0 | **0.2** |
| Mean ASR ↓ | 38.0 | 27.6 | 30.0 | **18.0** |

Table 6: **Mean attack success rate (ASR, %) of the final detectors trained with adversarial samples generated by different importance estimators** (lower is better). The best (lowest) ASR in each row is highlighted in bold.

the gradient-based scheme on many languages, but do not provide systematic gains. In particular, Attn-TASTE underperforms on some low-resource languages (e.g., German and Russian), which suggests that attention weights may be less reliable as fine-grained importance signals for adversarial code-switching.

To further understand the effect of different importance metrics on the *strength* of adversarial training, we also compare the attack success rate (ASR) of the final detectors under a suite of multilingual attacks. Table 6 reports the mean ASR (averaged over languages) for each attack and each importance estimator, expressed as percentages (lower is better).

With the full set of attacks (including back-translation and paraphrasing), the gradient-based TASTE attains the lowest mean ASR (18.0%), substantially lower than Attn-TASTE (38.0%), Random-TASTE (30.0%), and Perturb-TASTE (27.6%), indicating the strongest overall robustness. For six out of eight attacks (Insert, Code-Switching, Delete, Repeat, Swap, and almost tied on Human Obfuscation), TASTE achieves the best or near-best ASR, showing that gradient-based importance is particularly effective across a broad range of perturbation patterns. Perturb-TASTE and Random-TASTE obtain slightly lower ASR than TASTE on a few specific attacks (e.g., Back-Translation and Paraphrasing, where they reach 12.9–12.8% and 4.2%, respectively), but these localized gains do not translate into a better *average* robustness profile: both methods still have higher overall mean ASR than TASTE. Attn-TASTE consistently exhibits the highest ASR across attacks, confirming that attention weights alone are a weaker signal for constructing strong training-time adversaries in this multilingual setting.

We note that Back-Translation and Paraphrasing are phrase-/semantic-level transformations that are not perfectly aligned with our token-level code-switch perturbation family; thus unguided randomness can occasionally act as a stronger regularizer on these two attacks, but it does not translate into better overall robustness (mean ASR).

Taken together with Table 5, these results show that gradient-based importance scores provide the best trade-off: they preserve high average accuracy across 9 languages while also yielding the lowest overall attack success rate among all variants. This empirically supports the choice of gradient-based

| Surrogate model | en | ar | zh | de | ru | bg | id | it | ur | Mean |
|---|---|---|---|---|---|---|---|---|---|---|
| GPT-2 (124M) | 0.971 | 0.828 | 0.701 | 0.695 | 0.617 | 0.817 | 0.840 | 0.694 | 0.697 | 0.762 |
| ALBERT-base-v2 (12M) | 0.965 | 0.788 | 0.632 | 0.647 | 0.525 | 0.779 | 0.834 | 0.684 | 0.780 | 0.737 |
| Pythia-70M (70M) | 0.972 | 0.826 | 0.707 | 0.593 | 0.553 | 0.808 | 0.876 | 0.596 | 0.847 | 0.753 |

Table 8: Accuracy of TASTE under different surrogate models.

importance as the default design in TASTE, even when considering a richer set of attacks including back-translation and paraphrasing.

### B.3 IMPACT OF SURROGATE MODEL

Due to the black-box assumption from the attacker's perspective, we do not know the architecture or the training dataset of the target detector. Therefore, we adopt a GPT2-based model (Radford et al., 2019) trained on the dataset from Wang et al. (2024a) as the surrogate model in the main text. In standard adversarial training approaches under strict black-box constraints (Li et al., 2020b; Yoo & Qi, 2021; Li et al., 2025), the choice of surrogate model can significantly influence the target detector's performance. Specifically, using a surrogate model with a similar architecture and dataset often results in more precise identification of important tokens, thereby generating stronger adversarial samples that can better mislead the target detector.

To eliminate the potential confounding effects of surrogate model and training dataset choices on experimental results, we conduct experiments using the same mBERT backbone and the same training dataset as the target detector as the surrogate. Additionally, to provide a fine-grained analysis of the impact of detector architecture and dataset, we also compare configurations using GPT2 + M4GT dataset (same dataset for training TASTE, named as IND) and mBERT + Wang et al. (2024a) dataset (different dataset for training TASTE, named as OOD). The detailed results are presented in Table 7.

As shown in Table 7, even when replacing the surrogate model with a detector's backbone model or using the same training dataset, the performance of the target detector does not exhibit significant improvement. This is because, during each step of adversarial training, we utilize the target detector's predicted labels to continuously update the surrogate model, ultimately enabling the surrogate to approximate the target detector's predictions closely. This further confirms the applicability and robust-

| Surrogate Model | Dataset (Type) | Avg Acc ↑ | Avg ASR (%) ↓ |
|---|---|---|---|
| GPT2 | Wang et al. (2024a) (OOD) | 76.2 | 18.0 |
| GPT2 | M4GT (IND) | 76.1 | 18.3 |
| mBERT | Wang et al. (2024a) (OOD) | 75.3 | 18.9 |
| mBERT | M4GT (IND) | 76.4 | 17.7 |

Table 7: The results of different surrogate models and datasets that the surrogate models are trained with.

ness of our adversarial training framework, suggesting that regardless of the surrogate model architecture or the choice of training dataset, the final performance of the target detector remains relatively stable without substantial variations.

We further investigate the impact of surrogate model size on the experimental results. Our framework only requires that the surrogate model provide informative gradients to guide the translation-based attacker; in principle, it does not depend on a specific architecture or parameter scale. To clarify the impact of surrogate size, we experiment with two smaller surrogates, `albert/albert-base-v2` (12M parameters) (Lan et al., 2019) and `EleutherAI/pythia-70m` (70M parameters) (Biderman et al., 2023), and compare them with the original GPT-2 (124M) used in the main experiments. We report the results in Table 8.

From Table 8, several trends can be observed. First, using much smaller surrogates still yields strong multilingual performance: both ALBERT-base-v2 (12M) and Pythia-70M (70M) remain competitive with the original GPT-2 (124M), with cross-lingual mean accuracies of 0.737 and 0.753, respectively, compared to 0.762 for GPT-2. Second, the Pythia-70M surrogate is particularly close to GPT-2: it even slightly outperforms GPT-2 on several languages (e.g., en: 0.972 vs. 0.971, id: 0.876 vs. 0.840, zh: 0.707 vs. 0.701), while being weaker on others (e.g., de, ru). This suggests that TASTE

| Model | en | ar | zh | de | ru | bg | id | it | ur | Mean |
|---|---|---|---|---|---|---|---|---|---|---|
| TASTE (0% noise) | 0.971 | 0.828 | 0.701 | 0.695 | 0.617 | 0.817 | 0.840 | 0.694 | 0.697 | 0.762 |
| TASTE (10% noise) | 0.956 | 0.769 | 0.687 | 0.706 | 0.563 | 0.812 | 0.830 | 0.666 | 0.918 | 0.767 |
| TASTE (20% noise) | 0.940 | 0.826 | 0.630 | 0.756 | 0.512 | 0.742 | 0.658 | 0.763 | 0.833 | 0.740 |

Table 9: **Accuracy of TASTE under different levels of pseudo-label noise during training.** Higher accuracy means higher performance.

mainly relies on a reasonable gradient signal from the surrogate, rather than on the full capacity of a 124M-parameter model.

Third, the ALBERT-base-v2 surrogate, despite being an order of magnitude smaller (12M), still delivers a mean accuracy of 0.737, indicating that TASTE is not tightly coupled to a specific large surrogate and can be instantiated with very lightweight models when computational resources are limited. Overall, these results demonstrate that the performance of TASTE is robust to the choice and size of the surrogate model: smaller surrogates can be used with only moderate degradation, and in some languages Pythia-70M even matches or slightly exceeds the GPT-2–based variant, showing that the framework can flexibly adapt to different computational budgets.

### B.4 IMPACT OF LABEL NOISE ON THE TRAINING PROCESS

In Section 4.1, we mention that the attacker (surrogate model) is trained on pseudo-labels produced by the detector. In the real world setting, the pseudo-label from the detector may contains noise. If these pseudo-labels are flipped, the attacker's gradients are corrupted, which may in turn hurt both accuracy and robustness of the final detector. To directly quantify this effect, we conduct an experiment where, during training, we randomly flip a fixed proportion of the pseudo-labels used for the surrogate (10% and 20%), and then evaluate the final detector under the same setting as in the main results.

We first report the accuracy across 9 languages for three settings: no label noise (original TASTE), 10% noise, and 20% noise. The results are given in Table 9.

**Observations from Table 9.**

- Overall, moderate label noise does not catastrophically degrade the detector:
  - The mean accuracy with 10% noise (76.7%) is essentially on par with the clean setting (76.2%), and even slightly higher, which can be attributed to a mild regularization effect.
  - With 20% noise, the mean accuracy drops to 74.0%, i.e., a decrease of only about 2.2 points compared to the clean TASTE.
- Per-language trends show some expected degradation (e.g., zh, ru, id), but even at 20% noise, the accuracies remain at a reasonable level and do not collapse, indicating that the co-training dynamics between attacker and detector are fairly robust to pseudo-label noise.

Next, we examine how label noise affects robustness, measured via attack success rate (ASR) under the same multilingual attack suite as in Section 5.2. Here we focus on TASTE only, and report mean ASR (averaged over languages) for 8 attacks under the three noise levels. The results are shown in Table 10.

**Observations from Table 10.**

- As expected, injecting label noise into the surrogate makes the final detector more vulnerable:
  - The mean ASR increases from 18.0% (clean) to 27.4% (10% noise) and 22.2% (20% noise).
  - The degradation is most pronounced for code-switching and surface-level perturbations such as Insert and Repeat, where noisy pseudo-labels directly disturb the gradients used to select important tokens.

| Attack | TASTE (0% noise) | TASTE (10% noise) | TASTE (20% noise) |
|---|---|---|---|
| Insert | 38.8 | 51.9 | 48.6 |
| Back-Translation | 16.3 | 13.9 | 10.2 |
| Code-Switching | 19.7 | 63.8 | 40.8 |
| Delete | 14.6 | 24.3 | 19.9 |
| Human Obfuscation | 5.4 | 4.5 | 2.5 |
| Paraphrasing | 10.5 | 6.2 | 6.2 |
| Repeat | 38.4 | 53.0 | 49.0 |
| Swap | 0.2 | 1.5 | 0.3 |
| Mean ASR ↓ | 18.0 | 27.4 | 22.2 |

Table 10: **Mean attack success rate (ASR, %) of TASTE under different levels of pseudo-label noise (lower is better).**

- For semantic-preserving attacks such as Back-Translation and Paraphrasing, the ASR does not consistently worsen with noise; in some cases, it even slightly improves (e.g., Back-Translation drops from 16.3% to 10.2% at 20% noise). A plausible explanation is that moderate noise acts as an additional regularizer that discourages the detector from overfitting to specific surface cues exploited by these attacks.

- Overall, even with 20% label noise, the mean ASR remains at 22.2%, which is still reasonably low compared to standard detectors without adversarial training, confirming that the attacker–detector co-evolution is robust but not immune to pseudo-label corruption.

Taken together, these results show that while pseudo-label noise does degrade robustness—as one would expect—the degradation is graceful rather than catastrophic. TASTE maintains strong multilingual accuracy and relatively low ASR even when up to 20% of the surrogate's training labels are flipped, suggesting that the gradient-based attacker remains reliable in practice under realistic levels of detector prediction noise.

## B.5 Out-Of-Distribution (OOD) Experiment

| Lang | Metric | Metric-based Detectors | | | Model-based Detectors | | | | | |
|---|---|---|---|---|---|---|---|---|---|---|
| | | Fast DetectGPT | Binoculars | LRR | XLM-Roberta-Base | mDeBERT-v3-Base | mBERT | RADAR* | GREATER-D * | TASTE (Ours)* |
| en | Acc ↑ | 0.742 | 0.663 | 0.369 | 0.902 | 0.906 | 0.869 | 0.903 | 0.914 | **0.935** |
| | F1 ↑ | 0.754 | 0.633 | 0.538 | 0.899 | 0.904 | 0.866 | 0.894 | 0.906 | **0.934** |
| ru | Acc ↑ | **0.659** | 0.518 | 0.489 | 0.499 | 0.431 | 0.461 | 0.478 | 0.566 | 0.632 |
| | F1 ↑ | 0.587 | 0.385 | 0.087 | 0.385 | 0.338 | 0.457 | 0.433 | 0.526 | **0.601** |
| zh | Acc ↑ | **0.876** | 0.734 | 0.360 | 0.838 | 0.785 | 0.652 | 0.614 | 0.703 | 0.753 |
| | F1 ↑ | **0.872** | 0.714 | 0.526 | 0.836 | 0.783 | 0.625 | 0.518 | 0.606 | 0.743 |
| bg | Acc ↑ | 0.691 | 0.670 | 0.444 | 0.521 | 0.354 | 0.768 | 0.766 | 0.588 | **0.793** |
| | F1 ↑ | 0.635 | 0.630 | 0.043 | 0.478 | 0.330 | 0.767 | 0.764 | 0.504 | **0.787** |
| ur | Acc ↑ | 0.839 | **0.973** | 0.486 | 0.600 | 0.566 | 0.703 | 0.679 | 0.587 | 0.728 |
| | F1 ↑ | 0.837 | **0.972** | 0.017 | 0.598 | 0.475 | 0.681 | 0.601 | 0.508 | 0.710 |
| Avg. | Acc ↑ | 0.761 | 0.711 | 0.429 | 0.672 | 0.608 | 0.691 | 0.688 | 0.672 | **0.768** |
| | F1 ↑ | 0.737 | 0.667 | 0.242 | 0.639 | 0.566 | 0.679 | 0.642 | 0.610 | **0.755** |

Table 11: **Performance on SemEval Task8 (OOD) Dataset.** We report accuracy (*Acc*) and F1 score (*F1*). The best result in each row is in **bold**, the second-best is underlined. * indicates adversarial-training-based methods.

To evaluate the model's generalization ability on unseen datasets, we assessed detectors' performance on the SemEval Task 8 dataset Wang et al. (2024c). The result is shown in Table 11. Overall, our TASTE attains the best macro performance across languages, yielding the highest averaged *Acc* (0.768) and *F1* (0.755), with Fast DetectGPT the second best (*Acc* 0.761, *F1* 0.737). At the per-language level, TASTE leads on *en* (*Acc* 0.935/*F1* 0.934) with +0.021/+0.028 over the next best (GREATER-D), and on *bg* (*Acc* 0.793/*F1* 0.787) with clear margins over the strongest baselines (mBERT: 0.768/0.767). In *ru*, Fast DetectGPT tops *Acc* (0.659) while TASTE offers the best *F1*

| Annotator | Readability ↑ | | Coherence ↑ | | Overall quality ↑ | |
|---|---|---|---|---|---|---|
| | O | A | O | A | O | A |
| A1 | $4.29 \pm 0.79$ | $3.84 \pm 1.07$ | $4.59 \pm 0.60$ | $3.79 \pm 1.30$ | $4.24 \pm 0.43$ | $4.06 \pm 0.75$ |
| A2 | $4.03 \pm 0.96$ | $3.80 \pm 1.18$ | $4.08 \pm 0.64$ | $3.84 \pm 0.69$ | $3.98 \pm 0.95$ | $3.48 \pm 1.08$ |
| A3 | $4.57 \pm 0.64$ | $4.30 \pm 0.70$ | $4.39 \pm 0.55$ | $4.14 \pm 0.65$ | $4.61 \pm 0.56$ | $4.22 \pm 0.63$ |
| Mean | 4.30 | 3.98 | 4.35 | 3.92 | 4.28 | 3.92 |

Table 12: **Human evaluation scores (Original Example (O) vs Adversarial Example (A)) for readability, coherence, and overall quality.** Since the scores are bounded in [1, 5], the mean ± std interval should not be interpreted as the actual attainable score range, but only as an indicator of the dispersion of the annotations.

(0.601), indicating more balanced precision–recall under distribution shift. Two challenging regimes emerge: (*i*) *ur*, where the metric-based Binoculars excels (*Acc/F1* 0.973/0.972), likely benefiting from its generator-observer divergence heuristic on this domain; and (*ii*) *zh*, where Fast DetectGPT dominates (*Acc/F1* 0.876/0.872) while TASTE trails (0.753/0.743). Classical linear metrics (LRR) are consistently weak (e.g., *bg*: 0.444/0.043; *ur*: 0.486/0.017), underscoring the limitation of shallow cues under OOD shift. Taken together, the results suggest that adversarially trained, model-based detectors deliver the most reliable OOD robustness on average, but language-specific gaps remain (notably *ur* and *zh*), motivating future work on adaptive calibration and script-/morphology-aware multilingual representations.

### B.6   QUALITY EVALUATION OF ADVERSARIAL EXAMPLE

To evaluation the quality of adversarial example generated by TASTE's attacker, we recruited three human annotators (A1–A3), who independently evaluated 100 pairs of original examples and adversarial examples generated by the TASTE attacker. Each pair was rated along three dimensions: readability, coherence, and overall quality. Scores range from 1 to 5. The result can be found in Table 12.

From this table, we observe that adversarial examples remain highly readable and coherent: all mean scores are close to or above 4.0 on a 1–5 scale, indicating that annotators still perceive them as natural and easy to understand. At the same time, there is a consistent but moderate degradation when moving from original to adversarial texts: the average drops are about 0.32 for readability (4.30 → 3.98), 0.43 for coherence (4.35 → 3.92), and 0.36 for overall quality (4.28 → 3.92). This pattern suggests that TASTE 's perturbations slightly reduce perceived quality-as expected for adversarial modifications-yet do not introduce severe grammatical errors or destroy semantic coherence. We also note that some annotators (e.g., A2 on overall quality) are more sensitive to the artifacts introduced by the attack and give lower scores, while others show milder drops, which is consistent with typical inter-annotator variability in human evaluation.

**Annotation protocol.**   All three annotators were senior undergraduate students in engineering majors at our university, each with substantial multilingual experience. Before the main study, we provided a written guideline that defined the three dimensions (readability, coherence, overall quality) and described the 1–5 scale with concrete examples for "poor" (1), "acceptable" (3), and "excellent" (5) texts. The annotators then went through a short pilot phase (20 practice pairs not used in the final analysis), where they independently rated the examples and received feedback to calibrate their use of the scale.

During the main annotation phase, the 100 pairs were randomly shuffled, and for each pair the order of the original and adversarial text was randomized. The interface only showed two anonymized texts (Text A and Text B) without revealing which one was adversarial or any model-related information, and annotators were explicitly instructed not to guess the source or authenticity of the texts, but to focus solely on perceived linguistic quality. For each text, they were asked to assign three separate integer scores in $\{1, \ldots, 5\}$ for readability, coherence, and overall quality, interpreting them as a standard Likert scale. No examples could be skipped, but annotators were allowed to take short breaks and complete the task in a single session of less than one hour.

| Detector | en | de | zh | bg | id | Mean |
|---|---|---|---|---|---|---|
| Binoculars | 0.2903 | 0.3645 | 0.3031 | 0.2027 | 0.5638 | 0.3449 |
| Fast DetectGPT | 0.5327 | **0.3861** | **0.6152** | 0.2027 | **0.7001** | 0.4874 |
| LRR | 0.0432 | 0.1042 | 0.1023 | 0.0641 | 0.0000 | 0.0628 |
| mBERT | 0.3740 | 0.0531 | 0.3277 | 0.2058 | 0.3195 | 0.2560 |
| mDeBERT-v3-Base | 0.6300 | 0.2246 | 0.3492 | 0.1595 | 0.5289 | 0.3780 |
| XLM-RoBERTa-Base | 0.1914 | 0.0757 | 0.2834 | 0.1060 | 0.0992 | 0.1512 |
| RADAR | 0.6372 | 0.0644 | 0.1346 | 0.1475 | 0.2662 | 0.2500 |
| GREATER-D | 0.7442 | 0.0886 | 0.2306 | 0.1844 | 0.2745 | 0.3045 |
| TASTE | **0.8716** | 0.2856 | 0.3469 | **0.5206** | 0.6663 | **0.5382** |

Table 13: **TPR@FPR=1% of each detector in M4 Dataset.** We highlight the best result in each group in **bold** and the second-best result with underline.

Annotators were compensated at an hourly rate comparable to standard student research assistantships at our institution. The study did not collect any personally identifiable information beyond basic demographic and educational background.

## B.7 COMPLEMENTARY EVALUATION UNDER TPR@FPR=1%

To complement the accuracy and F1 metrics reported in the main paper, we further follow common practice in security-sensitive detection and evaluate all detectors under the operating point TPR@FPR=1% on the M4 dataset. We consider five languages: one training language (English), two dictionary languages (Chinese and German), and two fully zero-shot languages (Indonesian and Bulgarian). The results are summarized in Table 13.

From Table 13, we observe that under this stringent operating point, TASTE achieves the best average performance across the five languages (mean TPR of 0.5382), surpassing strong zero-shot baselines such as Fast DetectGPT (0.4874) and Binoculars (0.3449). On English (the training language), TASTE attains a TPR of 0.8716 at 1% FPR, substantially improving over both GREATER-D (0.7442) and RADAR (0.6372), which are specifically designed for machine-generated text detection.

For the two zero-shot languages (Bulgarian and Indonesian), TASTE remains highly competitive. On Bulgarian, it achieves the highest TPR at 1% FPR (0.5206 vs. 0.2027 for Fast DetectGPT), and on Indonesian it reaches the second-best result (0.6663 vs. 0.7001 for Fast DetectGPT), indicating strong out-of-distribution robustness despite the absence of language-specific supervision. On the two dictionary languages (Chinese and German), Fast DetectGPT attains the highest scores, while TASTE is slightly lower but still competitive. Overall, TASTE maintains the best cross-language mean under TPR@FPR=1%, showing that the performance trends observed with Acc/F1 in the main paper remain consistent when evaluated with a more security-relevant metric.

We further examine the out-of-distribution robustness of all detectors on the SemEval 2024 Task 8 dataset, which is disjoint from M4. To ensure a fair comparison, we reuse the same thresholds as in Table 13, i.e., each detector's threshold is calibrated on M4 to achieve TPR@FPR=1%, and then directly applied to SemEval without any additional tuning. We focus on the overlapping languages Bulgarian/English/Chinese (bg/en/zh) and report both TPR and FPR, as well as their means across these three languages; the results are shown in Table 14.

As shown in Table 14, TASTE achieves the highest mean TPR (0.5700) on the OOD SemEval dataset across bg/en/zh, outperforming strong zero-shot detectors such as Binoculars (0.4952) and Fast DetectGPT (0.4589), as well as other mBERT-based baselines. In particular, on Bulgarian, TASTE substantially improves recall (0.5633) compared to all competitors, and on English it attains the second-highest TPR (0.7697), close to mDeBERT-v3-Base (0.7998), while remaining competitive on Chinese.

Crucially, these gains in TPR do not come at the cost of excessively high FPR on OOD data. The average FPR of TASTE (1.96%) remains low, noticeably below that of other mBERT-based detec-

| Detector | bg | | en | | zh | | Mean | |
|---|---|---|---|---|---|---|---|---|
| | TPR | FPR | TPR | FPR | TPR | FPR | TPR | FPR |
| Binoculars | 0.1958 | 0.0073 | 0.6122 | 0.0076 | 0.6777 | 0.0097 | 0.4952 | 0.0082 |
| Fast DetectGPT | 0.2811 | 0.0052 | 0.5675 | 0.0048 | 0.5281 | 0.0031 | 0.4589 | 0.0044 |
| LRR | 0.0000 | 0.0000 | 0.0010 | 0.0000 | 0.0030 | 0.0000 | 0.0013 | 0.0000 |
| mBERT | 0.2339 | 0.0219 | 0.5345 | 0.0412 | 0.4383 | 0.0369 | 0.4022 | 0.0333 |
| mDeBERT-v3-Base | 0.1586 | 0.0110 | 0.7998 | 0.0165 | 0.4895 | 0.0259 | 0.4826 | 0.0178 |
| XLM-RoBERTa-Base | 0.1195 | 0.0269 | 0.2488 | 0.0412 | 0.3882 | 0.0279 | 0.2522 | 0.0320 |
| RADAR | 0.1305 | 0.0345 | 0.5561 | 0.0518 | 0.2662 | 0.0313 | 0.3176 | 0.0392 |
| GREATER-D | 0.2014 | 0.0214 | 0.6362 | 0.0336 | 0.2745 | 0.0287 | 0.3707 | 0.0279 |
| TASTE | 0.5633 | 0.0179 | 0.7697 | 0.0319 | 0.3771 | 0.0090 | 0.5700 | 0.0196 |

Table 14: **TPR and FPR of each detector on the SemEval 2024 Task 8 (OOD) dataset at the thresholds used in Table 13.**

| Detector | Inference time (s) | Time / example (ms) |
|---|---|---|
| Fast DetectGPT | 2862.47 | 194.40 |
| Binoculars | 3173.01 | 215.48 |
| LRR | 1361.85 | 92.49 |
| mBERT | 76.41 | 5.19 |
| mDeBERT-v3-Base | 133.97 | 9.10 |
| XLM-RoBERTa-Base | 82.80 | 5.62 |
| RADAR | 77.66 | 5.27 |
| GREATER-D | 75.90 | 5.15 |
| TASTE | 76.09 | 5.17 |

Table 15: End-to-end inference time on the M4 test set for each detector. "Inference time" is the wall-clock time for one full pass over all 14,725 test examples on a single RTX 4090 GPU under identical batching settings; "Time / example" is the corresponding average latency per input.

tors such as RADAR (3.92%) and mBERT/XLM-RoBERTa, and comparable to mDeBERT-v3-Base (1.78%). Although Fast DetectGPT achieves a lower average FPR (0.44%), this comes with a considerably lower mean TPR (0.4589 vs. 0.5700). Taken together, these results indicate that TASTE achieves a more favorable recall–FPR trade-off at a security-relevant operating point, and that its advantages on M4 largely transfer to a challenging OOD benchmark.

## B.8 ADDITIONAL EVALUATION ON EFFICIENCY

In addition to detection performance, we also evaluate the computational efficiency of all detectors. As reported in Section 6.1, the training time of our method is about 40 minutes for 3 epochs on a single RTX 4090 GPU, which is a reasonably low training cost for a multilingual detector. To better understand the deployment-time overhead, we further measure the end-to-end inference latency of each detector on the M4 test set.

The test set contains 14,725 examples in total. For each detector, we measure the wall-clock time (in seconds) required to complete a *single full pass* over all 14,725 examples on a single RTX 4090 GPU under identical batching and implementation settings, and then derive the average time per example in milliseconds. The results are summarized in Table 15.

The results reveal a clear efficiency gap between metric-based and model-based detectors. Metric-based approaches such as Fast DetectGPT, Binoculars, and LRR incur very large inference latency (about 92–215 ms per example), primarily due to the cost of repeated perturbation, scoring, or

generative calls. In contrast, model-based detectors (mBERT, mDeBERT-v3-Base, XLM-RoBERTa-Base, RADAR, GREATER-D, and TASTE) operate in the single-digit millisecond regime (roughly 5–9 ms per example). In particular, GREATER-D and TASTE are among the fastest, with average latencies of 5.15 ms and 5.17 ms per example, respectively.

Viewed in terms of throughput, TASTE is more than an order of magnitude faster than metric-based detectors at inference time: compared to Fast DetectGPT and Binoculars, it achieves speedups of approximately $38\times$ and $42\times$, respectively, and is about $18\times$ faster than LRR on a per-example basis. These findings indicate that, despite requiring a one-time training cost, TASTE and other model-based detectors are substantially more attractive for real-world, large-scale deployment scenarios, where inference latency is a critical constraint and can partially offset the upfront training overhead.

### B.9 ANALYSIS OF TRAINING PROCESS

To assess the stability of the joint training procedure, we monitor the losses of both the detector and the attacker throughout training. Figure 6 plots the representation-preserving loss $\mathcal{L}_{\mathrm{RPL}}$, the language-agnostic adversarial loss $\mathcal{L}_{\mathrm{LAAL}}$, and the attacker loss $\mathcal{L}_{\mathrm{att}}$ as a function of training steps.

At the beginning of training, all three losses exhibit mild oscillations, reflecting the initial adaptation of the surrogate and detector. After roughly 200 steps, the curves become much smoother: $\mathcal{L}_{\mathrm{RPL}}$ continues to decrease steadily, $\mathcal{L}_{\mathrm{LAAL}}$ decreases slightly and then stabilizes around a plateau, and $\mathcal{L}_{\mathrm{att}}$ drops quickly before converging to a low value. Overall, the losses of both components remain well-behaved over the entire training horizon, indicating that the attacker–detector co-training process is stable and does not suffer from divergence or collapse.

## C FORMAL GUARANTEE OF CROSS-LINGUAL GENERALIZATION BY $\mathcal{L}_{\mathrm{LAAL}}$

**Setting and Notation.** Let $\phi$ be the encoder of the target detector $\mathcal{D}_{\mathrm{tar}}(\cdot)$, and $h = \phi(x)$ the sentence-level representation (e.g., [CLS]). For each language/domain $k \in \{1, \dots, K\}$, let $P_k$ be the input distribution and $P_k^\phi$ the pushforward feature distribution under $\phi$. Let $f_{\mathrm{lang}}$ be the language discriminator in Eq. (7), trained adversarially via gradient reversal as described below Eq. (7). Let $y(x) \in \{0, 1\}$ be the ground-truth class (HWT/MGT). The attacker generates $\tilde{x}$ via code-switching (Section 4.2; Fig. 2), and the Robustness-Preserving Loss (RPL) Eq. (6) enforces $y(\tilde{x}) = y(x)$ at training time.

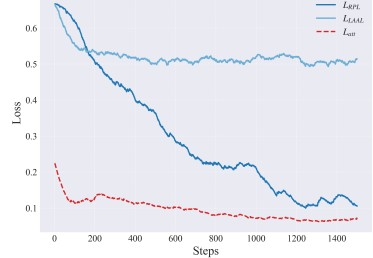

Figure 6: Exponentially weighted moving average (EWMA) of the training losses for the detector ($\mathcal{L}_{\mathrm{RPL}}, \mathcal{L}_{\mathrm{LAAL}}$) and the attacker ($\mathcal{L}_{\mathrm{att}}$) over optimization steps.

**Assumption (Semantics-preserving code-switching).** For any input $x$ and its code-switched $\tilde{x}$ produced by the attacker,

$$y(\tilde{x}) \;=\; y(x), \tag{10}$$

and $\tilde{x}$ differs from $x$ only by lexical-level cross-lingual substitutions that preserve sentence meaning.

**LAAL as a domain-adversarial objective.** Let $\pi = (\pi_1, \dots, \pi_K)$ be the prior over languages/domains used by $f_{\mathrm{lang}}$ in Eq. (7). Given a feature $h$, the *optimal* discriminator satisfies

$$f_{\mathrm{lang}}^\star(k \mid h) \;=\; \frac{\pi_k\, p_k^\phi(h)}{\sum_{j=1}^K \pi_j\, p_j^\phi(h)}, \tag{11}$$

where $p_k^\phi$ is the density of $P_k^\phi$.

**Lemma 1** (LAAL minimizes generalized JS among languages). *Let $\mathcal{L}_{\mathrm{LAAL}}$ be Eq. (7). Define the generalized Jensen–Shannon divergence $\mathrm{GJS}_\pi(P_1^\phi, \dots, P_K^\phi) = \sum_{k=1}^K \pi_k\, \mathrm{KL}\!\left(P_k^\phi \,\big\|\, \sum_{j=1}^K \pi_j P_j^\phi\right)$. Then the minimum LAAL loss over discriminators equals*

$$\min_{f_{\mathrm{lang}}} \mathcal{L}_{\mathrm{LAAL}}(\phi, f_{\mathrm{lang}}) \;=\; H(\pi) \;-\; \mathrm{GJS}_\pi(P_1^\phi, \dots, P_K^\phi), \tag{12}$$

*where $H(\pi)$ is the entropy of $\pi$. Consequently, maximizing Eq. (7) w.r.t. $\phi$ via gradient reversal is equivalent to*

$$\phi^\star \ \in \ \arg\min_{\phi} \ \mathrm{GJS}_\pi\big(P_1^\phi,\ldots,P_K^\phi\big), \tag{13}$$

*i.e., LAAL drives the language/domain feature distributions toward each other.*

*Proof.* By Eq. (11), substituting $f_{\mathrm{lang}}^\star$ into Eq. (7) gives

$$\begin{aligned}
\min_{f_{\mathrm{lang}}} \mathcal{L}_{\mathrm{LAAL}} &= \ -\sum_{k=1}^{K} \pi_k \, \mathbb{E}_{h \sim P_k^\phi} \log \frac{\pi_k p_k^\phi(h)}{\sum_{j=1}^{K} \pi_j p_j^\phi(h)} \\
&= \ -\sum_{k=1}^{K} \pi_k \log \pi_k \ + \ \sum_{k=1}^{K} \pi_k \, \mathrm{KL}\left( P_k^\phi \,\Big\|\, \sum_{j=1}^{K} \pi_j P_j^\phi \right) \\
&= \ H(\pi) \ - \ \mathrm{GJS}_\pi(P_1^\phi,\ldots,P_K^\phi), \tag{14}
\end{aligned}$$

which yields Eq. (12). Since the encoder $\phi$ receives the *reversed* gradient $-\lambda_{\mathrm{lang}} \, \partial \mathcal{L}_{\mathrm{LAAL}}/\partial h$ (see the paragraph below Eq. (7)), it maximizes the minimum in Eq. (12), equivalently minimizing $\mathrm{GJS}_\pi$ as in Eq. (13). $\qquad\square$

**From language invariance to domain discrepancy.** Let $d_{\mathcal{H}\Delta\mathcal{H}}(P,Q)$ be the standard discrepancy distance between two distributions for a hypothesis class $\mathcal{H}$. For two languages/domains $k$ and $k'$, define the mixed class-conditional feature distributions

$$\bar{P}_k^\phi \ = \ \sum_{c \in \{0,1\}} \pi_{c|k} \, P_{k,c}^\phi, \qquad \bar{P}_{k'}^\phi \ = \ \sum_{c \in \{0,1\}} \pi_{c|k'} \, P_{k',c}^\phi, \tag{15}$$

where $\pi_{c|k}$ is the class prior within language $k$ and $P_{k,c}^\phi$ the class-conditional feature distribution. By standard inequalities between $f$-divergences and total variation, there exists a universal constant $C > 0$ such that

$$d_{\mathcal{H}\Delta\mathcal{H}}\big(\bar{P}_k^\phi, \bar{P}_{k'}^\phi\big) \ \leq \ 2\,\mathrm{TV}\big(\bar{P}_k^\phi, \bar{P}_{k'}^\phi\big) \ \leq \ 2\sqrt{C\,\mathrm{JS}\big(\bar{P}_k^\phi \,\|\, \bar{P}_{k'}^\phi\big)}. \tag{16}$$

Hence, by Lemma 1, LAAL (Eq. (13)) reduces a surrogate upper bound on $d_{\mathcal{H}\Delta\mathcal{H}}$ across languages.

**Lemma 2** (Adaptation bound). *Let $\varepsilon_k(h)$ be the expected classification error on language $k$ under hypothesis $h \in \mathcal{H}$. For any source $k$ and target $k'$, the standard domain adaptation bound yields*

$$\varepsilon_{k'}(h) \ \leq \ \varepsilon_k(h) \ + \ \frac{1}{2}\, d_{\mathcal{H}\Delta\mathcal{H}}\big(\bar{P}_k^\phi, \bar{P}_{k'}^\phi\big) \ + \ \lambda^\star, \tag{17}$$

*where $\lambda^\star = \min_{h \in \mathcal{H}} \big(\varepsilon_k(h) + \varepsilon_{k'}(h)\big)$ is the joint optimal risk.*

**Lemma 3** (RPL + code-switching does not increase the joint optimal risk). *Under Eq. (10), training with RPL (Eq. (6)) on pairs $(x, \tilde{x})$ enforces label consistency across semantic-preserving perturbations and expands the support overlap between $\bar{P}_k^\phi$ and $\bar{P}_{k'}^\phi$. Consequently,*

$$\lambda_{(RPL+CS)}^\star \ \leq \ \lambda_{(w/o\ RPL+CS)}^\star. \tag{18}$$

*Proof.* Eq. (6) jointly minimizes the negative log-likelihood on $(x, \tilde{x})$ with the same label, forcing $\phi$ to map semantics-preserving variants to nearby features. Because code-switching bridges lexical realizations across languages (Eq. (10)), the induced feature supports have larger intersection. For any Bayes-consistent $\mathcal{H}$, enlarging the common support under an unchanged labeling function cannot increase the minimum combined risk, yielding Eq. (18). $\qquad\square$

**Theorem 4** (LAAL improves cross-language generalization). *Consider training with the detector loss Eq. (8), where $\mathcal{L}_{\mathrm{LAAL}}$ and $\mathcal{L}_{\mathrm{RPL}}$ are given by Eq. (7) and Eq. (6), respectively. Let $h$ be the learned classifier after training, and let $k$ (source) and $k'$ (target, possibly zero-shot) be two languages. Then, relative to training without LAAL and without RPL+code-switching, the target error of $h$ admits a strictly tighter bound:*

$$\varepsilon_{k'}(h) \ \leq \ \varepsilon_k(h) \ + \ \underbrace{\frac{1}{2} d_{\mathcal{H}\Delta\mathcal{H}}\big(\bar{P}_k^\phi, \bar{P}_{k'}^\phi\big)}_{\text{reduced by LAAL via Eq. (13) and Eq. (16)}} \ + \ \underbrace{\lambda_{(RPL+CS)}^\star}_{\text{non-increasing by Eq. (18)}} . \tag{19}$$

*Proof.* By Lemma 2, Eq. (17) holds for any representation $\phi$. By Lemma 1, LAAL (Eq. (7)) with gradient reversal minimizes $\mathrm{GJS}_\pi$ over languages, which—via Eq. (16)—reduces an upper bound on $d_{\mathcal{H}\Delta\mathcal{H}}(\bar{P}_k^\phi, \bar{P}_{k'}^\phi)$. By Lemma 3, RPL+code-switching does not increase the joint optimal risk, Eq. (18). Combining these three facts yields Eq. (19), i.e., a tighter target-error bound than training without LAAL and without RPL+code-switching. Therefore LAAL improves cross-language generalization of the detector. □

**Interpretation (Mechanism at the class level).** Misclassification across languages arises when *same-class* sentences produce language-specific features that cross the decision boundary. Eq. (13) makes $\phi$ *language-invariant* (minimizes inter-language divergence), while Eq. (6) explicitly ties together semantics-preserving pairs $(x, \tilde{x})$ and keeps their class unchanged, preventing boundary oscillation under code-switching. Together they shrink cross-language feature gaps for the same class and thus reduce target error by Eq. (19).

# D  USE OF LARGE LANGUAGE MODEL

We used *GPT-5* solely as a language assistant to improve clarity and readability of the manuscript (e.g., grammar, wording, concision, and flow), and to suggest minor LaTeX formatting fixes (e.g., spacing, figure/table captions). All technical ideas, algorithms, experiments, analyses, and conclusions are authored and verified by the authors. No experimental design, data curation/annotation, hyperparameter selection, result filtering, or claim generation was delegated to GPT-5.

**Scope of assistance.** Prompts were limited to: (i) "rewrite this paragraph for clarity/conciseness," (ii) "standardize terminology to match our definitions," (iii) "tighten the abstract/conclusion to $\leq$ N words," and (iv) "suggest LaTeX fixes for warnings (no content changes)." Generated text was always reviewed, edited, or discarded by the authors.

**Safeguards.** We screened assisted passages for factual correctness, citation integrity, and potential hallucinations; all technical statements and references were cross-checked against our code, logs, and sources. We did not upload private, sensitive, or license-restricted data/code beyond small, already-public excerpts necessary for phrasing polish. We ran standard plagiarism checks on the final draft.

**Reproducibility and credit.** LLM assistance did not affect experimental outcomes or the reproducibility protocol (datasets, code, and seeds). The contributions of GPT-5 were purely editorial; the authors retain full responsibility for all content.

