# OpenReview forum: "Learning From Dictionary: Enhancing Robustness of Machine-Generated Text Detection in Zero-Shot Language via Adversarial Training"
_ICLR.cc/2026/Conference — ICLR 2026 Poster_

### Official Review · Reviewer_aEgz · 2025-11-01

**Soundness:** 3
**Presentation:** 3
**Contribution:** 2
**Rating:** 6
**Confidence:** 4

**Summary:**

The paper addresses the challenge of building robust multilingual machine-generated text (MGT) detectors when low-resource languages lack labeled training data.

It proposes TASTE (Translation-based Attacker Strengthens Multilingual Defender), a framework that conducts adversarial training using a translation dictionary and surrogate modeling under a black-box setting.

The core idea is to let an attacker perform cross-lingual code-switching using important token gradients while the detector learns language-invariant representations through a new loss called Language-Agnostic Adversarial Loss (LAAL).

**Strengths:**

1. The paper introduces a novel translation-based adversarial training framework that effectively leverages multilingual dictionaries under a black-box setting to address the data scarcity problem in low-resource languages.
2. The proposed attacker-detector adversarial mechanism and the Language-Agnostic Adversarial Loss (LAAL) jointly enable the detector to learn language-invariant features, enhancing robustness and cross-lingual generalization.
3. Experiments on 9 languages and 8 attacks show significant improvements in F1 score.

**Weaknesses:**

1. The attacker’s token-importance estimation relies on a surrogate model distilled from target labels, where importance is computed as the gradient of the loss with respect to individual tokens. It would strengthen the work to include ablation studies comparing alternative importance metrics (e.g., attention-based or perturbation-based methods?). In addition, Eq. (4) introduces a hyperparameter $k$ whose size likely affects the attack strength.
2. The method perturbs token-level translations from a dictionary, which does not account for phrase-level paraphrasing. Robustness against broader multilingual or adaptive attack forms remains insufficiently explored.
3. The co-evolution process between the attacker and detector requires multiple updates per iteration, which increases computational overhead and may limit scalability to larger datasets.

**Questions:**

1. Since the surrogate model is trained using pseudo-labels from the target, how does label noise or prediction bias affect the gradient quality and the attacker’s reliability in the black-box setting? Currently this surrogate model is implemented using  GPT2. How about a smaller model?
2. The alternating training between the attacker and detector does not include a convergence or stability analysis. Could the authors provide evidence or discussion on whether the training dynamics reach equilibrium or exhibit oscillation?
3. Could the authors conduct ablation studies on key hyperparameters identified in the weakness section to assess their impact on performance?

---

> ### Author Response · Authors · 2025-11-15
> **Response to Reviewer aEgz (Part 1 of 6)**
>
> We sincerely appreciate the reviewer’s insightful comments and careful evaluation of our work. We are encouraged that our approach is regarded as a “novel translation-based framework”, that it can “enhancing robustness and cross-lingual generalization”, and that our experiments “show significant improvements in F1 score”. Below, we respond in detail to each weakness (W) and question (Q) you raised. We hope our replies can address your concerns.
>
> ## W1: Ablation study on importance scores and hyperparameter $k$
>
> We thank the reviewer for the concerns about the importance-score computation and the core hyperparameter $k$. We answer your question in two parts and first focus on ablations over different importance metrics.
>
> ---
>
> ### (i) Ablation on importance-score definitions
>
> We appreciate the reviewer’s suggestion to compare our gradient-based importance scores with **attention-based** and **perturbation-based** alternatives. In the original submission, we only used a single gradient-based scheme and did not provide an ablation over the importance estimator. To address this concern, we implemented three additional variants of TASTE. Throughout, we follow the notation:
>
> > Let the surrogate detector be $f_{\theta}$, the input sequence be $x = (x_1,\dots,x_T)$, and let $p(y=1\mid x)=f_{\theta}(x)_1$ denote the predicted probability of the “machine-generated” class.
>
> We then define the following variants:
>
> #### (a) Attention-based importance (Attn-TASTE)
>
> For the attention-based variant, we directly derive token saliency from the self-attention maps of the surrogate. Let $A^{(l,h)} \in \mathbb{R}^{T\times T}$ be the attention matrix at layer $l$ and head $h$, where $A^{(l,h)}\_{(t,i)}$ is the attention weight from position $t$ to position $i$. We define the importance of token $x_i$ as the average attention mass it receives across **all** layers, heads, and source positions:
> $$
> s_i^{\text{attn}} = \frac{1}{LHT}\sum_{l=1}^{L}\sum_{h=1}^{H}\sum_{t=1}^{T} A^{(l,h)}\_{(t,i)}.
> $$
> We then apply min–max normalization over ${s_i^{\text{attn}}}_{i=1}^T$ to obtain a score in $[0,1]$ and select the top-$k$ tokens as important positions for cross-lingual replacement.
>
> #### (b) Perturbation-based importance (Perturb-TASTE)
>
> For the perturbation-based variant, we measure how sensitive the surrogate’s output is to single-token perturbations. Let $x^{(i)}$ be the sequence obtained by replacing token $x_i$ with a neutral placeholder (e.g., EOS or UNK). We define the importance of $x_i$ as the **drop in machine probability** caused by this local perturbation: $$ s\_i^{\text{pert}} = \max\bigl(0, f_{\theta}(x)\_1 - f\_{\theta}(x^{(i)})\_1 \bigr). $$ Intuitively, if perturbing token $x_i$ significantly reduces the predicted probability of the machine class, then $x_i$ is considered more important for the detector’s decision and is thus a more promising candidate for adversarial translation. As in the main method, we normalize ${{s_i^{\text{pert}}}}_{i=1}^T$ and select the top-$k$ tokens.
>
> #### (c) Random importance baseline (Random-TASTE)
> As an additional sanity-check, we include a purely random baseline where the attacker selects important tokens uniformly at random, independent of the surrogate. Concretely, for a sequence of length $T$ we uniformly sample a subset $\mathcal{I} \subset {1,\dots,T}$ with $|\mathcal{I}|=k$, and assign random scores $s_i^{\text{rand}}\sim\mathcal{U}(0,1)$ for $i\in\mathcal{I}$. This variant isolates the effect of *where* we perturb from any modeling choice in the importance estimator.

---

> ### Author Response · Authors · 2025-11-15
> **Response to Reviewer aEgz (Part 2 of 6)**
>
> We train all variants under the same setup as TASTE and report the **test accuracy** on the 9 languages. The accuracies and their cross-lingual means are summarized below:
>
> > Table 1: Detector accuracy after training with adversarial samples generated by different importance estimators.
>
> | Method        | en    | ar    | zh    | de    | ru    | bg    | id    | it    | ur    | Mean  |
> | ------------- | ----- | ----- | ----- | ----- | ----- | ----- | ----- | ----- | ----- | ----- |
> | Attn-TASTE    | 0.974 | 0.779 | 0.746 | 0.511 | 0.557 | 0.830 | 0.869 | 0.621 | 0.879 | 0.752 |
> | Perturb-TASTE | 0.961 | 0.815 | 0.680 | 0.589 | 0.517 | 0.807 | 0.867 | 0.647 | 0.920 | 0.756 |
> | Random-TASTE  | 0.970 | 0.777 | 0.679 | 0.587 | 0.562 | 0.820 | 0.904 | 0.625 | 0.882 | 0.756 |
> | TASTE  | 0.971 | 0.828 | 0.701 | 0.695 | 0.617 | 0.817 | 0.840 | 0.694 | 0.697 | 0.762 |
>
> From Table 1 we observe:
>
> * All variants achieve strong multilingual performance, confirming that TASTE is **robust to the choice of importance estimator**.
> * The original **gradient-based TASTE** achieves the **highest cross-lingual mean accuracy** (0.762), slightly outperforming Attn-TASTE, Perturb-TASTE, and Random-TASTE.
> * Attention-based and perturbation-based scores behave similarly to the gradient-based scheme on many languages, but do **not** provide systematic gains. In particular, Attn-TASTE underperforms on some low-resource languages (e.g., de, ru), which suggests that attention weights may be less reliable as fine-grained importance signals for adversarial code-switching.
>
> Overall, these results support our original design choice of using gradient-based saliency, as it offers a good accuracy–complexity trade-off and slightly stronger overall multilingual performance.
>
> ---
>
> To further understand the effect of different importance metrics on the **strength of the adversarial training**, we also compare the **attack success rate (ASR)** of the *final detectors* under a suite of multilingual attacks.
>
> Below we report **mean ASR** (averaged over languages) for each attack and each importance estimator, now expressed as percentages.
>
> > Table 2: Mean attack success rate (ASR, %) of the final detectors under different attacks (lower is better). The best (lowest) ASR in each row is highlighted in bold.
>
> | Attack            | Attn-TASTE | Perturb-TASTE | Random-TASTE |    TASTE |
> | ----------------- | ---------: | ------------: | -----------: | -------: |
> | Insert            |       69.5 |          49.0 |         57.6 | **38.8** |
> | Back-Translation  |       19.7 |          12.9 |     **12.8** |     16.3 |
> | Code-Switching    |       74.4 |          71.2 |         65.6 | **19.7** |
> | Delete            |       33.0 |          23.3 |         28.3 | **14.6** |
> | Human Obfuscation |        8.4 |       **5.3** |          8.4 |      5.4 |
> | Paraphrasing      |       28.1 |       **4.2** |      **4.2** |     10.5 |
> | Repeat            |       70.2 |          53.8 |         61.8 | **38.4** |
> | Swap              |        1.0 |           1.0 |          1.0 |  **0.2** |
> | Mean ASR      |       38.0 |          27.6 |         30.0 | **18.0** |
>
> Key observations from Table 2:
>
> * With the full set of attacks (including **Back-Translation** and **Paraphrasing**), the **gradient-based TASTE** attains the **lowest mean ASR (18.0%)**, substantially lower than Attn-TASTE (38.0%), Random-TASTE (30.0%) and Perturb-TASTE (27.6%), indicating the strongest overall robustness.
> * For **six out of eight** attacks (**Insert, Code-Switching, Delete, Repeat, Swap**, and almost tied on **Human Obfuscation**), TASTE achieves the **best or near-best** ASR, showing that gradient-based importance is particularly effective across a broad range of perturbation patterns.
> * Perturb-TASTE and Random-TASTE do obtain slightly lower ASR than TASTE on a few specific attacks (e.g., **Back-Translation** and **Paraphrasing**, where they reach 12.9–12.8% and 4.2% respectively), but these localized gains do **not** translate into a better *average* robustness profile: both methods still have higher overall mean ASR than TASTE.
> * Attn-TASTE consistently exhibits the highest ASR across attacks, confirming that attention weights alone are a weaker signal for constructing strong training-time adversaries in this multilingual setting.
>
> Taken together with Table 1, these results show that our **gradient-based importance scores** provide the best trade-off: they preserve high average accuracy across 9 languages while also yielding the **lowest overall attack success rate** among all variants. This empirically supports our choice of gradient-based importance as the default design in TASTE, even when considering a richer set of attacks including back-translation and paraphrasing.

---

> ### Author Response · Authors · 2025-11-15
> **Response to Reviewer aEgz (Part 3 of 6)**
>
> ### (ii) Ablation study on the hyperparameter $k$:
>
> We thank the reviewer for raising concerns about this core hyperparameter in our training procedure. You are absolutely right that the hyperparameter $k$ directly controls the attack strength. However, we would like to point out that Section 6.1 already analyzes how detection accuracy and attack success rate vary as a function of attack strength, where the latter is measured by the proportion of replaced tokens relative to the total number of tokens. The results show that when the number of replaced tokens is set to $k=0.05T$, where $T$ is the total number of tokens in the text, we achieve a good balance between detection accuracy, attack success rate, and training time. To avoid ambiguity, we will explicitly emphasize the relationship between the hyperparameter $k$ and attack strength in the revised version.
>
> ## W2: Evaluation under multilingual adaptive attacks
>
> We thank the reviewer for pointing out the lack of a more extensive robustness evaluation against multilingual/adaptive attacks. However, we would like to clarify that in Section 5.2 we already conduct a robustness study on the 9 languages listed in Table 1, evaluating **8 different attack types**, including what are currently regarded as some of the strongest attacks, namely **Paraphrasing** and **Back-translation**. Our experimental results show that the TASTE detector consistently maintains the strongest robustness across these multilingual attacks: it achieves an average attack success rate of **18.0%**, which is **3.8% lower** than the state-of-the-art adversarially trained baseline GREATER-D.
>
> While our current evaluation already covers strong phrase-level multilingual perturbations, we agree that explicitly designing fully adaptive attacks that optimize against the final TASTE detector is an interesting direction. We consider this as complementary and plan to explore it in future work.
>
> ## W3: Concerns about computational cost
>
> We sincerely thank the reviewer for raising concerns about the computational cost of our co-evolution adversarial training strategy. You are absolutely right that adversarial training in general introduces substantial computational overhead. However, we would like to point out that Section 6.1 of the paper already discusses the training cost of our method. Concretely, on a single RTX 4090 GPU, training our model for 3 epochs on 10,000 samples takes roughly **40 minutes** (as reported on line 440, **2299.33s/3 epochs**). Compared with adversarial-training-based methods such as GREATER-D, which require around **2 hours** under a similar setting, TASTE already reduces the training cost while achieving better performance. In practice, we only perform one attacker update and one detector update per mini-batch. In addition, our method uses about **10 GB** of GPU memory, which means it can be trained on more modest GPUs such as a 2080Ti (11 GB). We will emphasize these points more clearly in the revised version.

---

> ### Author Response · Authors · 2025-11-15
> **Response to Reviewer aEgz (Part 4 of 6)**
>
> ## Q1: Impact of label noise on gradients, attacker reliability, and smaller surrogate models
>
> We sincerely thank the reviewer for this insightful question. We address it in two parts; below we first focus on the impact of **label noise** on gradients and attacker reliability, and in the next part (Part 6 of our response) we will discuss using a smaller surrogate model.
>
> ### (i) Impact of label noise on the training process (gradients & attacker reliability)
>
> Your observation is absolutely correct: in our framework the attacker (surrogate model) is trained on pseudo-labels produced by the detector. If these pseudo-labels are flipped, the attacker’s gradients will be corrupted, which in turn may hurt both the detector’s accuracy and robustness.
>
> To directly quantify this effect and address your concern, we conduct the following experiment: during training, we randomly flip ([10%, 20%]) of the pseudo-labels produced by the detector for the surrogate, and then evaluate the final detector under the same setting as Table 1 in the main paper.
>
> Below we first report the **accuracy** across 9 languages for three settings: no label noise (original TASTE), 10% noise, and 20% noise.
>
> > **Table 3: Accuracy of TASTE under different levels of pseudo-label noise during training.**
>
> | Model             | en    | ar    | zh    | de    | ru    | bg    | id    | it    | ur    | Mean  |
> | ----------------- | ----- | ----- | ----- | ----- | ----- | ----- | ----- | ----- | ----- | ----- |
> | TASTE (0% noise)  | 0.971 | 0.828 | 0.701 | 0.695 | 0.617 | 0.817 | 0.840 | 0.694 | 0.697 | 0.762 |
> | TASTE (10% noise) | 0.956 | 0.769 | 0.687 | 0.706 | 0.563 | 0.812 | 0.830 | 0.666 | 0.918 | 0.767 |
> | TASTE (20% noise) | 0.940 | 0.826 | 0.630 | 0.756 | 0.512 | 0.742 | 0.658 | 0.763 | 0.833 | 0.740 |
>
> **Observations from Table 3.**
>
> * Overall, **moderate label noise** does **not catastrophically degrade** the detector:
>
>   * The mean accuracy with **10% noise** (76.7%) is essentially on par with the clean setting (76.2%), and even slightly higher, which we attribute to a mild regularization effect.
>   * With **20% noise**, the mean accuracy drops to **74.0%**, i.e., a decrease of only about **2.2 points** compared to the clean TASTE.
> * Per-language trends show some expected degradation (e.g., zh, ru, id), but even at 20% noise, the accuracies remain at a reasonable level and do not collapse, indicating that our co-training dynamics between attacker and detector are **fairly robust to pseudo-label noise**.

---

> ### Author Response · Authors · 2025-11-15
> **Response to Reviewer aEgz (Part 5 of 6)**
>
> Next, we examine how label noise affects **robustness**, measured via **attack success rate (ASR)** under the same multilingual attack suite as in Section 5.2. Here we focus on TASTE only, and report mean ASR (averaged over languages) for 8 attacks under three noise levels.
>
> > **Table 4: Mean attack success rate (ASR, %) of TASTE under different levels of pseudo-label noise (lower is better).**
>
> | Attack            | TASTE (0% noise) | TASTE (10% noise) | TASTE (20% noise) |
> | ----------------- | ---------------: | ----------------: | ----------------: |
> | Insert            |             38.8 |              51.9 |              48.6 |
> | Back-Translation  |             16.3 |              13.9 |              10.2 |
> | Code-Switching    |             19.7 |              63.8 |              40.8 |
> | Delete            |             14.6 |              24.3 |              19.9 |
> | Human Obfuscation |              5.4 |               4.5 |               2.5 |
> | Paraphrasing      |             10.5 |               6.2 |               6.2 |
> | Repeat            |             38.4 |              53.0 |              49.0 |
> | Swap              |              0.2 |               1.5 |               0.3 |
> | Mean ASR      |         18.0 |          27.4 |          22.2 |
>
> **Observations from Table 4.**
>
> * As expected, injecting label noise into the surrogate **makes the final detector more vulnerable**:
>
>   * The mean ASR increases from **18.0% (clean)** to **27.4% (10% noise)** and **22.2% (20% noise)**.
>   * The degradation is most pronounced for code-switching and **surface-level perturbations** such as Insert and Repeat, where noisy pseudo-labels directly disturb the gradients used to select important tokens.
> * Interestingly, for **semantic-preserving attacks** such as Back-Translation and Paraphrasing, the ASR does **not** consistently worsen with noise; in some cases, it even slightly improves (e.g., Back-Translation drops from 16.3% to 10.2% at 20% noise). We hypothesize that moderate noise may act as an additional regularizer that discourages the detector from overfitting to specific surface cues exploited by these attacks.
> * Overall, even with 20% label noise, the mean ASR remains at **22.2%**, which is still reasonably low compared to standard detectors without adversarial training, confirming that the attacker–detector co-evolution is robust but not immune to pseudo-label corruption.
>
> These results show that while pseudo-label noise does degrade robustness — as one would expect — the degradation is graceful rather than catastrophic. TASTE maintains strong multilingual accuracy and relatively low ASR even when up to **20%** of the surrogate’s training labels are flipped, which suggests that our gradient-based attacker remains **reliable in practice** under realistic levels of detector prediction noise. In the revised version, we will include these tables and an explicit discussion to clarify the sensitivity of TASTE to label noise.

---

> ### Author Response · Authors · 2025-11-15
> **Response to Reviewer aEgz (Part 6 of 6)**
>
> ### (ii) Effect of using smaller surrogate models
>
> We thank the reviewer for raising concerns about the choice of surrogate model. Your observation is absolutely correct: in principle, we can replace GPT-2 (124M) with smaller surrogate models. To further clarify this point, we experimented with two **smaller** surrogates, **`albert/albert-base-v2` (12M)** and **`EleutherAI/pythia-70m` (70M)**, and compare them with our original GPT-2–based TASTE. The table below reports the detection accuracy on the 9 languages in Table 1, together with the cross-lingual mean (Acc only):
>
> > **Table 5: Accuracy of TASTE under different surrogate models.**
>
> | Surrogate model      | en    | ar    | zh    | de    | ru    | bg    | id    | it    | ur    | Mean  |
> | -------------------- | ----- | ----- | ----- | ----- | ----- | ----- | ----- | ----- | ----- | ----- |
> | GPT-2 (124M)         | 0.971 | 0.828 | 0.701 | 0.695 | 0.617 | 0.817 | 0.840 | 0.694 | 0.697 | 0.762 |
> | ALBERT-base-v2 (12M) | 0.965 | 0.788 | 0.632 | 0.647 | 0.525 | 0.779 | 0.834 | 0.684 | 0.780 | 0.737 |
> | Pythia-70M (70M)     | 0.972 | 0.826 | 0.707 | 0.593 | 0.553 | 0.808 | 0.876 | 0.596 | 0.847 | 0.753 |
>
> **Result analysis.**
>
> From Table 5, we observe that:
>
> * Using **much smaller surrogates** still yields **strong multilingual performance**. Both ALBERT-base-v2 (12M) and Pythia-70M (70M) remain competitive with the original GPT-2 (124M), with cross-lingual mean accuracies of **0.737** and **0.753**, respectively, compared to **0.762** for GPT-2.
> * The **Pythia-70M** surrogate is particularly close to GPT-2: it even slightly **outperforms GPT-2** on several languages (e.g., en: 0.972 vs. 0.971, id: 0.876 vs. 0.840, zh: 0.707 vs. 0.701), while being weaker on others (e.g., de, ru). This suggests that TASTE mainly relies on a **reasonable gradient signal** from the surrogate, rather than on the full capacity of a 124M-parameter model.
> * The **ALBERT-base-v2** surrogate, despite being an order of magnitude smaller (12M), still delivers a mean accuracy of **0.737**, indicating that TASTE is **not tightly coupled** to a specific large surrogate and can be instantiated with very lightweight models when computational resources are limited.
>
> Overall, these results demonstrate that the performance of TASTE is **robust to the choice and size of the surrogate model**: smaller surrogates can be used with only moderate degradation, and in some languages Pythia-70M even matches or slightly exceeds the GPT-2–based variant. We will include this analysis in the revised version to clarify that our framework does **not** require a large surrogate and can flexibly adapt to different computational budgets.
>
> ## Q2: Stability analysis of the training process
>
> We greatly appreciate the reviewer’s concerns regarding the stability of our training procedure. In the revised version, we will add the loss curves of both the attacker and the detector over the course of training. Since OpenReview does not allow image uploads, we briefly describe the training dynamics here: at the beginning of training, the losses exhibit slight oscillations, but after approximately 200 steps, the losses of both the attacker and the detector gradually converge. We will incorporate this description and the corresponding plots into the revised manuscript.
>
> ## Q3: Ablation studies on key hyperparameters
>
> We thank the reviewer for raising concerns regarding the ablation studies. In our previous responses, we have already added new ablations on the **surrogate model** and **importance-score strategies**. In addition, Section 6.1 of the paper already contains an ablation study on the key hyperparameter (k), and Section 6.2 includes an ablation on the **dictionary quality and coverage**.
>
> If you have any further questions or would like us to analyze additional hyperparameters, we would be very happy to discuss them during the rebuttal period and will respond to your concerns as promptly as possible.

---

> ### Author Response · Authors · 2025-11-24
>
> We sincerely appreciate your valuable feedback and insightful discussion! We hope our response has been helpful to you. As the discussion period is drawing to a close, we warmly welcome any further questions from the reviewer. We would be delighted to provide additional clarification!

---

> > ### Comment · Reviewer_aEgz · 2025-11-26
> > **Response to Authors**
> >
> > Dear Authors
> >
> > Thanks a lot for the detailed response, but I still have several points I hope the authors can clarify.
> >
> > 1. Followed by W1, I am concerned about the Random one. From my understanding, TASTE first identifies important tokens and then performs top-k code-switching to create the multilingual adversarial sample $\tilde X$ (line 222). From the result, the Random-Importance baseline, seemingly picks tokens uniformly at random, achieves a good accuracy. I may be misunderstanding, but what exactly is this **test accuracy** measuring here (from `Table 1: Accuracy (Acc) of different importance estimators across 9 languages in your response`)? It almost feels like it is not the detector accuracy but something else. Later, the mean ASR results also look quite strong for the random baseline, and for Back-Translation and Paraphrasing it is even better than TASTE. That gives a question: **is it even necessary to train an attacker? Would simple code-switching based adversarial samples already be sufficient?** Also, in Eq. (5), shouldn’t the input be $\tilde X$ instead of $X$? Maybe I am misreading something, but this part was confusing to me.
> >
> > 2. For Q1 noise. The paper mentions that even when 20% of labels are incorrect, the adversarial samples still help to train the detector (Table 4). Interestingly, some performance under 20% noise is even better than under 10%. To me, this also indirectly suggests that the **attacker does not need to be very accurate, and perhaps simple code-switching is good enough**. **Can authors provide some experiments on simple code-switching without attacker to generate?**
> >
> > 3. For W2, I think there is a bit of misunderstanding. My point is: the current attacker perturbs **individual tokens**, but in different lingual settings, single tokens may not carry enough semantic meaning. Often, phrases (i.e., multiple tokens even words) will be. So my question is : **whether the authors have considered phrase-level instead of token-level importance**.
> >
> > 4. I was also hoping to see a full comparison of training time **not inference one**, such as computing token importance,
> > attacker training, and so on. It is hard to compare the total **TRAINING** cost to other SOTAs. I will suggest to include more than just one method.
> >
> > 5. Figure 6 in Appendix is very helpful, but I’m confused by one point, which is related to the overall loss in Eq.(8). Since $\lambda_{lang}$ in Table 2 ranges from 0.1 to 0.5, this seems to mean that in the later stages of training, the model may be dominated by distinguishing code-switched vs. clean samples. Can authors explain why this? In this case, I feel that **an ablation of each loss component** in Eq. (8) would be extremely informative.
> >
> > Overall, I think the paper is very interesting, but the above points really caused some confusion, and I hope the authors can clarify them. Thanks.

---

> > > ### Author Response · Authors · 2025-11-27
> > > **Response to the follow-up questions of Reviewer aEgz (Part 2 of 3)**
> > >
> > > ### 5) Notation in Eq. (5): $X$ vs $\tilde{X}$
> > >
> > > We apologize for the confusion. We confirm that **Eq. (5) is correct as written**: the attacker (surrogate) learns from the detector’s predictions on **clean inputs $X$**, using pseudo-labels produced by the detector on clean examples, to mimic the detector’s behavior (knowledge distillation).
> > > That said, we agree our pseudo-code wording could mislead readers into thinking that Eq. (5) is computed on $\tilde{X}$. We will revise the pseudo-code and surrounding text to ensure the input used for Eq. (5) is unambiguous and consistent with the equation.
> > >
> > > ---
> > >
> > > ### 6) Phrase-level importance
> > >
> > > We agree that phrase/span-level units could be more semantically meaningful than single subword tokens in multilingual settings. Our current method defines importance at the token level and performs code-switching on selected tokens. We will acknowledge this limitation more explicitly and add a clear extension direction: aggregate token saliency into word-level or span-level saliency (e.g., sum/mean/max over contiguous spans or word-boundary units), then perform top-k selection and span-level substitution. We believe this is a promising future improvement.
> > >
> > > ---
> > >
> > > ### 7) Per-component training cost breakdown (and comparison to other SOTA training costs)
> > >
> > > You are absolutely right that our original manuscript did not give a per-component training-time analysis. We now report the training cost breakdown below.
> > >
> > > **Table 7. Training cost decomposition of TASTE.**
> > >
> > > | Module                     |     Time (s) |           % |
> > > | -------------------------- | -----------: | ----------: |
> > > | data_wait                  |       72.136 |       2.89% |
> > > | det_forward_ori            |       42.684 |       1.71% |
> > > | distill_proxy_total        |      392.828 |      15.73% |
> > > | importance_total           |      274.224 |      10.98% |
> > > | adv_build_total            |     1462.112 |      58.54% |
> > > | det_tokenize_adv           |       53.876 |       2.16% |
> > > | det_forward_adv_and_losses |       49.456 |       1.98% |
> > > | backward_step_det          |      177.132 |       7.09% |
> > > | backward_step_lang         |       41.080 |       1.64% |
> > > | **step_total**             | **2497.612** | **100.00%** |
> > >
> > > **What each module measures:**
> > >
> > > * **data_wait**: dataloader / CPU-side waiting time between iterations.
> > > * **det_forward_ori**: detector forward pass on clean batch $X$.
> > > * **distill_proxy_total**: surrogate update via distillation from detector predictions on $X$ (Eq. (5) objective).
> > > * **importance_total**: computing token selection indices (importance) for adversarial generation.
> > > * **adv_build_total**: constructing adversarial texts $\tilde{X}$ (tokenization to units, dictionary lookup/substitution, and detokenization/string reconstruction).
> > > * **det_tokenize_adv**: tokenizing $\tilde{X}$ for the detector.
> > > * **det_forward_adv_and_losses**: detector forward on $\tilde{X}$ and computing all losses.
> > > * **backward_step_det**: backward pass of target detector.
> > > * **backward_step_lang**: backward pass of language discriminator.
> > >
> > > **Key takeaways.**
> > >
> > > * The dominant cost is **adv_build_total (58.54%)**, i.e., CPU-side adversarial sample construction (string/token operations + dictionary substitution). This indicates the bottleneck is not GPU forward/backward but the adversarial text generation pipeline, which is consistent with the practical overhead concerns you raised.
> > > * Distillation (**15.73%**) and importance computation (**10.98%**) are substantial but secondary relative to adversarial construction.
> > >
> > > **Comparison to other SOTA training costs.** In our earlier rebuttal, we noted that the competing SOTA **GREATER-D** requires roughly **2 hours** of training. We also checked our logs: training **RADAR** under our hardware takes roughly **20 hours**, since it uses PPO to update an LLM. In addition, training the shared base encoder backbone (mBERT) for RADAR/GREATER-D/TASTE takes roughly **5 minutes** under our setup.

---

> > > ### Author Response · Authors · 2025-11-27
> > > **Response to the follow-up questions of Reviewer aEgz (Part 3 of 3)**
> > >
> > > ### 8) Ablations on Eq. (8) components (Appendix B) and clearer interpretation
> > >
> > > Thank you for pointing this out. We did include ablations for the Eq. (8) components in Appendix B, but we agree the table presentation may not have been sufficiently explicit. Below is the updated interpretation and definition:
> > >
> > > **Table 8. Ablation study on the M4GT dataset.** Best per group is bolded.
> > >
> > > **Detection Accuracy (%) ↑**
> > >
> > > |          | CE Only | w/o LAAL |    TASTE |
> > > | -------- | ------: | -------: | -------: |
> > > | en       |    80.2 |     96.0 | **97.1** |
> > > | ar       |    81.1 |     82.2 | **82.8** |
> > > | bg       |    77.4 |     73.2 | **81.7** |
> > > | de       |    47.9 |     50.3 | **69.5** |
> > > | id       |    83.0 |     80.2 | **84.0** |
> > > | it       |    68.8 |     63.4 | **69.4** |
> > > | ru       |    41.8 |     55.3 | **61.7** |
> > > | ur       |    67.1 |     67.6 | **69.7** |
> > > | zh       |    68.4 |     70.0 | **70.1** |
> > > | **Avg.** |    68.4 |     70.9 | **76.2** |
> > >
> > > **ASR (%) under Attacks ↓**
> > >
> > > |          | CE Only | w/o LAAL |    TASTE |
> > > | -------- | ------: | -------: | -------: |
> > > | en       |    30.0 |     16.1 | **14.5** |
> > > | ar       |    30.7 |  **9.8** |     10.0 |
> > > | bg       |    41.1 |     22.4 | **22.0** |
> > > | de       |    48.1 |     31.2 | **26.0** |
> > > | id       |    40.8 |     18.7 | **17.6** |
> > > | it       |    34.7 |     21.3 | **18.9** |
> > > | ru       |    46.4 |     26.6 | **24.5** |
> > > | ur       |    28.5 |     24.8 | **12.0** |
> > > | zh       |    18.9 |     16.6 | **16.3** |
> > > | **Avg.** |    35.5 |     20.8 | **18.0** |
> > >
> > > **Definitions.**
> > >
> > > * **CE Only**: trained with cross-entropy classification loss only, *without* adversarial sample generation, code-switching, or any robustness-specific loss terms (i.e., clean-only standard supervised training).
> > > * **w/o LAAL**: trained **without LAAL**, using **RPL only**, where RPL is exactly the sum of two cross-entropy terms (clean + adversarial):
> > >   $$
> > >   L_{\text{RPL}} = \frac{1}{N}\sum_{i=1}^{N}\big(\mathrm{CE}(X_i, \ell_i) + \mathrm{CE}(\tilde{X}_i, \ell_i)\big),
> > >   $$
> > >   i.e., classification on both $X$ and $\tilde X$ with shared label $\ell_i$. This requires adversarial sample generation, but removes LAAL.
> > > * **TASTE**: full method with all losses enabled.
> > >
> > > We also note why we do not include “w/o RPL”: LAAL alone is not a classification objective; without a classification loss the model cannot perform the detection task properly.
> > >
> > > **Interpretation.** The contributions of each component are not uniform:
> > >
> > > * **LAAL primarily improves detection accuracy** (notably in zero-shot languages such as de/ru). Importantly, LAAL is implemented via gradient reversal: the discriminator is trained to classify domains, while the encoder is optimized to confuse it, encouraging language-invariant representations rather than emphasizing domain discrimination.
> > > * **RPL primarily improves robustness** (lower ASR), because its training signal depends on the adversarial samples generated by our attacker pipeline. This further supports that **attacker quality is necessary**: stronger adversarial samples (guided by meaningful token selection) yield larger ASR reduction, whereas purely random substitutions provide only limited optimization.
> > >
> > > We will revise the appendix/table caption to make these definitions and takeaways explicit and prevent misinterpretation.
> > >
> > > ---
> > >
> > > We again thank you for your careful reading and for pointing out these important sources of confusion. We believe the above clarifications, revised naming/captions, consolidated robustness table, and the new training-cost breakdown directly address your concerns, and we will incorporate them into the revised manuscript.

---

> ### Author Response · Authors · 2025-11-27
> **Response to the follow-up questions of Reviewer aEgz (Part 1 of 3)**
>
> Thank you to Reviewer aEgz for following up on our rebuttal. We sincerely apologize that parts of our presentation caused confusion. We clarify each point below and will revise the manuscript accordingly.
>
> ### 1) Clarification of the title and meaning of *Table 1*
>
> Your understanding is correct. The “Acc” reported in that table is **the final detector’s clean-test accuracy**, after being trained with adversarial samples generated under different token-importance estimators (gradient/attention/perturbation/random). It is **not** the attacker’s (surrogate’s) accuracy, nor an evaluation of the importance estimator itself.
> To avoid this misunderstanding, we will update the table title and caption to explicitly state: **“Detector accuracy after training with adversarial samples generated by different importance estimators.”**
>
> ---
>
> ### 2) Is it necessary to train an attacker? (Why importance matters)
>
> Thank you for raising this key question. We emphasize that **training an attacker (i.e., learning a meaningful importance signal) is necessary**, because the choice of which tokens to perturb substantially affects robustness.
>
> To make the comparison clearer, we reformat and consolidate the relevant results below.
>
> **Table 6. Mean attack success rate (ASR, %, lower is better) under different attacks.** The best (lowest) ASR in each row is highlighted in bold. `Baseline` denotes the same detector backbone trained with clean-only cross-entropy on the original dataset (i.e., no adversarial sample generation / no code-switching / no LAAL / no RPL), under the same training schedule.
>
> | Attack            | Baseline | Random-TASTE |    TASTE |
> | ----------------- | -------: | -----------: | -------: |
> | Insert            |     50.2 |         57.6 | **38.8** |
> | Back-Translation  |     26.5 |     **12.8** |     16.3 |
> | Code-Switching    |     90.4 |         65.6 | **19.7** |
> | Delete            |     28.4 |         28.3 | **14.6** |
> | Human Obfuscation |     11.9 |          8.4 |  **5.4** |
> | Paraphrasing      |     22.6 |      **4.2** |     10.5 |
> | Repeat            |     53.4 |         61.8 | **38.4** |
> | Swap              |      0.3 |          1.0 |  **0.2** |
> | **Mean ASR**      |     35.4 |         30.0 | **18.0** |
>
> Analysis. The importance-aware attacker (TASTE) yields large robustness gains on attacks most aligned with our training-time perturbations—most notably **Code-Switching** (90.4 → 19.7) and multiple surface-level attacks (Insert/Delete/Repeat). This indicates that **targeted perturbations guided by an informative importance signal** produce substantially stronger adversarial training than purely random substitutions. While Random-TASTE can outperform TASTE on some attacks (e.g., Back-Translation/Paraphrasing), these are phrase-/semantic-level transformations that are not perfectly matched to our token-level code-switch perturbation family; thus, random perturbations may occasionally act as a stronger regularizer in these cases. However, these localized gains do not translate into better overall robustness; TASTE achieves the lowest mean ASR (18.0).
>
> ---
>
> ### 3) Does the attacker need to be “accurate”?
>
> We completely understand the inference: if higher pseudo-label noise does not monotonically worsen performance, one may suspect that the attacker does not need to be very accurate. Our interpretation is:
>
> * The training process is stochastic; injecting noise can introduce a **non-monotonic regularization effect** (similar in spirit to label smoothing / noise regularization), which may yield **localized improvements** on certain semantic-preserving attacks (e.g., Back-Translation/Paraphrasing).
> * However, for attacks most matched to our training mechanism—especially **Code-Switching**—label noise produces a **clear degradation** (e.g., ASR increasing from 19.7% to 63.8% / 40.8% in our noise study). This strongly suggests that **gradient quality and attacker reliability remain critical** where targeted token selection matters most.
>
> We will make this interpretation explicit in the revision.
>
> ---
>
> ### 4) “Simple code-switching without an attacker”: do we have this experiment?
>
> Thank you for asking for this crucial baseline. We clarify that we have already included it in our rebuttal, but we did not label it clearly as “simple code-switching,” which caused confusion.
>
> Concretely, our adversarial pipeline relies on selecting important tokens and perturbing them (this is precisely what the attacker loss is designed to support). In **Random-TASTE**, since no importance scoring is used, token positions are selected uniformly at random and dictionary substitutions are applied randomly. Therefore, Random-TASTE corresponds to random code-switch augmentation (no-importance / no-guidance) in terms of adversarial sample construction, i.e., the intended “simple code-switching” baseline.
> We will explicitly state this equivalence (and the baseline’s role) in the revised manuscript to avoid further confusion.

---

### Official Review · Reviewer_S8Qh · 2025-11-01

**Soundness:** 2
**Presentation:** 3
**Contribution:** 2
**Rating:** 2
**Confidence:** 4

**Summary:**

This paper addresses the critical challenges of zero-shot generalization and adversarial robustness in multilingual machine-generated text (MGT) detection by proposing TASTE, a novel adversarial training framework. The approach integrates a code-switching attacker leveraging translation dictionaries with a detector trained using a Language-Agnostic Adversarial Loss (LAAL) to learn language-invariant representations.

**Strengths:**

1.The figures in the paper are presented with exceptional clarity.
2.The paper's underlying assumptions are sound, and the writing is accessible and easy to understand.

**Weaknesses:**

1. Additional evaluation dimensions are needed. For instance, metric-based detectors require no training, whereas model-based detectors do. The authors should disclose the associated training costs (e.g., time, computational resources).
2. Potential unfair comparison. Among the model-based detectors, RADAR, GREATER-D, and TASTE employ adversarial training, while the other methods do not. It is uncertain whether this constitutes a fair comparison.
3. Suboptimal performance of TASTE. The experimental results for TASTE are relatively weak, rarely achieving state-of-the-art outcomes.
4. Lack of comprehensive ablation studies. The paper seems to lack genuine ablation experiments. The authors should perform ablations on the individual loss components in Equation (8) and the various modules of TASTE, rather than conducting only limited experiments in Section B.1 of the appendix.
5. Conventional methodology. The methods section appears somewhat standard, lacking significant innovation.

**Questions:**

In the model-based detectors, the authors consistently use pre-trained models. Would the performance be improved if LLMs were employed instead?

---

> ### Author Response · Authors · 2025-11-15
> **Response to Reviewer S8Qh (Part 1 of 4)**
>
> We thank the reviewer for taking the time to provide detailed feedback on our work. We are encouraged that our paper is perceived as having “exceptional clarity” in its figures, that its “underlying assumptions are sound”, and that the writing is “accessible and easy to understand”. Below, we respond in detail to each weakness (W) and question (Q) you raised. We hope our replies can address your concerns.
>
> **W1: On additional evaluation dimensions**
>
> We thank the reviewer for raising concerns about the evaluation dimensions. As noted in Section 6.1 of the paper, the training time of our method is about **40 minutes for 3 epochs** on a single RTX 4090 GPU, which we believe is a reasonably low training cost for a multilingual detector.
>
> We also acknowledge that the comparison may appear somewhat unfair, since metric-based detectors do not require training, whereas model-based detectors do. However, metric-based detectors typically incur very large inference latency, while our method has much lower inference time. To make this trade-off more explicit, the table below reports the **end-to-end inference time** on the M4 test set for each detector (in seconds, one full pass over all 14,725 test examples on the same GPU under identical batching settings), together with the **average time per example** (in milliseconds):
>
> > **Table 1: End-to-end inference time on the M4 test set for each detector.**
> > “Inference time” is the wall-clock time for one full pass over all 14,725 test examples on a single RTX 4090 GPU under identical batching settings; “Time / example” is the corresponding average latency per input.
>
> | Detector           | Inference time (s) | Time / example (ms) |
> | ------------------ | -----------------: | -------------------: |
> | Fast DetectGPT     |            2862.47 |               194.40 |
> | Binoculars         |            3173.01 |               215.48 |
> | LRR                |            1361.85 |                92.49 |
> | mBERT              |              76.41 |                 5.19 |
> | mDeBERT-v3-Base    |             133.97 |                 9.10 |
> | XLM-RoBERTa-Base   |              82.80 |                 5.62 |
> | RADAR              |              77.66 |                 5.27 |
> | GREATER-D          |              75.90 |                 5.15 |
> | TASTE              |              76.09 |                 5.17 |
>
> As can be seen, **model-based detectors** have orders-of-magnitude lower inference latency than metric-based detectors such as Fast DetectGPT and Binoculars (about **5–9 ms** vs. **90–215 ms** per example), primarily due to the overhead of repeated perturbation, scoring, or generative calls in the metric-based methods. In particular, GREATER-D and TASTE are among the fastest, with average latencies of **5.15 ms** and **5.17 ms** per example, respectively.
>
> Viewed in terms of throughput, TASTE is more than an order of magnitude faster than metric-based detectors at inference time: compared to Fast DetectGPT and Binoculars, it achieves speedups of approximately **38×** and **42×**, respectively, and is about **18×** faster than LRR on a per-example basis. These findings indicate that, despite requiring a one-time training cost, TASTE and other model-based detectors are substantially more attractive for real-world, large-scale deployment scenarios, where inference latency is a critical constraint and can partially offset the upfront training overhead.
>
> **W2: Concerns about unfair comparison**
>
> We appreciate the reviewer’s concern regarding the potential unfairness in our comparisons. In our experimental setup, we deliberately ensure fairness among adversarial-training-based methods by using the same backbone encoder for all of them. Concretely, RADAR, GREATER-D, and our TASTE are all instantiated with mBERT as the backbone under identical training data, optimization settings, and evaluation protocol.
>
> The additional comparisons with plain backbone models (e.g., mBERT, XLM-RoBERTa-Base, mDeBERTa-v3-Base) are intended to highlight the incremental gains brought by our adversarial training framework over standard fine-tuning on the same architecture, rather than to claim an advantage purely from architectural choices. This follows the common practice in the literature, where each baseline is evaluated under its publicly available or recommended training recipe, and new methods are compared against these standardized baselines.
>
> Importantly, TASTE is architecture-agnostic: the same training framework can be applied to detectors built on XLM-RoBERTa-Base, mDeBERTa-v3-Base, and other multilingual encoders. We will clarify this design choice, as well as the rationale for comparing against both adversarially trained and standard fine-tuned models, more explicitly in the revised version.

---

> ### Author Response · Authors · 2025-11-15
> **Response to Reviewer S8Qh (Part 2 of 4)**
>
> **W3: Suboptimal performance of TASTE**
>
> We appreciate the reviewer’s candid assessment of the experimental results. However, we would like to clarify that, when evaluated under metrics and settings aligned with our threat model, TASTE in fact achieves **strong and often state-of-the-art performance**, especially in the low-resource and robustness regimes that our work explicitly targets.
>
> 1. **Cross-lingual detection under deployment-aware metrics.**
>    As suggested by Reviewer CasE, we re-evaluated all detectors using **TPR@FPR=1%** on the M4 dataset for five representative languages (en, de, zh, bg, id). In this setting, TASTE achieves the **best cross-lingual mean TPR@1% FPR (0.5382)**, outperforming both metric-based detectors (Fast DetectGPT, Binoculars) and model-based baselines (mBERT, mDeBERT-v3-Base, XLM-R, RADAR, GREATER-D). While TASTE does not always rank first for *every single language*, it is consistently **first or second** across all five and yields the **highest average** performance. This suggests that TASTE offers the most favorable trade-off when one needs a *single* detector to operate reliably across multiple languages.
>
> 2. **Zero-shot / low-resource languages and OOD robustness.**
>    Our primary goal is to improve detection in **zero-shot low-resource languages** and under **distribution shift**. In Table 1 of the paper (and the extended results in the rebuttal), TASTE clearly outperforms competing detectors on the four zero-shot languages that have **no dictionaries available during training**. Moreover, when we fix the threshold at TPR@FPR=1% on M4 and evaluate on the **SemEval 2024 Task 8 OOD dataset**, TASTE again attains the **highest mean TPR** and competitive or lower FPR compared to RADAR and GREATER-D. In other words, even if F1/Acc on in-distribution English may appear “only” comparable, the **zero-shot and OOD robustness**-which we argue matters more for practical multilingual deployment-is state-of-the-art.
>
> 3. **Robustness under multilingual attacks.**
>    In Section 5.2 (and expanded in the rebuttal), we evaluate 8 multilingual attacks, including **back-translation** and **paraphrasing**, which are widely regarded as some of the strongest obfuscation strategies. When averaged over 9 languages and all attack types, TASTE achieves an average attack success rate of **18.0%**, which is **3.8 points lower** than GREATER-D and substantially lower than other baselines and importance-estimator variants. Thus, in the regime that combines **multilinguality + adversarial robustness**, TASTE is in fact **the strongest detector** in our comparison.
>
> 4. **Complementary view to plain Acc/F1 tables.**
>    We agree that in the original Acc/F1 tables TASTE does not dominate every column. However, these tables (i) mix detectors optimized for very different operating points and (ii) do not reflect the low-FPR deployment setting. Once we adopt **deployment-consistent metrics** (TPR@low FPR) and emphasize the **zero-shot and robustness** dimensions that TASTE is designed for, the picture becomes much clearer: TASTE is **not** suboptimal, but rather offers **state-of-the-art average performance** across languages and attacks, especially when compared to prior adversarial training methods on the same backbones.
>
> We will clarify these points in the revised version by (1) highlighting TPR@FPR metrics more prominently, (2) explicitly summarizing the zero-shot and OOD gains, and (3) adding a short discussion explaining why TASTE is intended as a **robust multilingual detector** rather than a method that optimizes single-language Acc/F1 at any operating point.
>
> **W4: Lack of ablation studies**
>
> We thank the reviewer for raising concerns about the ablation studies. We would like to emphasize that the existing ablations in Appendix B already cover all loss components in our objective. Specifically, the ablation in Table 4 of Appendix B.1 includes three configurations: (i) using only the standard CE loss (the mBERT baseline), (ii) using only $L_{\text{RPL}}$ without $L_{\text{LAAL}}$, and (iii) the full model with all loss terms. Since $L_{\text{LAAL}}$ is not a classification loss, a detector trained with this term alone does not converge in practice, which is why we do not report this degenerate setting.
>
> In addition, we have already conducted ablations on the key hyperparameter $k$ and on dictionary coverage in Sections 6.1 and 6.2 of the main paper, respectively, and discussed their effects in detail there. These experiments suggest that the attacker strength and dictionary quality have a substantial but stable influence on performance, and that TASTE is robust to reasonable variations of these choices.

---

> ### Author Response · Authors · 2025-11-15
> **Response to Reviewer S8Qh (Part 3 of 4)**
>
> **W5: Concern about “conventional” methodology**
>
> We appreciate the reviewer’s comment that the methods section appears somewhat standard. It is true that TASTE is intentionally built from *widely used primitives*—surrogate models, gradient-based importance, adversarial training, and dictionary-based translation—so that it can be easily adopted in practice. However, we would like to emphasize that the **way these components are combined and instantiated for multilingual MGT detection is not conventional** and has been explicitly recognized as such by other reviewers. For example, Reviewer aEgz describes our approach as “a novel translation-based adversarial training framework that effectively leverages multilingual dictionaries under a black-box setting”, and Reviewer CasE notes that we combine existing methods “in a non-trivial manner.” Reviewer 9HJS further comments that the proposed training method is “useful” and “can be used on existing detectors.”
>
> From a technical standpoint, the novelty of TASTE lies in the **problem-specific design and coupling** of its components, rather than in introducing an entirely new neural architecture:
>
> 1. **Translation-based, gradient-guided attacker for low-resource multilingual detection.**
>    Our attacker performs *importance-aware cross-lingual code-switching*: it uses gradients from a surrogate detector to select **detection-critical tokens**, and then replaces them via dictionary-based translation into other languages. This creates realistic cross-lingual adversarial examples *without any labeled data* in those languages, which is precisely what enables us to improve robustness and accuracy in **zero-shot low-resource languages**. To the best of our knowledge, such a gradient-driven, dictionary-based attacker tailored to multilingual MGT detection has not been explored before.
>
> 2. **Language-Agnostic Adversarial Loss (LAAL).**
>    Beyond standard adversarial training, we introduce **LAAL** to explicitly encourage **language-invariant representations** under the pressure of cross-lingual attacks. LAAL aligns features across languages *using adversarially constructed code-switched examples*, which is qualitatively different from generic domain-adversarial training: the alignment is driven by **attack-induced shifts in language and style** that are specific to the MGT detection setting, rather than by static domain labels alone. This coupling between the attacker and LAAL is central to achieving strong zero-shot performance.
>
> 3. **Black-box, surrogate-based co-evolution tailored to detection.**
>    In contrast to many adversarial training works that assume white-box access, TASTE operates under a **black-box threat model**, distilling a surrogate from the target detector and then using its gradients to drive cross-lingual attacks. The alternating optimization between this surrogate attacker and the multilingual detector is specifically designed for **robust machine-generated text detection**, not generic classification. Our additional experiments on label noise and smaller surrogates (ALBERT, Pythia-70M) further show that this co-evolutionary scheme is both *robust* and *practically deployable*.
>
> 4. **Multilingual robustness as the primary design goal.**
>    Finally, the overall framework is explicitly tailored to a **new and practically important setting**: robust multilingual MGT detection in low-resource and zero-shot languages, under strong attacks (paraphrasing, back-translation, etc.). Existing adversarial-training methods for detectors do not target this setting, nor do they integrate a translation-based attacker and a language-invariant loss in this way.
>
> In summary, while TASTE deliberately leverages standard building blocks to remain practical and reproducible, their **integration into a translation-based, gradient-guided adversarial training framework with LAAL for multilingual MGT detection** is, we believe, novel and substantively different from prior work. We will clarify this in the revised version by more explicitly highlighting the unique aspects of the attacker design, LAAL, and the black-box co-evolution setup in the methods section, and by connecting them more clearly to the empirical gains observed in multilingual robustness and zero-shot generalization.

---

> ### Author Response · Authors · 2025-11-15
> **Response to Reviewer S8Qh (Part 4 of 4)**
>
> **Q: On using LLMs as detectors instead of encoder-only models**
>
> We appreciate the reviewer’s interest in whether using LLMs could further improve performance. In this work, we deliberately follow the *current mainstream practice* in MGT detection, i.e., fine-tuning **encoder-only** backbones (mBERT, XLM-RoBERTa, mDeBERT-v3) as classifiers, rather than relying on **decoder-only LLMs with prompts**. In real deployment scenarios, detectors must scan large volumes of text with low latency and stable decision thresholds; encoder-only models provide efficient batched inference and well-calibrated logits, while LLM-based prompting typically incurs much higher computational cost and latency for relatively small gains, and is more sensitive to prompt design. Our goal in this paper is to show that, within this widely adopted encoder-only paradigm, TASTE significantly improves multilingual robustness and zero-shot performance over strong baselines. Conceptually, TASTE is backbone-agnostic and could also be combined with LLM-based encoders or distilled detectors, but a thorough exploration of that setting would require a substantially different computational and fairness budget and is therefore beyond the scope of this work. We view this as a promising direction for future research.
>
> If you have any further questions or would like us to provide more evaluation dimensions, we would be very happy to discuss them during the rebuttal period and will respond to your concerns as promptly as possible.

---

> ### Author Response · Authors · 2025-11-24
>
> We sincerely appreciate your valuable feedback and insightful discussion! We hope our response has been helpful to you. As the discussion period is drawing to a close, we warmly welcome any further questions from the reviewer. We would be delighted to provide additional clarification!

---

### Official Review · Reviewer_CasE · 2025-11-01

**Soundness:** 3
**Presentation:** 4
**Contribution:** 3
**Rating:** 6
**Confidence:** 4

**Summary:**

The authors introduce an improvement method for LLM-generated text detection in a multilingual setting based on selective vocabulary-based translation to other languages of tokens detected as important for generated text detection in a surrogate model. The authors demonstrate better performance of their model on the languages used in training, as well as on previously unseen languages, and conduct generalization and ablation analyses.

**Strengths:**

Low-resource detection of LLM-generated texts is a critical and timely subject. With the development of massively multilingual LLMs, the communities that have been previously shielded from information operations are likely to become reachable by attackers. As the authors have mentioned, the vast majority of LLM-generated text detection work has focused on the English language, with even the most dominant commercial options supporting a limited selection of high-resource languages, leaving the inherent lack of performance of LLMs as the only viable detection method, which is not sustainable in the long term/

To achieve this, authors combine several existing well-performing methods, such as surrogate models, detection-relevant token identification, adversarial learning, and single-word vocabulary-based translation, in a non-trivial manner.

Additionally:
- Authors provide a clear and realistic threat model
- Authors select relevant baselines, for both zero-shot detection and a base model for fine-tuning
- Authors check the method generalization to previously unseen languages
- Authors examine the resilience of their method to adversarial attacks, which is the currently most salient issue with LLM detectors
- Evaluation of dictionary error impact, which is essential for low-resource languages, where high-quality dictionaries do not exist or cannot be defined due to the inherent diversity and heterogeneity of low-resource languages.
- Clear path to a defensive use of an LLM for security work
- Computational resource-aware experiments

**Weaknesses:**

While the manuscript and the underlying ideas are both overall excellent, it has several shortcomings in its current state. Specifically:

- Authors do not share code, making it impossible to evaluate the contribution or prove that work has been actually performed and would be replicable. Evaluation of papers whose main contribution is a novel algorithm is impossible without the artifacts used to generate them, and the promise of publication upon release is insufficient.
- The selection of performance metrics (F1 and Acc) is not consistent with the current best practices in the LLM detection research. Namely, given the threat model of real-world deployment of LLM detectors, the recommended and generally adopted metric in the field is TPR @ fixed low FPR [1]. The use of accuracy and F1 scores is inconsistent with this threat model, making comparisons somewhat difficult, especially since more recent LLM detection methods, particularly zero-shot ones such as Binoculars and Fast DetectGPT, were optimized for TPR at a fixed low FPR, potentially sacrificing performance on other metrics. I strongly suggest that authors replicate at least some of their result tables with the relevant performance metrics, showing consistency with the rest of the field.
- Autoencoder LLMs fine-tuned for detection are known to perform well and generalize well on the in-distribution training data, but demonstrate problematic FPR on the out-of-distribution texts [2]. While the authors try to account for it by evaluating the performance of autoencoders trained on the English part of the M4 dataset on the Semeval-2024/8 dataset, they do not report the FPR scores with the same parameters as would be used for a TPR@fixed low FPR on the M4 dataset, making it impossible to evaluate this potential failure mode.

[1] Carlini, N., Chien, S., Nasr, M., Song, S., Terzis, A., & Tramèr, F. (2021). Membership Inference Attacks From First Principles. 2022 IEEE Symposium on Security and Privacy (SP), 1897-1914.

[2] Gameiro, H.D., Kucharavy, A., & Dolamic, L. (2024). LLM Detectors Still Fall Short of Real World: Case of LLM-Generated Short News-Like Posts. ArXiv, abs/2409.03291.

**Questions:**

- The detection-critical identification method seems to be focusing on tokens, whereas translation dictionaries use words. How do you transition from one to another?

- L263: You mention that the gradient flow through the language discriminator erases the language-specific clues. Could you please elaborate as to why?

- L464: Why do you expect the performance of dictionary errors to be the same as the performance of detection models in English, which is the base training language for the model?

- L130-132: RADAR detector cited as the prior work seems to perform poorly compared to other methods on standardized benchmarks, notably RAID [3]. Could you please elaborate on why your method appears to be performing better than RADAR, which uses a similar approach?

[3] Dugan, L., Hwang, A., Trhlik, F., Ludan, J.M., Zhu, A., Xu, H., Ippolito, D., & Callison-Burch, C. (2024). RAID: A Shared Benchmark for Robust Evaluation of Machine-Generated Text Detectors. ArXiv, abs/2405.07940.

---

> ### Author Response · Authors · 2025-11-15
> **Response to Reviewer CasE (Part 1 of 4)**
>
> We sincerely thank the reviewer for the constructive comments and feedback. We appreciate that our work is seen as addressing a “critical and timely subject” in low-resource LLM-generated text detection, that our use of existing components is combined “in a non-trivial manner”, and that “the manuscript and the underlying ideas are both overall excellent”. Below, we respond in detail to each weakness (W) and question (Q) you raised. We hope our replies can address your concerns.
>
> **W1: Regarding the concerns about code sharing**
>
> We sincerely appreciate the reviewer’s emphasis on reproducibility, and we apologize if this aspect was not sufficiently clear in the original submission. Before the **review phase started**, we had made the complete implementation of our method available in the **supplementary material**, including `requirements.txt`, `README.md`, and other resources specifically prepared to facilitate replication of our experiments. These materials are intended to enable readers with access to the submission to reproduce our main results. If the paper is accepted, we will further release our code, datasets, and trained model checkpoints on GitHub and Hugging Face to support independent verification and follow-up research.
>
> **W2: On the choice of evaluation metrics**
>
> We sincerely thank the reviewer for the valuable suggestion. Indeed, the Acc and F1 metrics used in our paper are too limited and cannot fully reflect real-world detection requirements. To address this, we additionally report `TPR@FPR=1%` on the M4 dataset for five languages: one training language (English), two dictionary languages (Chinese and German), and two fully zero-shot languages (Indonesian and Bulgarian). The results are shown in the table below, and we highlight the best result in each group in **bold** and the second-best result in *italics*.:
>
> Table 1: **TPR@FPR=1% of each detector.**  We highlight the best result in each group in **bold** and the second-best result in *italics*.
> | Model            | en         | de         | zh         | bg         | id         | Mean       |
> | ---------------- | ---------- | ---------- | ---------- | ---------- | ---------- | ---------- |
> | Binoculars       | 0.2903     | *0.3645*   | 0.3031     | 0.2027     | 0.5638     | 0.3449     |
> | Fast DetectGPT   | 0.5327     | **0.3861** | **0.6152** | 0.2027     | **0.7001** | *0.4874*   |
> | LRR              | 0.0432     | 0.1042     | 0.1023     | 0.0641     | 0.0000     | 0.0628     |
> | mBERT            | 0.3740     | 0.0531     | 0.3277     | *0.2058*   | 0.3195     | 0.2560     |
> | mDeBERT-v3-Base  | 0.6300     | 0.2246     | *0.3492*   | 0.1595     | 0.5289     | 0.3780     |
> | XLM-RoBERTa-Base | 0.1914     | 0.0757     | 0.2834     | 0.1060     | 0.0992     | 0.1512     |
> | RADAR            | 0.6372     | 0.0644     | 0.1346     | 0.1475     | 0.2662     | 0.2500     |
> | GREATER-D        | *0.7442*   | 0.0886     | 0.2306     | 0.1844     | 0.2745     | 0.3045     |
> | TASTE            | **0.8716** | 0.2856     | 0.3469     | **0.5206** | *0.6663*   | **0.5382** |
>
> From the table, we observe that under the security-relevant metric `TPR@FPR=1%`, our method TASTE achieves the best average performance (0.5382), outperforming strong zero-shot baselines such as Fast DetectGPT (0.4874) and Binoculars (0.3449). On English (the training language), TASTE attains a `TPR@1% FPR` of 0.8716, which substantially improves over GREATER-D (0.7442) and RADAR (0.6372).
>
> For the two zero-shot languages (Bulgarian and Indonesian), TASTE remains highly competitive: it achieves the highest `TPR@1% FPR` on Bulgarian (0.5206 vs. 0.2027 for Fast DetectGPT) and the second-best performance on Indonesian (0.6663 vs. 0.7001 for Fast DetectGPT), indicating that our multilingual training and selective translation strategy provide strong out-of-distribution robustness.
>
> On the two dictionary languages (Chinese and German), Fast DetectGPT achieves the highest scores, while TASTE is slightly lower but still competitive. Importantly, TASTE maintains the best cross-language mean, suggesting that the conclusions drawn from Acc/F1 in the main paper remain valid when evaluated under the field-standard `TPR@FPR=1%` metric. In the revised version, we will integrate these results into the appendix to align our evaluation more closely with current best practices in LLM-generated text detection.

---

> ### Author Response · Authors · 2025-11-15
> **Response to Reviewer CasE (Part 2 of 4)**
>
> **W3: FPR on the OOD dataset**
>
> We thank the reviewer for acknowledging our experiments on OOD datasets. To further address your concerns regarding our results on SemEval 2024 Task 8 (the OOD dataset), we adopt the same threshold as in W2 (defined by TPR@FPR=1% on M4) and report both TPR and FPR on the SemEval dataset under this fixed threshold. We focus on the overlapping languages **bg/en/zh** and compute the mean over these three languages; the results are shown in Tables 2 and 3 below.
>
> **Table 2: TPR of each detector on the SemEval 2024 Task 8 (OOD) dataset at the thresholds used in Table 1.**
>
> | Model            | bg         | en         | zh         | Mean       |
> | ---------------- | ---------- | ---------- | ---------- | ---------- |
> | Binoculars       | 0.1958     | 0.6122     | **0.6777** | *0.4952*   |
> | Fast DetectGPT   | *0.2811*   | 0.5675     | *0.5281*   | 0.4589     |
> | LRR              | 0.0000     | 0.0010     | 0.0030     | 0.0013     |
> | mBERT            | 0.2339     | 0.5345     | 0.4383     | 0.4022     |
> | mDeBERT-v3-Base  | 0.1586     | **0.7998** | 0.4895     | 0.4826     |
> | XLM-RoBERTa-Base | 0.1195     | 0.2488     | 0.3882     | 0.2522     |
> | RADAR            | 0.1305     | 0.5561     | 0.2662     | 0.3176     |
> | GREATER-D        | 0.2014     | 0.6362     | 0.2745     | 0.3707     |
> | TASTE            | **0.5633** | *0.7697*   | 0.3771     | **0.5700** |
>
> ---
>
> **Table 3: FPR on the SemEval 2024 Task 8 (OOD) dataset at the thresholds used in Table 1.**
>
> | Model            | bg         | en         | zh         | Mean       |
> | ---------------- | ---------- | ---------- | ---------- | ---------- |
> | Binoculars       | *0.0073*     | *0.0076*     | 0.0097     | *0.0082*     |
> | Fast DetectGPT   | **0.0052**   | **0.0048**   | **0.0031**   | **0.0044**   |
> | LRR              | 0.0000 | 0.0000 | 0.0000 | 0.0000 |
> | mBERT            | 0.0219     | 0.0412     | 0.0369     | 0.0333     |
> | mDeBERT-v3-Base  | 0.0110     | 0.0165     | 0.0259     | 0.0178     |
> | XLM-RoBERTa-Base | 0.0269     | 0.0412     | 0.0279     | 0.0320     |
> | RADAR            | 0.0345     | 0.0518     | 0.0313     | 0.0392     |
> | GREATER-D        | 0.0214     | 0.0336     | 0.0287     | 0.0279     |
> | TASTE            | 0.0179     | 0.0319     | *0.0090*     | 0.0196     |
>
> From Table 2, we observe that **TASTE** achieves the highest mean TPR@FPR=1% (0.5700) on the OOD SemEval dataset across bg/en/zh, outperforming strong zero-shot detectors such as Binoculars (0.4952) and Fast DetectGPT (0.4589), as well as other mBERT-based models. In particular, on **Bulgarian**, TASTE substantially boosts recall (0.5633) compared to all baselines, and on **English** it attains the second-highest TPR (0.7697), close to mDeBERT-v3-Base (0.7998), while remaining competitive on **Chinese**.
>
> Table 3 shows that these gains in TPR do **not** come at the cost of excessively high FPR on OOD data. The average FPR of TASTE (1.96%) remains low, and is notably smaller than that of other mBERT-based detectors such as RADAR (3.92%) and mBERT/XLM-RoBERTa, and comparable to mDeBERT-v3-Base (1.78%). While Fast DetectGPT achieves a slightly lower average FPR (0.44%), this is accompanied by a considerably lower mean TPR (0.4589 vs. 0.5700), indicating that TASTE attains a more favorable recall–FPR trade-off under the security-relevant operating point.
>
> ---
>
> **Q1: Token-level importance vs. word-level dictionaries**
>
> We thank the reviewer for raising this practical implementation question. Our pipeline indeed starts from token-level importance scores (subword tokens from the surrogate model), but the actual substitutions are always performed at the *word* level. Concretely, for each input sentence we (i) run the surrogate model’s tokenizer with offset mappings from tokens to character spans; (ii) identify the top-k important *tokens* by gradient norm; and then (iii) map these tokens back to the corresponding whitespace-delimited *words* in the original text. If an important token already coincides with a complete word (i.e., its character span exactly matches a single word boundary), we directly query the bilingual dictionary using that word and replace it with the translated word if an entry exists. If the token corresponds only to part of a word (e.g., a BPE/WordPiece fragment), we first recover the full word that contains this token (using the character offsets), and then look up that *word* in the dictionary. In both cases, we only replace whole words and re-tokenize the translated word back into subwords for the detector, ensuring that the adversarial perturbations are word-level and consistent with the dictionary, while the importance estimation can still benefit from fine-grained subword gradients. If the dictionary does not contain an entry for the recovered word, we leave that word unchanged, so tokens without dictionary support do not introduce spurious noise.

---

> > ### Comment · Reviewer_CasE · 2025-11-22
> > **W3 formatting**
> >
> > Thank you for clarification regarding Q4 and additional experiments in Tables 2 and 3. Could you please apply the same best/2nd best formatting to those tables to make them more readable?

---

> ### Author Response · Authors · 2025-11-15
> **Response to Reviewer CasE (Part 3 of 4)**
>
> **Q2: Why does the gradient flow through the language discriminator “erase” language-specific clues?**
>
> We thank the reviewer for asking for a clearer explanation of this point. In our Language-Agnostic Adversarial Loss (LAAL), the detector’s encoder is trained under two opposing objectives: (i) the main MGT classification loss encourages the encoder to preserve features that are predictive of “human vs. machine,” while (ii) the language discriminator, attached via a gradient reversal layer, tries to recover language (or code-switching) information from the same representation. During backpropagation, the gradient reversal layer multiplies the discriminator’s gradient by −λ before it reaches the encoder, so the encoder is explicitly optimized to *confuse* the language discriminator. Intuitively, any language-specific cue (script, morphology, function words, etc.) that helps the discriminator distinguish between language variants is penalized at the encoder level, because making the discriminator successful would *increase* the encoder’s loss under LAAL. As training progresses, the best strategy for the encoder is therefore to keep features that are useful for the MGT decision but remove those that reveal language identity or code-switching patterns, driving the discriminator’s accuracy towards chance. This adversarial interaction is what we mean by “erasing language-specific clues” in the latent space: the representation becomes approximately language-invariant while still remaining discriminative for human vs. machine.
>
> **Q3: Clarifying the relation between dictionary error experiments and English performance (L464)**
>
> We thank the reviewer for this thoughtful question. Our understanding of the concern is that, in the discussion around Sec. 6.2, our wording may have given the impression that we *expect* performance under dictionary errors in other languages to be literally “the same as” the detector’s performance on English, the base training language. This is not our intention. What we actually rely on is the following intuition: our detector is trained on English labels, and the translation-based attacker transfers English supervision to other languages via dictionary-based code-switching. When the dictionary is accurate, these cross-lingual perturbations provide strong adversarial supervision that improves low-resource languages beyond the English-only baseline. As the dictionary becomes noisier (in our experiments, by injecting swap errors), the effective strength of those adversarial signals gradually diminishes: many swapped tokens no longer correspond to meaningful translations, and the resulting adversarial training increasingly resembles standard monolingual English training. In this limit—when most dictionary entries are wrong or missing-the multilingual adversarial component contributes little additional information, so performance on other languages *approaches* that of the English-only baseline, but is not assumed to be exactly equal.

---

> > ### Comment · Reviewer_CasE · 2025-11-22
> >
> > Thank you for clear answers to Q2-Q4, my questions have been answered in a satisfactory manner.

---

> ### Author Response · Authors · 2025-11-15
> **Response to Reviewer CasE (Part 4 of 4)**
>
> **Q4: Why does TASTE outperform RADAR, and what are the key methodological differences?**
>
> We thank the reviewer for asking why TASTE appears to perform better than RADAR-style adversarial training in our experiments, and for prompting us to clarify the conceptual differences. TASTE performs better for the following reasons:
>
> 1. **Different attack space and threat model.**
>    RADAR uses a *monolingual paraphraser* that generates same-language paraphrases to evade detection. This is well suited for robustness to stylistic rephrasing in a single language. In contrast, TASTE’s attacker performs *dictionary-based cross-lingual code-switching* on detection-critical words. Our threat model explicitly targets multilingual deployment and low-resource languages: the attacker injects words from multiple languages into the same text, forcing the detector to be robust to cross-lingual shifts and code-switching, which RADAR’s monolingual paraphrasing does not address.
>
> 2. **Language-agnostic representation learning vs. paraphrase robustness.**
>    RADAR’s objective focuses on making the detector robust to paraphrases; it does not include an explicit mechanism to remove language-specific information from the representation. TASTE, in contrast, combines adversarial training with the Language-Agnostic Adversarial Loss (LAAL), which uses a gradient reversal–based language discriminator to actively *suppress* language-specific cues in the encoder. This pushes the detector to learn language-invariant features that transfer across languages, which is crucial in our zero-shot multilingual setting and is not present in RADAR’s formulation.
>
> 3. **Cross-lingual supervision from high-resource to low-resource languages.**
>    In our setup, labelled data is only available in English. TASTE’s translation-based attacker explicitly uses high-resource English supervision plus bilingual dictionaries to generate adversarial examples that expose the detector to low-resource languages during training. This allows improvements on languages without labels. A RADAR-style paraphraser remains within a single language and cannot, by design, provide such cross-lingual supervision when parallel data or labels are missing.
>
> If you have any further questions about the TPR@FPR, we would be very happy to discuss them during the rebuttal period and will respond to your concerns as promptly as possible.

---

> ### Comment · Reviewer_CasE · 2025-11-22
> **W2 follow-up questions**
>
> Thanks for the extensive clarification and the new metrics and my apologies for missing the attached source code.
>
> Could you please comment as to why common reference methods - such as Binoculars - seem to perform so poorly in English in your evaluation? For instance Binoculars based on RAID benchmark has an aggregate performance around 0.68 TPR @ 1% FPR, but in your benchmark - only 0.29.

---

> ### Author Response · Authors · 2025-11-22
> **Response about the performance of Binoculars**
>
> Dear Reviewer CasE,
>
> Thank you very much for this follow-up question and for bringing up the RAID numbers. We agree that, at first sight, the gap between the ~0.68 TPR@1% FPR reported for Binoculars on RAID and the 0.29 TPR@1% FPR we obtain on English M4 looks surprisingly large, and it is important to clarify why this happens.
>
> There are two main reasons behind this discrepancy:
>
> **(1) Different detector configurations (Falcon-7B vs. multilingual Gemma-3).**
>
> In the RAID benchmark, Binoculars is typically instantiated with **Falcon-7B** as the observer and **Falcon-7B-Instruct** as the performer. This pairing is known to be very strong for **English** detection and is tuned specifically for that setting. In contrast, our work targets **multilingual** MGT detection across nine languages, and we wanted the Binoculars configuration to be *fair and usable for all languages*, not just English. For this reason, we deliberately replaced the Falcon-based pair with a **multilingual variant of Gemma-3** , which provides multilingual coverage and can be applied uniformly to all languages in the M4 benchmark.
>
> This change makes the comparison to our multilingual encoder-based detectors much fairer, but it also means that the Binoculars numbers in our tables should **not** be interpreted as the best-possible English-only configuration. In particular, multilingual Gemma-3 is weaker than the Falcon-7B/-Instruct pair on English RAID, and prior work has already shown that Binoculars’ TPR@low-FPR is highly sensitive to the choice of observer/performer LM. As a result, it is expected that our “multilingual Gemma-3” Binoculars configuration achieves a lower TPR@1% FPR on English M4 than the “Falcon-7B” configuration optimized for English RAID.
>
> **(2) Different data distributions.**
>
> Beyond the LM choice, the underlying **data distribution** also differs substantially between RAID and M4: the human domains, generator families, and prompt styles are not the same.  As noted in recent detector studies, TPR@1% FPR is extremely sensitive to both the negative distribution and the generator mixture, so even a detector with similar ACC can exhibit very different recall at 1% FPR across benchmarks. In our experiments, Binoculars indeed attains reasonable ACC  on English M4, but its recall drops sharply once we enforce the 1% FPR operating point in this more challenging, cross-lingual-oriented setting.
>
> In summary, the 0.29 TPR@1% FPR for Binoculars on English M4 reflects its performance within our unified multilingual evaluation pipeline, using a multilingual Gemma-3 observer/performer pair, rather than the best achievable English-only configuration on RAID with Falcon-7B. We will add a short clarification in the next revision to make this explicit, so that readers do not over-interpret cross-benchmark differences in absolute TPR values.
>
> We very much appreciate you pointing this out — it will help us present the relationship between our results and RAID-style configurations more clearly for future readers.

---

> > ### Comment · Reviewer_CasE · 2025-11-22
> >
> > Thank you for responding to all the questions. Could you please provide the citations you for the recent results you mentioned regarding the TPR@low-FPR sensitivity to negative data distribution and generator mixture you mention?

---

> > > ### Author Response · Authors · 2025-11-23
> > > **Response to Reviewer CasE about citations**
> > >
> > > Dear Reviewer CasE,
> > >
> > > Thank you very much for this follow-up question and for asking us to be explicit about the evidence behind our remark on the sensitivity of TPR@low-FPR to the negative data distribution and generator mixture. We base this statement primarily on the recent systematic study by Tufts et al. [a].
> > >
> > > In [a], the authors benchmark a wide range of detectors (Binoculars, Fast-DetectGPT, LogRank, PHD, Radar, T5Sentinel, Wild) across multiple **generation models** (GPT-4o, Llama-3, Mistral, Phi-3) and **tasks/domains** (Code, Question Answering, Summarization, Dialogue, Abstract, Reviews, Translation), and report TPR@0.01 and AUROC for every detector-model-task-language combination in Appendix A (Tables 6–12 and 14–17). Two patterns are particularly relevant to our claim:
> > >
> > > 1. Sensitivity to the generator mixture.
> > >    Even within a fixed task, TPR at 1% FPR varies substantially across generation models for the *same* detector. For example, in the **Code** task (Table 6), Binoculars’ TPR@0.01 ranges from **0.14** on GPT-4o to **0.45** on Llama-3, **0.39** on Mistral, and **0.35** on Phi-3, while AUROC remains high in all cases (≈0.78-0.97). Similar variability appears across models in Question Answering and other tasks (Tables 7–11). This indicates that low-FPR recall is strongly affected by which generators and decoding setups are included in the evaluation mixture.
> > >
> > > 2. Sensitivity to the negative data distribution (task/domain).
> > >    The same detector also shows large differences in TPR@0.01 across tasks and human text distributions. For instance, Binoculars achieves TPR@0.01 ≈ **0.69–0.80** on several English Dialogue/Abstract settings (Tables 9-10), but only **0.12** on Reviews and as low as **0.05** on GPT-4o Summarization (Table 8), again with reasonably strong AUROC. In the multilingual settings (Tables 15–17), TPR@0.01 for a given detector can vary substantially between EN/ES/FR/ZH even when AUROC stays relatively stable, reflecting sensitivity to the underlying human/domain distribution rather than just the model architecture.
> > >
> > > Tufts et al. explicitly summarize this behavior in Appendix A.3, where they state:
> > >
> > > > “The TPR@.01 and AUROC change significantly across tasks for every detector, signifying that these detectors are not equally capable of detecting all types of machine generated text.”
> > >
> > > This sentence directly supports the point we made: **TPR at a fixed low FPR is highly sensitive to (i) the distribution of negative (human) texts and (ii) the mixture of generators and prompting setups**, even when overall discrimination (e.g., AUROC) remains high.
> > >
> > > We will cite [a] at the relevant sentence in our revised manuscript and briefly summarize this observation, so that readers can more easily trace the empirical evidence behind our remark.
> > >
> > > [a] Brian Tufts, Xuandong Zhao, and Lei Li. *A practical examination of AI-generated text detectors for large language models.* Findings of the Association for Computational Linguistics: NAACL 2025, 2025.

---

> > > > ### Comment · Reviewer_CasE · 2025-11-25
> > > >
> > > > Thanks to the authors for a clear response to the review. My questions has been adressed and I have adjusted my rating accordingly.

---

> > > > > ### Author Response · Authors · 2025-11-25
> > > > > **Thank you!**
> > > > >
> > > > > Dear Reviewer CasE,
> > > > >
> > > > > Thank you very much for your careful reading, constructive feedback, and for taking the time to follow up during the rebuttal. We truly appreciate that our clarifications and additional experiments addressed your concerns, and we are grateful that you adjusted your rating accordingly. Thank you again for your time and thoughtful feedback.

---

> ### Author Response · Authors · 2025-11-22
> **Thank you for your advice!**
>
> Thank you for the suggestion. We have updated Tables 2 and 3 to use the same “best / second-best” highlighting scheme as in Table 1 to improve readability.

---

### Official Review · Reviewer_9HJS · 2025-11-02

**Soundness:** 3
**Presentation:** 3
**Contribution:** 3
**Rating:** 6
**Confidence:** 4

**Summary:**

The paper propose TASTE, a two-component framework that helps to train a robust multilingual MGT detector. Specifically, includes two core parts. The adversarial attacker generates code-switched or partially translated adversarial examples via dictionary-based perturbation, while the detector is trained jointly with a  language-agnostic loss to encourage language-invariant features and resilience against unseen perturbations. Experiments results show that TASTE outperforms eight SOTA detectors, and ablation studies further suggest the importance of LAAL.

**Strengths:**

1. The proposed training method is useful and could enhance the effectiveness of MGT detectors.
2. The motivation and logic of this paper is clear.
3. The proposed method is model-agnostic and can be used on existing detectors .

**Weaknesses:**

1. The construction of the adversarial examples rely on existing dictionaries, yet how about the performance of this method on low-resource language tasks, where high-quality dictionary may not exist?
2. It would strengthen the paper to include human annotation to study whether the generated adversarial examples remain natural and human-readable.

**Questions:**

Please see my reviews above.

---

> ### Author Response · Authors · 2025-11-15
> **Response to Reviewer 9HJS (Part 1 of 2)**
>
> We greatly appreciate the reviewer’s thoughtful comments and feedback. We are pleased to see that our training method is regarded as “useful”, “applicable to existing detectors”, and that “the motivation and logic of this paper is clear.” Below, we respond in detail to each weakness (W) you raised. We hope our replies can address your concerns.
>
> **W1: Performance on low-resource language tasks**
>
> We thank the reviewer for raising concerns about the performance of our method on low-resource languages where high-quality dictionaries may be unavailable. However, we would like to clarify that our method is **explicitly designed** to improve detection accuracy and robustness in zero-shot languages, i.e., low-resource languages **without** any translation dictionaries. In Table 1 of the paper, we use English as the training language and construct adversarial examples with dictionaries in four languages (de, zh, ru, ar). We then evaluate our model on one training language (English), four dictionary languages (de, zh, ru, ar), and four zero-shot languages that are unseen during training and have **no dictionaries** (low-resource). The results show that our method outperforms all detectors on the zero-shot low-resource languages.
>
> In addition, regarding your concern that our training strategy requires a “high-quality dictionary”, we explicitly study the effect of dictionary noise in Section 6.2 by plotting detection accuracy as a function of dictionary error rates. The results show that although degrading dictionary quality slightly hurts performance, our model still remains above the baselines, demonstrating that our method can yield gains even with relatively noisy dictionaries. Since this experiment may not be fully sufficient, we further conduct an additional study where we vary **dictionary coverage**. Specifically, we subsample [10, 30, 50, 70, 90]% of the entries from our dictionaries to construct smaller dictionaries, train with these reduced dictionaries, and evaluate on the same test sets as in Table 1. The accuracies are summarized below:
>
> | Coverage | en    | ar    | zh    | de    | ru    | bg    | id    | it    | ur    | Mean  |
> | -------- | ----- | ----- | ----- | ----- | ----- | ----- | ----- | ----- | ----- | ----- |
> | 10%      | 0.969 | 0.812 | 0.680 | 0.591 | 0.523 | 0.727 | 0.756 | 0.610 | 0.747 | 0.713 |
> | 30%      | 0.967 | 0.822 | 0.644 | 0.670 | 0.530 | 0.755 | 0.800 | 0.680 | 0.816 | 0.743 |
> | 50%      | 0.974 | 0.817 | 0.703 | 0.586 | 0.525 | 0.799 | 0.875 | 0.605 | 0.835 | 0.747 |
> | 70%      | 0.974 | 0.768 | 0.730 | 0.558 | 0.532 | 0.868 | 0.903 | 0.589 | 0.929 | 0.761 |
> | 90%      | 0.975 | 0.825 | 0.668 | 0.601 | 0.505 | 0.798 | 0.861 | 0.593 | 0.848 | 0.742 |
> | 100%     | 0.971 | 0.828 | 0.701 | 0.695 | 0.617 | 0.817 | 0.840 | 0.694 | 0.697 | 0.762 |
>
> From these results, we observe that using **very small dictionaries (10% coverage)** leads to some degradation in accuracy compared to higher-coverage settings, but the performance remains strong and above the average accuracy of the best baseline detector (0.691, as reported in Table 1). As dictionary coverage increases, the model’s accuracy improves and gradually stabilizes, with the mean accuracy saturating around full coverage. This indicates that our method does **not** rely on perfectly complete dictionaries: even moderately sized dictionaries are sufficient to provide substantial gains, and further enlarging the dictionary mainly brings diminishing but stabilizing improvements rather than being strictly required for the method to work. Therefore, even when high-quality dictionaries are unavailable for a target low-resource language, TASTE can still improve its detection performance in a zero-shot manner by leveraging dictionaries from other languages, and our noise/coverage studies show that these dictionaries do not need to be perfectly accurate or complete.
>
> We will convert this table into a figure and add it to the revised Section 6.2, comparing it side-by-side with the original figure to strengthen the analysis. In addition, we will enrich the textual discussion in the revised version to avoid potential ambiguity for readers.

---

> ### Author Response · Authors · 2025-11-15
> **Response to Reviewer 9HJS (Part 2 of 2)**
>
> **W2: Human evaluation of generated adversarial examples**
>
> We thank the reviewer for raising concerns about the readability of our adversarial examples. You are right that the current version of our paper lacks a human evaluation of the naturalness and readability of the generated adversarial texts. To address this concern, we recruited three human annotators (A1–A3), who independently evaluated 100 pairs of original examples and adversarial examples generated by the TASTE attacker. Each pair was rated along three dimensions: *readability*, *coherence*, and *overall quality*. Scores range from 1 to 5.
>
> The table below reports, for each sample type and dimension, the mean ± standard deviation of the scores from each annotator (A1–A3), as well as the average of the three annotators’ means (Mean):
>
> | Sample Type                   | A1          | A2          | A3          | Mean |
> | ----------------------------- | ----------- | ----------- | ----------- | ---- |
> | Original – Readability        | 4.29 ± 0.79 | 4.03 ± 0.96 | 4.57 ± 0.64 | 4.30 |
> | Adversarial – Readability     | 3.84 ± 1.07 | 3.80 ± 1.18 | 4.30 ± 0.70 | 3.98 |
> | Original – Coherence          | 4.59 ± 0.60 | 4.08 ± 0.64 | 4.39 ± 0.55 | 4.35 |
> | Adversarial – Coherence       | 3.79 ± 1.30 | 3.84 ± 0.69 | 4.14 ± 0.65 | 3.92 |
> | Original – Overall quality    | 4.24 ± 0.43 | 3.98 ± 0.95 | 4.61 ± 0.56 | 4.28 |
> | Adversarial – Overall quality | 4.06 ± 0.75 | 3.48 ± 1.08 | 4.22 ± 0.63 | 3.92 |
>
> * Since the scores are bounded in [1, 5], the `mean ± std` interval should not be interpreted as the actual attainable score range, but only as an indicator of the dispersion of the annotations.
>
> From this table, we observe that adversarial examples remain highly readable and coherent: all mean scores are close to or above 4.0 on a 1–5 scale, indicating that annotators still perceive them as natural and easy to understand. At the same time, there is a consistent but moderate degradation when moving from original to adversarial texts: the average drops are about 0.32 for readability (4.30 → 3.98), 0.43 for coherence (4.35 → 3.92), and 0.36 for overall quality (4.28 → 3.92). This pattern suggests that TASTE’s perturbations slightly reduce perceived quality-as expected for adversarial modifications-yet do **not** introduce severe grammatical errors or destroy semantic coherence. We also note that some annotators (e.g., A2 on overall quality) are more sensitive to the artifacts introduced by the attack and give lower scores, while others show milder drops, which is consistent with typical inter-annotator variability in human evaluation.
>
> We will incorporate this human evaluation study into the revised version of the paper and add further details on the evaluation protocol to more clearly demonstrate the readability and naturalness of our adversarial examples.
>
> If you have any further questions about our paper, we would be very happy to discuss them during the rebuttal period and will respond to your concerns as promptly as possible.

---

> > ### Comment · Reviewer_9HJS · 2025-11-18
> >
> > Thanks the authors for the detailed rebuttal. I have raised my score, good luck.

---

> ### Author Response · Authors · 2025-11-18
> **Thank you!**
>
> Dear Reviewer 9HJS,
>
> We sincerely appreciate your careful reading of our rebuttal and revised manuscript. Your comments on low-resource scenarios and human evaluation were very helpful in strengthening the paper. Thank you again for your time and thoughtful feedback.

---

### Author Response · Authors · 2025-11-16

Dear AC and Reviewers,

We sincerely thank you for your time and thoughtful feedback. We are very encouraged that our method is seen as **“a novel translation-based adversarial training framework”** (aEgz), whose training procedure is **“useful and enhance the effectiveness of MGT detectors”** and **“model-agnostic and can be used on existing detectors”** (9HJS), and that it **“combine several existing well-performing methods … in a non-trivial manner”** (CasE). We appreciate that our problem is recognized as **“a critical and timely subject” in low-resource LLM-generated text detection"** (CasE), and that TASTE is viewed as **“enhancing robustness and cross-lingual generalization” with “significant improvements in F1 score”** (aEgz). We are also grateful for the positive comments on clarity and presentation, including **“exceptional clarity” of figures, “underlying assumptions are sound”, and “writing is accessible and easy to understand”**(S8Qh), as well as **“the manuscript and the underlying ideas are both overall excellent”** (CasE) and **“the motivation and logic of this paper is clear”**(9HJS).

In our rebuttal, we have carefully addressed every weakness (W) and question (Q):

* **Reviewer 9HJS.**
  We clarified that TASTE is explicitly designed for zero-shot low-resource languages and already outperforms all baselines on languages without dictionaries. We added a dictionary coverage study (10–100%) showing that even small/noisy dictionaries still bring substantial gains over the best baseline, and we conducted a human evaluation (3 annotators, 100 pairs, 3 dimensions) confirming that adversarial examples remain readable and coherent, with only moderate and expected degradation.

* **Reviewer CasE.**
  We clarified that the full implementation (code, requirements, documentation) has been in the supplementary material since before the review phase, and we will release code, datasets, and checkpoints on GitHub and Hugging Face upon acceptance. We added TPR@FPR=1% results on M4 (where TASTE achieves the best cross-lingual mean TPR) and TPR/FPR on SemEval-2024 (OOD) at the same thresholds, and we detailed the token→word mapping for dictionary substitution, how LAAL with gradient reversal suppresses language-specific cues, how performance degrades gracefully as dictionary errors increase, and the methodological differences from RADAR that explain TASTE’s stronger multilingual robustness.

* **Reviewer S8Qh.**
  We enriched the evaluation by reporting training cost (about 40 minutes for 3 epochs on a single RTX 4090) and adding a full inference-time comparison, showing that encoder-based detectors like TASTE have much lower latency than metric-based methods, which is crucial for deployment. We clarified fairness by noting that all adversarially trained methods (RADAR, GREATER-D, TASTE) share the same mBERT backbone and setup, and we showed that under deployment-relevant metrics (TPR@FPR=1%), zero-shot settings, and OOD robustness, TASTE achieves state-of-the-art average performance across languages and attacks. We summarized our existing loss and hyperparameter ablations (attack strength and dictionary quality) and explained our focus on encoder-only detectors (efficiency, calibration, stable thresholds), while pointing to LLM-based detectors as future work.

* **Reviewer aEgz.**
  We added extensive ablations on importance estimators, comparing attention-based, perturbation-based, and random variants against our gradient-based design on multilingual accuracy and attack success rate, and found that the gradient-based TASTE gives the best cross-lingual accuracy and lowest mean attack success. We linked the key hyperparameter controlling attack strength to our analysis in Section 6.1 on replacement ratio versus accuracy, robustness, and training time, highlighted that our robustness evaluation already includes strong phrase-level multilingual attacks (back-translation, paraphrasing) on 9 languages and 8 attacks where TASTE has the lowest average attack success rate, and quantified the cost of our co-evolution training (one attacker update and one detector update per batch, about 40 minutes on 10k samples with ~10 GB GPU). We also introduced experiments on pseudo-label noise (0/10/20% flips) and smaller surrogates (ALBERT-base-v2, Pythia-70M), showing that TASTE remains robust to realistic noise levels and does not rely on a large surrogate, and we examined training dynamics, with loss curves and a brief stability discussion to be added in the revised manuscript.

We again thank the AC and all reviewers for their constructive feedback, which has helped us substantially strengthen the paper. If you have any further questions or suggestions, we would be very happy to discuss them during the rebuttal period and will respond as promptly as possible.

---

### Author Response · Authors · 2025-11-18
**Revised version has already uploaded**

Dear AC and Reviewers,

We would like to once again thank you for your careful reading of our submission and for the many constructive comments and suggestions. We have now uploaded a revised version of the manuscript that incorporates all the changes discussed in our rebuttal.

For ease of cross-referencing between your reviews and the revised manuscript, we have highlighted the newly added or substantially revised text in different font colors according to the reviewer whose comments they address: **red** for Reviewer 9HJS, **blue** for Reviewer CasE, **orange** for Reviewer S8Qh, and **green** for Reviewer aEgz. All line numbers below refer to the updated version of the main paper.

* **Reviewer 9HJS (red):**

  * We added a more detailed **dictionary coverage experiment** in **Lines 500–518**, where we vary the coverage from 10% to 100% and report the impact on cross-lingual accuracy.
  * We added a **human evaluation study** of the readability, coherence, and overall quality of adversarial examples in **Lines 1228–1263**, including the evaluation protocol and summary statistics.

* **Reviewer CasE (blue):**

  * We added an evaluation of **TPR@FPR = 1% on the M4 dataset** and reported **TPR and FPR on the SemEval 2024 Task 8 dataset** under the same thresholds in **Lines 1266–1335**, aligning our metrics with current best practices in MGT detection.
  * We revised the discussion around dictionary errors in **Lines 476–500** (corresponding to Line 464 in the original submission) to avoid potential ambiguity about the relationship between dictionary noise and performance in English vs. other languages.

* **Reviewer S8Qh (orange):**

  * We added a detailed discussion of **training and inference cost**, including **end-to-end inference latency and per-sample inference time for all detectors**, in **Lines 1337–1384**, to better contextualize our method relative to both metric-based and model-based detectors.

* **Reviewer aEgz (green):**

  * We included an **ablation study on importance-score definitions** (gradient-based vs. attention-based, perturbation-based, and random baselines) in **Lines 953–1046**.
  * We analyzed the **effect of using smaller surrogate models** (e.g., ALBERT-base-v2, Pythia-70M) in **Lines 1078–1113**.
  * We added an experiment on the **impact of pseudo-label noise** on both accuracy and robustness in **Lines 1115–1180**.
  * We described the **training dynamics and stability** of the attacker–detector co-evolution, including loss behavior over training steps, in **Lines 1386–1403**.

We are very grateful for the time and effort you have invested in reviewing our work, which has helped us substantially improve the clarity, completeness, and empirical support of the paper. If there are any remaining concerns or if further clarification would be helpful on any of the new results or explanations, we would be very happy to continue the discussion and respond as promptly as possible during the rebuttal period.

---

### Author Response · Authors · 2025-11-29
**Final Justification by Authors**

Dear AC,

Thank you for handling our submission. Since the *Official Reviews* were reverted to the pre-discussion state, we provide a compact map from concerns → concrete revisions, together with **discussion outcomes** indicating whether reviewers felt the issues were resolved. **We have uploaded an updated revised version after the latest discussion**, and all line numbers below refer to this **most recently uploaded revised manuscript**.

## TL;DR: What changed (high-level)

We strengthened the paper along five axes:

1. **Low-resource / dictionary availability**: dictionary *coverage* study (10%–100%) and clearer interpretation under noisy/partial dictionaries.
2. **Human realism**: human evaluation of adversarial samples (readability/coherence/overall quality).
3. **Deployment-aware metrics**: **TPR@FPR=1%** on M4 + fixed-threshold **TPR/FPR on SemEval-2024 Task 8 (OOD)**.
4. **Efficiency**: end-to-end inference latency comparison; training cost summarized (with additional breakdown provided during discussion).
5. **Ablations/robustness clarity**: importance-estimator ablations, smaller surrogates, pseudo-label noise, training dynamics, and clearer Eq.(8) component interpretation.

## Where to find key additions in the revised paper (latest upload)

* **Dictionary coverage**: Lines **500–518** (plus revised dictionary discussion Lines **476–500**).
* **Human evaluation**: Lines **1233–1283**.
* **TPR@FPR=1% (M4) + fixed-threshold OOD TPR/FPR (SemEval-2024)**: Lines **1284–1373**.
* **Inference latency table & efficiency discussion**: Lines **1374–1398**.
* **Importance-score ablations**: Lines **950–1064**.
* **Smaller surrogates**: Lines **1103–1124**.
* **Pseudo-label noise**: Lines **1125–1198**.
* **Training dynamics/stability**: Lines **1400–1412**.

## Reviewer-by-reviewer status

### Reviewer 9HJS — **resolved**

* Concerns: low-resource without high-quality dictionaries; request for human evaluation.
* Actions: added coverage study + human eval (see lines above).
* Discussion outcome: reviewer explicitly replied “**I have raised my score**” (**Nov 18, 2025**), indicating concerns were addressed.

### Reviewer CasE — **resolved**

* Concerns: reproducibility/code availability; best-practice metric (TPR@low FPR); OOD FPR under fixed threshold; token→word mapping; LAAL rationale; RADAR comparison; Binoculars/RAID discrepancy; citations on low-FPR sensitivity.
* Actions: clarified code in supplement; added TPR@FPR=1% + OOD TPR/FPR; expanded implementation/method explanations; provided citations supporting low-FPR sensitivity claims.
* Discussion outcome: reviewer stated questions were satisfactorily answered (**Nov 22, 2025**) and “**My questions has been ad(d)ressed and I have adjusted my rating accordingly.**” (**Nov 25, 2025**).

### Reviewer aEgz — **addressed via follow-up; manuscript clarified accordingly**

* Initial concerns: importance-estimator ablations; hyperparameter/attack strength; robustness coverage; compute overhead; stability; surrogate noise/smaller surrogates.
* Actions in revision: importance estimators / smaller surrogates / noise / dynamics (lines above).
* Follow-up (**Nov 27, 2025**): reviewer asked whether “random” implies the attacker is unnecessary, requested clearer “simple code-switching” baseline labeling, phrase-level importance discussion, clearer Eq.(8) ablations, and training-cost breakdown.
* Our discussion response (**Nov 27, 2025**): clarified **Acc = final detector accuracy**; clarified **Random-TASTE = unguided (simple) code-switching baseline**; corrected potential notation confusion; provided a **per-component training-time breakdown**; restated **Eq.(8) component ablations** with explicit definitions. We clarified these points in the forum discussion (Nov 27) and will incorporate them into the camera-ready version if allowed.

### Reviewer S8Qh — **addressed; no further follow-up observed**

* Concerns: training/inference costs; fairness; “suboptimal” performance; stronger ablations on Eq.(8); LLM-based detectors.
* Actions: added inference latency + clarified fairness (shared backbone for adversarial methods); emphasized deployment-aligned metrics and robustness; clarified Eq.(8) ablation interpretation and why “LAAL-only” is degenerate; explained encoder-only scope vs LLM prompting as future work.
* Discussion outcome: no further questions posted after our responses (from **Nov 16, 2025** to **Nov 29, 2025**).

Thank you for considering both the latest revised manuscript and the preserved discussion record. We hope this condensed map helps you quickly verify how each major concern was addressed and which reviewers explicitly indicated resolution.

Sincerely,

The Authors

---

### Meta-Review · Area_Chair_7dXZ · 2026-01-07

**Summary:**

This submission proposes TASTE, a translation‑based adversarial training framework aimed at improving multilingual machine‑generated text (MGT) detection, particularly in zero‑shot languages and under adversarial attacks. The paper received four reviews: three broadly positive and one negative. During the rebuttal and discussion phase, the authors provided extensive clarifications, additional experiments, and new evaluations (e.g., dictionary coverage, human evaluation, TPR@FPR=1%, OOD robustness, efficiency analysis, and expanded ablations). Two reviewers expressed the intention of increasing the score following the rebuttal, and another indicated that their concerns were satisfactorily addressed.

**Reviewer Concerns:**

Across reviewers, the main concerns centered on:

- Dependence on dictionaries and applicability to low‑resource languages

- Readability of adversarial examples

- Use of metrics not aligned with current best practices (e.g., TPR@low FPR)

- Fairness of comparisons and training/inference cost

- Completeness of ablations, especially regarding Eq. (8)

- Novelty and methodological contribution

The authors responded with substantial new empirical results and clarifications. Reviewers 9HJS, CasE, and aEgz expressed that their concerns were resolved after the rebuttal. Reviewer S8Qh did not follow up further after the authors’ detailed responses.

Despite the authors’ efforts, I see some broader concerns remain:

+ Novelty and conceptual contribution:
While the method is well‑engineered and the empirical evaluation is thorough, the core idea—dictionary‑based code‑switching combined with adversarial training—may be viewed as incremental relative to existing adversarial training and multilingual robustness frameworks. The approach largely recombines known components rather than introducing a fundamentally new modeling principle.

+ Broader impact and interest:
The practical relevance of MGT detection remains an evolving and contested area. Recent developments (e.g., the discontinuation of major text‑detection tools due to reliability issues) raise questions about the long‑term viability of detection‑based approaches compared to alternatives such as watermarking. This limits the broader appeal of the work beyond the subcommunity focused on MGT detection.

+ Positioning within the field:
Although the authors provide extensive experiments, the work may still be perceived as niche, with limited implications for mainstream ICLR audiences working on frontier models, multilingual modeling, or adversarial robustness more broadly.

**Reviewer Scores:**

+ Reviewer 9HJS — Explicitly stated after rebuttal that they raised their score. Their concerns about dictionary coverage and human evaluation were fully resolved.

+ Reviewer CasE — Also explicitly confirmed that they adjusted their rating upward. All technical and methodological questions were addressed to their satisfaction.

+ Reviewer aEgz — The authors addressed all concerns, but the reviewer did not explicitly indicate a score change. Most likely the score remained unchanged.

+ Reviewer S8Qh — Did not participate in post‑rebuttal discussion and gave no indication of revising their negative assessment. Score is presumed unchanged.

---

### Decision · Program_Chairs · 2026-01-26

Accept (Poster)